JCB | Journal of Cell Biology

# Unveiling the cell biology of hippocampal neurons with dendritic axon origin

Yuhao Han[1,2,3,4], Daniela Hacker[1], Bronte Catharina Donders[1], Christopher Parperis[9], Roland Thuenauer[5,6,8], Christophe Leterrier[9], Kay Grünewald[3,4,7], and Marina Mikhaylova[1,2]

In mammalian axon-carrying–dendrite (AcD) neurons, the axon emanates from a basal dendrite, instead of the soma, to create a privileged route for action potential generation at the axon initial segment (AIS). However, it is unclear how such unusual morphology is established and whether the structure and function of the AIS in AcD neurons are preserved. By using dissociated hippocampal cultures as a model, we show that the development of AcD morphology can occur prior to synaptogenesis and independently of the in vivo environment. A single precursor neurite first gives rise to the axon and then to the AcD. The AIS possesses a similar cytoskeletal architecture as the soma-derived AIS and similarly functions as a trafficking barrier to retain axon-specific molecular composition. However, it does not undergo homeostatic plasticity, contains lesser cisternal organelles, and receives fewer inhibitory inputs. Our findings reveal insights into AcD neuron biology and underscore AIS structural differences based on axon onset.

## Introduction

Neurons in the mammalian brain typically have a single axon and multiple highly branched dendrites that emerge directly from the soma. Mostly, excitatory synaptic inputs are received at the dendrites and propagate along them toward the soma in the form of membrane depolarization. Following somatic integration and upon reaching a specific threshold at the axon initial segment (AIS), these inputs trigger the generation of action potentials (APs)—a crucial mechanism enabling neurotransmission.

In addition to this classical view of neuronal morphology, it has been reported that the axon can also emerge from a basal dendrite (Gonda et al., 2023; Hodapp et al., 2022; Thome et al., 2014; Triarhou, 2014; Wahle et al., 2022). Already in 1899, Ramón y Cajal observed such neurons in invertebrate abdominal ganglia (Triarhou, 2014). Later, neurons with dendritic axon origin were also discovered in the neocortex and hippocampus of mammalian brains, including those of humans (Gonda et al., 2023; Hodapp et al., 2022; Peters et al., 1968; Thome et al., 2014; Wahle et al., 2022). Given their axonal origin, these cells are classified as axon-carrying–dendrite (AcD) neurons. A typical feature of AcD neurons is that the axon and the AcD branch out from a common dendritic root, which is referred to as the stem dendrite (Hodapp et al., 2022; Thome et al., 2014; Wahle et al., 2022). In the CA1 region of the hippocampus of adult rodents, AcD neurons represent ~50% of the entire excitatory neuron population (Thome et al., 2014). However, in young animals, AcD neuron incidence in the same region is only ~20% (Benavides-Piccione et al., 2020; Thome et al., 2014), indicating that there might be an age-dependent increase in the prevalence of AcD neurons. Electrophysiological experiments have suggested that, in AcD neurons, inputs arriving at the AcD have a higher probability of triggering APs than those arriving at the regular somatic dendrites (Thome et al., 2014). Since the axon is adjacent to the AcD, synaptic inputs from the AcD can bypass somatic integration and directly flow into the axon as APs (Thome et al., 2014). This novel AcD-to-axon paradigm of AP transmission was shown to allow AcD neurons to evade global peri-somatic inhibition and can therefore be selectively activated during sharp-wave ripples (Hodapp et al., 2022), a process associated with memory consolidation. Collectively, these findings revealed that the dendritic axon origin has a substantial impact on the electrophysiological behavior of hippocampal excitatory neurons. Despite their physiological importance, the cellular features of AcD neurons and the developmental sequence leading to such morphology have remained largely unexplored.

[1]AG Optobiology, Institute of Biology, Humboldt Universität zu Berlin, Berlin, Germany; [2]AG "Neuronal Protein Transport", Centre for Molecular Neurobiology, University Medical Center Hamburg-Eppendorf, Hamburg, Germany; [3]Centre for Structural Systems Biology, Hamburg, Germany; [4]Structural Cell Biology of Viruses, Leibniz Institute of Virology (LIV), Hamburg, Germany; [5]Advanced Light and Fluorescence Microscopy (ALFM) Facility, Centre for Structural Systems Biology, Hamburg, Germany; [6]Technology Platform Light Microscopy, University of Hamburg, Hamburg, Germany; [7]Department of Chemistry, University of Hamburg, Hamburg, Germany; [8]Technology Platform Microscopy and Image Analysis (TP MIA), Leibniz Institute of Virology (LIV), Hamburg, Germany; [9]Aix Marseille Université, CNRS, INP UMR7051, NeuroCyto, Marseille, France.

Correspondence to Kay Grünewald: kay.gruenewald@cssb-hamburg.de; Marina Mikhaylova: marina.mikhaylova@hu-berlin.de.

The central dogma of neuron development is that neurons break their symmetry by initially forming the axon (Dotti et al., 1988; Schelski and Bradke, 2017; Takano et al., 2015; Yogev and Shen, 2017). Once the axon is established, the remaining neurites mature into dendrites, and synaptic connections between neurons are eventually formed (Dotti et al., 1988). This developmental process is mostly genetically encoded and is reproduced in dissociated cultures (Dotti et al., 1988; Schelski and Bradke, 2017; Takano et al., 2015; Wit and Hiesinger, 2023; Yogev and Shen, 2017). However, the timing, topology of subcellular domains, connectivity, and certain other specific aspects of development in vivo can vary between neuron types. This variation may be attributed to the distinct microenvironments formed by guidance molecules (Schelski and Bradke, 2017; Takano et al., 2015; Yogev and Shen, 2017). The morphology of AcD neurons poses a great challenge to this canonical sequence of neuron development. To date, it is unclear whether AcD neurons adhere to the previously described classical developmental stages (Dotti et al., 1988; Takano et al., 2015). It has yet to be determined whether differentiation into the AcD type is intrinsically encoded, or whether specific connectivity patterns, synaptic inputs, and gradient of guidance molecules are necessary.

The cytoskeleton plays an instrumental role in the establishment and maintenance of neuronal axo-dendritic polarity. During development, the cytoskeleton of the premature axon acquires distinctive characteristics, such as the uniform plus-end-out orientation of microtubules (MTs) (Katrukha et al., 2021; Tas et al., 2017; Yau et al., 2016). Axonal MTs are also stabilized by accumulating specific posttranslational modifications (PTMs) on tubulins, such as acetylation and detyrosination (Hammond et al., 2008; Katrukha et al., 2021; Park and Roll-Mecak, 2018; Tas et al., 2017). Contrarily, dendrites are characterized by a lower ratio of acetylated/tyrosinated MTs (Hammond et al., 2008; Kapitein and Hoogenraad, 2011) and a mixed orientation of MT plus ends (Kapitein and Hoogenraad, 2011; Tas et al., 2017; Yau et al., 2016). The unique axonal MT orientation together with PTMs facilitates the growth and maturation of the axon by enabling the targeted delivery of vesicles containing axon-specific proteins and lipids via kinesin and dynein motor proteins (Kapitein and Hoogenraad, 2011; Lipka et al., 2016). However, it is still an open question how axonal cargoes could be targeted to the axon in AcD neurons where axonal vesicles first must pass through the stem dendrite.

The AIS is a specialized structure located at the proximal part of the axon, extending ~20–60 μm along the axon (Leterrier, 2018; Rasband, 2010). The AIS cytoskeleton is highly ordered, with a molecular organization that is distinct from the rest of the axon. Individual MTs within the AIS are fasciculated by the AIS-specific MT-associated protein (MAP) TRIM46 (Harterink et al., 2019; Van Beuningen et al., 2015). Tetramers of scaffolding protein αII-spectrin and the AIS-specific scaffolding protein βIV-spectrin are localized in-between periodically arranged rings of filamentous actin (F-actin) to define the so-called membrane-associated periodic cytoskeleton (MPS) (Leterrier, 2018; Leterrier et al., 2015; Rasband, 2010; Vassilopoulos et al.,

2019). F-actin additionally forms intracellular patches along the AIS (Arnold and Gallo, 2014; Balasanyan et al., 2017). The master AIS organizer protein ankyrin-G (AnkG) binds to βIV-spectrin and associates the entire assembly with the plasma membrane (Fréal et al., 2016; Leterrier, 2018; Rasband, 2010). This specialized cytoskeleton provides a scaffolding platform for anchoring membrane proteins, such as cell adhesion molecules and voltage-gated ion channels. Another unique feature of the AIS is the presence of the endoplasmic reticulum specializations called the cisternal organelle, which is putatively responsible for $Ca^{2+}$ storage and release (Benedeczky et al., 1994; Konietzny et al., 2019; Bas Orth et al., 2007; Sánchez-Ponce et al., 2011).

The AIS serves not only as a molecular barrier to prevent somatodendritic proteins from entering the axon (Arnold and Gallo, 2014; Balasanyan et al., 2017; Leterrier, 2018; Rasband, 2010; Watanabe et al., 2012) but also as a trigger to initiate APs along the axon and thereby regulate neuronal excitability and homeostasis (Bender and Trussell, 2009; Hu et al., 2009; Leterrier, 2018; Rasband, 2010). It has been shown both in vivo and in vitro that the AIS of excitatory neurons undergoes structural remodeling to compensate for increased neuronal activity, a process known as AIS plasticity (Evans et al., 2013, 2015, 2017; Grubb and Burrone, 2010; Jahan et al., 2023; Jamann et al., 2021; Kole and Stuart, 2012; Kuba et al., 2006; Susuki and Kuba, 2016; Wefelmeyer et al., 2015; Yamada and Kuba, 2016). Furthermore, the AIS of excitatory neurons is innervated by inhibitory interneurons to form axo-axonic synapses for fine-tuning excitability during network activities (Nathanson et al., 2019). It is currently unknown how similar the AIS features of AcD neurons are to nonAcD neurons.

In this work, we seek to provide a more comprehensive cell biological profile of hippocampal AcD neurons by using primary hippocampal culture as a model system. We focused on the developmental processes of these neurons and provided a detailed characterization of the AIS, including its structural and functional properties, under both basal and enhanced neuronal activity conditions.

## Results

### Development of AcD neurons does not require specific neuronal connectivity patterns and extracellular guidance cues

In the brain, the gradient of guidance molecules, neuronal cell type identity, and specific connectivity patterns are major forces that shape neuronal morphology during differentiation (Schelski and Bradke, 2017; Takano et al., 2015; Yogev and Shen, 2017). To investigate how AcD neurons develop, we used dissociated hippocampal neurons as a simplified model system. We hypothesized that if AcD neurons form in dissociated cultures prior to synaptogenesis, factors other than specific neuronal interactions and the gradient of extracellular cues must be involved. Interestingly, immunofluorescent staining of primary neurons with AnkG (an AIS marker) and MAP2 (a somato-dendritic marker) clearly showed that a subset of neurons displays the AcD morphology (Fig. 1 A), as the AIS branches out from a dendrite and is located away from the

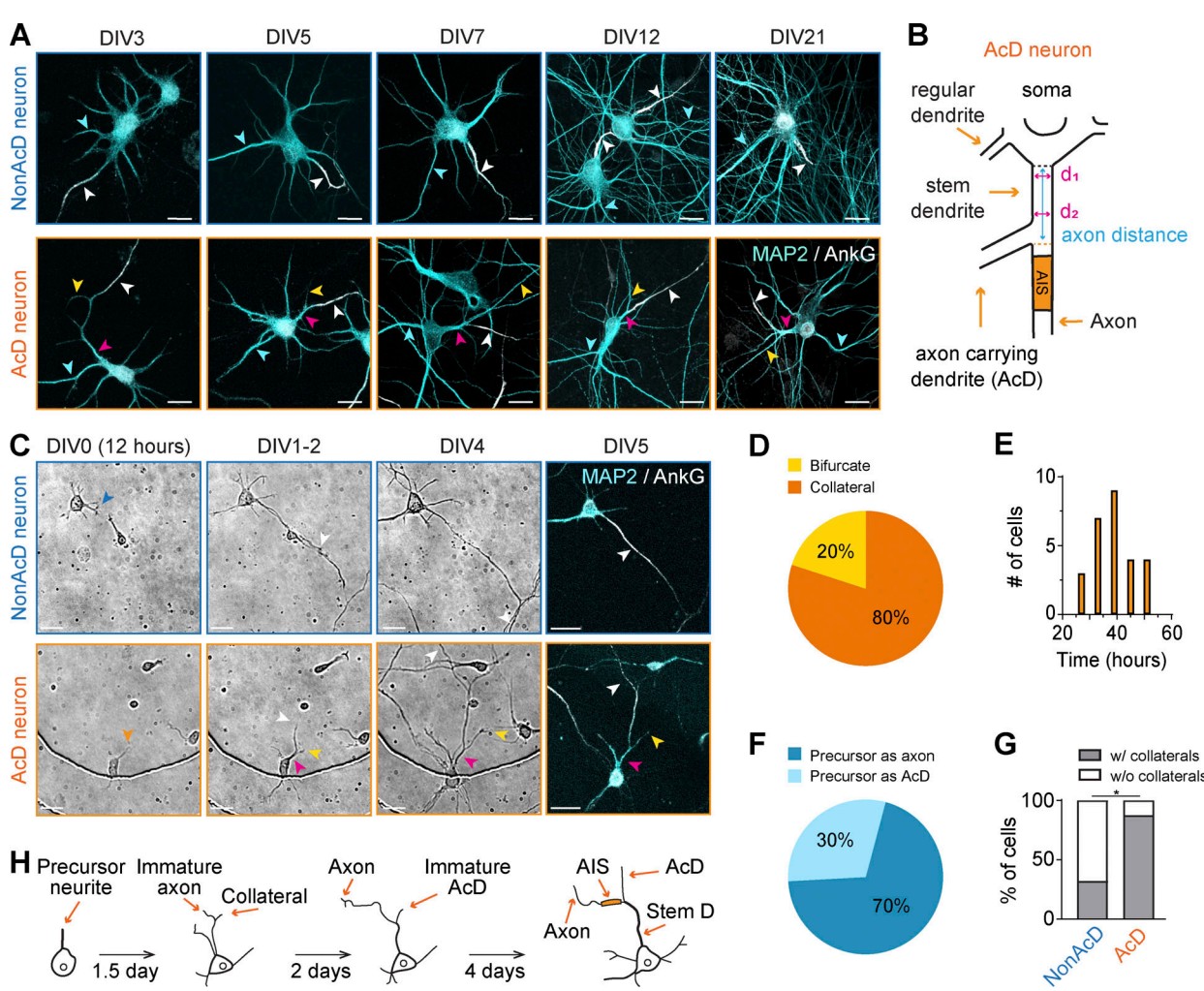

Figure 1. **Development of neurons with axon-carrying dendrite. (A)** Representative images of dissociated hippocampal primary neurons with nonAcD (top row) and AcD (bottom row) morphology at different ages. AnkG and MAP2 immunostaining indicate the axon/AIS and dendrites of a neuron, respectively. White arrowhead indicates AIS; yellow and pink arrowhead indicates AcD and stem dendrite of AcD neuron respectively; and cyan arrowhead indicates regular dendrite. Scale bar is 20 μm. **(B)** Schematic of AcD neuron. Black dashed line indicates the end of soma; orange dashed line indicates the beginning of an axon. Cyan solid line with double arrowheads indicates the axon distance. Magenta solid line with double arrowheads indicates the first ($d_1$) and the second ($d_2$) line used to calculate the averaged diameter of stem dendrite. **(C)** Time-lapse images of nonAcD (top row) and AcD neurons (bottom row) at different developmental stages. The precursor neurite in the displayed AcD neuron became an axon and collateral from the precursor neurite developed as an AcD. Orange arrowhead indicates the precursor neurite of AcD neuron; blue arrowhead indicates the precursor neurite of nonAcD neuron; white arrowhead indicates axon; yellow and pink arrowhead indicates AcD and stem dendrite of AcD neuron, respectively. Scale bar is 20 μm. Cell bodies with dendrites and the AIS are labeled by MAP2 and AnkG staining, respectively. Related to Videos 1 and 2. **(D)** Percentage of AcD neurons with a generated collateral or bifurcation of the precursor neurite. 1 culture preparation, AcD (total): $n = 27$ cells, collateral: $n = 22$ cells (∼80%), bifurcation: $n = 5$ cells (∼20%). **(E)** Time points of collateral formation at the precursor neurite in AcD neurons. One neuronal culture preparation, AcD (total): $n = 27$ cells, time points: 27 h ($n = 3$ cells), 33 h ($n = 7$ cells), 39 h ($n = 9$ cells), 45 h ($n = 4$ cells), 51 h ($n = 4$ cells). **(F)** Percentage of AcD neurons developing the precursor neurite into an axon or an AcD. 1 culture preparation, AcD: $n = 22$ cells, precursor as axon: $n = 16$ cells (∼70%), precursor as AcD: $n = 6$ cells (∼30%). **(G)** Percentage of nonAcD and AcD neurons that formed collaterals at the proximal region of the precursor neurite during development. One culture preparation, nonAcD: $n = 25$ cells (percentage of neurons with collaterals versus without collaterals: ∼32% versus ∼68%), AcD: $n = 23$ cells (percentage of neurons with collaterals versus without collaterals: ∼90% versus ∼10%). Chi-Square test, *$P < 0.05$. **(H)** Proposed model of AcD neuron development.

soma (Fig. 1 A). Already at 3 days in vitro (DIV), AcD neurons were present and the AnkG fluorescent signal began to accumulate at the proximal axon region (Fig. 1 A). AcD neurons were also found in cultures fixed at a later time point (DIV5, 7, 12, and 21) (Fig. 1 A), and the AnkG signal appeared to be more continuous and evident as the AIS becomes more mature (Fig. 1 A). The same timeline of AIS formation was also observed in nonAcD neurons (Fig. 1 A), which is consistent with a previous study (Fréal et al., 2016). Together, these results suggest that

the dendritic axon origin has no impact on the timing of AIS assembly, and the development of AcD neurons can occur independently of the in vivo environment and specific connectivity patterns.

For further investigation, we classified hippocampal neurons into AcD and nonAcD categories based on previously published standards (Hodapp et al., 2022; Thome et al., 2014) (Fig. 1 B and Fig. S1 A; see Materials and methods for details). The axon distance, which refers to the distance from the starting point of an

axon to the adjacent ending point of a corresponding cell body (Fig. 1 B and Fig. S1 A), and stem dendrite diameter (Fig. 1 B and Fig. S1 A) were used as main factors for AcD neuron classification. We considered neurons as AcD when their axon emerged from a MAP2-positive dendrite, with an axon distance longer than 2 µm and larger than the diameter of the stem dendrite. Otherwise, neurons were considered as nonAcD. To confirm whether AcD neurons included in our study comply with the classification standard, we measured their axon distance and stem dendrite diameter. We noticed that the axon distance of AcD neurons is widely spread (2–30 µm) with a median of 7.7 µm and is larger than the diameter of stem dendrite (Fig. S1, B–D). We also compared the AIS diameter between AcD and nonAcD neurons, and the difference is not statistically significant (Fig. S1 E).

Next, we quantified the percentage of AcD neurons in dissociated cultures at DIV5 and DIV7, at which neuronal polarity has just been established, and at the age of DIV21 when neurons are fully mature. We found that AcD neurons make up 15–20% of the entire population at DIV5 and DIV7 which then decreased to 10% at DIV21 (Fig. S1 F), suggesting that dissociated culture is a suitable model to further investigate the cell biology of AcD neurons.

## AcD neurons follow the classical developmental process and mainly generate the AcD from an axonal collateral

The developmental process giving rise to nonAcD neurons is classically divided into five stages (Dotti et al., 1988). First, the neuron attaches to a suitable substrate and a small number of nascent neurites form, one of which is targeted for axon development (stage 1–3; DIV0–3; Fig. 1 C and Video 1). The remaining neurites differentiate further and become dendrites (stage 4; DIV4–7; Fig. 1 C and Video 1). Finally, synaptic contacts are established (stage 5; after DIV7). The equivalent developmental sequence that produces AcD neurons has not been similarly characterized. Here, we conducted continuous time-lapse recordings of dissociated neurons from DIV0 to DIV5 and subsequently performed immunostaining with AnkG and MAP2 for post-hoc categorization of neuronal types. Time-lapse imaging data revealed that AcD neurons also first grow an axon from a precursor neurite and then establish the dendrites (Fig. 1 C and Fig. S1 G; and Video 2). Notably, during early developmental stages (DIV1–2), a collateral neurite was generated from the proximal region of the precursor neurite that is designated as the axon (Fig. 1 C and Fig. S1 G; and Video 2). This collateral underwent several rounds of elongation and retraction while the precursor neurite kept developing as the axon (Fig. S1 G and Video 2). Eventually, once the axon was defined, this collateral started to mature as the AcD and the former proximal region of the axon precursor neurite was transformed into the stem dendrite (Fig. 1 C and Fig. S1 G; and Video 2).

Quantitative analysis showed that about 80% of the AcD neurons generated a collateral at the proximal region of the precursor neurite (Fig. 1 D) and that the average time point of collateral formation was 39 h (DIV1.5) after plating (Fig. 1 E). About 20% of the AcD neurons resulted from a bifurcation of the precursor neurite growth cone to form the axon and AcD (Fig. 1 D and Fig. S1 H; and Video 3 B). Among the 80% of AcD neurons that formed collateral at the precursor neurite, most cells (70%) designated the precursor neurite as the axon and formed the collateral as AcD (Fig. 1 F and Fig. S1 G). Only a smaller population (30%) of neurons instead developed the collateral as the axon and eventually turned the precursor neurite into the AcD (Fig. 1 F and Fig. S1 G; and Video 3 A). We also assessed the occurrence of collateral genesis at the proximal region of the axon precursor neurite in nonAcD neurons. We found that the percentage of nonAcD neurons that formed collaterals at DIV1 is much lower than for AcD neurons (32% nonAcD neurons versus 90% AcD neurons; Fig. 1 G). Our findings suggest that AcD neurons follow the canonical in vitro developmental sequence (Dotti et al., 1988) observed in nonAcD neurons by first establishing the axon and then the dendrites (Fig. 1 H). AcDs are derived mainly from a collateral formed at the basal region of immature axons during early development (Fig. 1 H). However, alternative strategies exist during AcD neuron development, which are similar to what has been shown for nonAcD neurons in vitro (Dotti et al., 1988).

## The stem dendrite of AcD neurons has an axon-like MT orientation and is enriched in tyrosinated MTs

In nonAcD neurons, axons and dendrites have key differences in the cytoskeletal composition of MT orientations and PTMs which play a crucial role in guiding selective cargo transport (Kapitein and Hoogenraad, 2011; Katrukha et al., 2021; Lipka et al., 2016; Park and Roll-Mecak, 2018; Tas et al., 2017; Yau et al., 2016). We next asked whether these differences are conserved in the case of AcD neurons. We were particularly interested in the stem dendrite that connects both the axon and AcD to the soma. To track the growing MT plus ends, we moderately overexpressed MT plus-end binding protein-3 tagged with tdTomato (EB3-tdTomato) in mature (DIV14) dissociated hippocampal cultures. For axon identification, we live-labeled the AIS with a fluorescently conjugated antibody against the extracellular domain of the AIS-enriched membrane protein neurofascin (anti-NF-CF640R) (Fig. S1 I). We found that the AIS, axon, and regular somatic dendrites of AcD neurons displayed MT orientations similar to nonAcD neurons. As indicated by EB3-tdTomato trajectories, MT in the AIS and axon of both AcD and nonAcD neurons are uniformly oriented with the plus end towards the distal part of the axon but have mixed orientations in the regular dendrites (Fig. 2, A and C; and Video S4, A and B; and Video 5). We also observed the typical dendritic MT orientations (Yau et al., 2016) in the AcD, where the MT plus ends grew toward both directions (Fig. 2, B and C; and Video 4 A). By stark contrast, we found that in the stem dendrite of AcD neurons, nearly 90% of the EB3-Tdtomato trajectories showed a unidirectional (plus end-out) movement either toward the axon or the AcD (Fig. 2, B and C; and Video 4 A). This unidirectional plus end-out MT orientation in the stem dendrite highly resembles that of the axon. Interestingly, we found that in AcD neurons, the appearance of EB3 comets in the stem dendrite region is higher than in the axon and regular proximal dendrites (Fig. 2 D). This indicates that the stem dendrite may contain a higher number of dynamic MTs. We did not notice

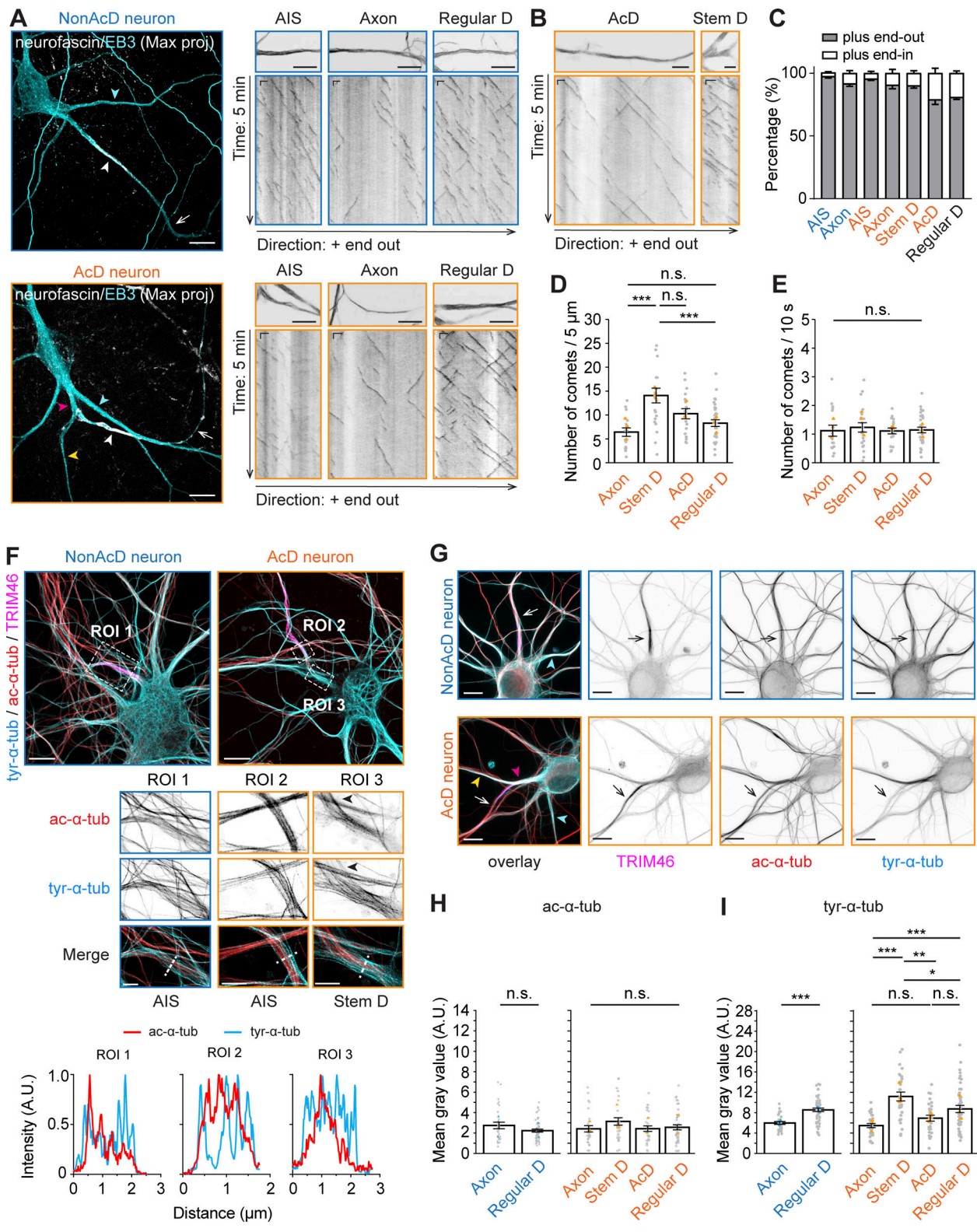

Figure 2. **The stem dendrite of AcD neuron exhibits axon-like MT organization. (A)** Left panel: Maximum intensity projection of DIV14 nonAcD (top) and AcD neurons (bottom) transfected with EB3-tdTomato for visualization of MT plus ends. The AIS is live-labeled with anti-NF-CF640R antibody. White arrowhead indicates the AIS and white arrow indicates the axon. Cyan arrowhead indicates regular somatic dendrite (Regular D); yellow and pink arrowhead indicates the AcD and stem dendrite (Stem D) of AcD neuron, respectively. Scale bar is 10 μm. Right panel: 5 min time projection and kymograph of EB3-tdTomato in the AIS, axon, and regular somatic dendrite (Regular D) of nonAcD (top) and AcD neuron (bottom) shown in the left panel. Scale bar is 10 μm. Kymograph scale is 10 s (vertical) and 2 μm (horizontal). Related to Video 4, A and B; and Video 5. **(B)** 5 min time projection and kymograph of EB3-tdTomato in the AcD and the stem dendrite (Stem D) of AcD neuron shown in A. Scale bar is 5 μm. Kymograph scale is 10 s (vertical) and 2 μm (horizontal). Related to Video

4 A. **(C)** Percentage of MT plus end orientations in different regions of nonAcD and AcD neurons. Mean ± SEM, three independent cultures, AIS (nonAcD) $n = 23$, AIS (AcD) $n = 17$, axon (nonAcD) $n = 21$, axon (AcD) $n = 15$, stem dendrite (Stem D) $n = 19$, AcD $n = 17$, regular somatic dendrite (Regular D) $n = 67$. **(D)** Density of EB3-tdTomato comets in different regions of AcD neurons. Mean ± SEM, three independent cultures, axon $n = 16$, stem dendrite (Stem D) $n = 21$, AcD $n = 18$, regular dendrite (Regular D) $n = 32$. Grey dot indicates value of individual cell. Orange triangle indicates mean of each independent culture. One-way ANOVA with Tukey's multiple comparisons test, *P < 0.05, **P < 0.01, ***P < 0.001. **(E)** Growth rate of EB3-tdTomato comets in different regions of AcD neurons. Mean ± SEM, 3 independent cultures, axon $n = 16$, stem dendrite (Stem D) $n = 21$, AcD $n = 18$, regular dendrite (Regular D) $n = 32$. Grey dot indicates the value of individual cell. Orange triangle indicates mean of each independent culture. One-way ANOVA with Tukey's multiple comparisons test, no significance (n.s.) P > 0.05. **(F)** Top row: Representative confocal images of DIV10 nonAcD and AcD neurons stained for tyrosinated (tyr) and acetylated (ac) tubulin, and the AIS marker TRIM46. Scale bar is 10 µm. Middle row: Single plane 2D gSTED image of tyrosinated and acetylated MTs corresponding to ROIs in top row. Scale bar is 2.5 µm. ROI1 is the axon (AIS region) of the displayed nonAcD neuron. ROI2 and ROI3 are the axon (AIS region) and stem dendrite of the displayed AcD neuron, respectively. Black arrowhead in ROI3 indicates the start of the axon. Bottom row: Intensity profile of tyrosinated and acetylated MTs indicated by white dashed lines in ROIs shown in middle row. **(G)** Representative confocal images of nonAcD and AcD neurons stained for tyrosinated (tyr) and acetylated (ac) tubulin, and the AIS marker TRIM46. Scale bar is 10 µm. White arrow in overlay image and black arrow in single-channel image indicates the proximal axon (AIS region). Cyan arrowhead in overlay image indicates the proximal part of the regular dendrite (Regular D); yellow and pink arrowhead in overlay image indicates the proximal part of the AcD and stem dendrite (Stem D) of AcD neuron, respectively. **(H)** Quantification of acetylated α-tubulin (ac-α-tub) fluorescent intensity in different regions of nonAcD (left) and AcD neurons (right). Mean ± SEM, three independent cultures, nonAcD neuron: proximal axon: $n = 29$ cells, proximal somatic dendrite (Regular D): $n = 52$ cells, AcD neuron: proximal axon: $n = 28$ cells, stem dendrite (Stem D): $n = 30$ cells, proximal AcD: $n = 30$ cells, proximal somatic dendrite (Regular D): $n = 43$ cells. Grey dot indicates value of individual cell. Orange and cyan triangle indicates mean of each independent culture. Mann–Whitney test (two-sided) for nonAcD neurons: not significant (n.s.) P > 0.05. One-way ANOVA with Tukey's multiple comparisons test for AcD neurons, no significance (n.s.) P > 0.05. **(I)** Quantification of tyrosinated α-tubulin (tyr-α-tub) fluorescent intensity in different regions of nonAcD (left) and (right) neurons. Mean ± SEM, three independent cultures, nonAcD neuron: proximal axon: $n = 29$ cells, proximal somatic dendrite (Regular D): $n = 53$ cells, AcD neuron: proximal axon: $n = 30$ cells, stem dendrite (Stem D): $n = 28$ cells, proximal AcD: $n = 29$ cells, proximal dendrite (Regular D): $n = 43$ cells. Grey dot indicates value of individual cell. Orange and cyan triangle indicate mean of each independent culture. Mann–Whitney test (two-sided) for nonAcD neurons: ***P < 0.001. One-way ANOVA with Tukey's multiple comparisons test for AcD neurons, *P < 0.05, **P < 0.01, ***P < 0.001. More detailed statistical information see Data S1.

any significant difference in EB3 comet density when other regions were compared (Fig. 2 D), and the growth rate of EB3 comets was similar between all analyzed compartments of AcD neurons (Fig. 2 E).

Next, we performed gated-stimulation emission depletion (gSTED) super-resolution microscopy to visualize MT PTMs in the axon and stem dendrite of AcD neurons. Specifically, we looked into MT tyrosination and acetylation, which are common PTMs that represent dynamic and stable MTs, respectively (Konishi and Setou, 2009; McKenney et al., 2016; Park and Roll-Mecak, 2018; Peris et al., 2009, 2022). Previous studies suggested that acetylated and tyrosinated MTs have different spatial arrangements in neurites (Katrukha et al., 2021; Tas et al., 2017). Furthermore, the acetylated MTs are more enriched in the axon (Kapitein and Hoogenraad, 2011), and this instructs the transport of axonal cargo driven by the kinesin-1 motor protein (Katrukha et al., 2021; Nakata and Hirokawa, 2003). Our data showed that both acetylated and tyrosinated MTs are present in the axon of AcD neurons and form bundles that are spatially segregated (Fig. 2 F). The acetylated MTs are placed near the central core of the axon and surrounded by tyrosinated MTs (Fig. 2 F). This spatial arrangement of MTs is the same as in the axon of nonAcD neurons (Fig. 2 F) and is also consistent with the previous results (Tas et al., 2017).

Of note, we observed that bundles of acetylated MT were enveloped by tyrosinated MTs (Fig. 2 F) and extended from the stem dendrite directly into the axon (Fig. 2 F). To test if there are compartment-specific accumulations of these PTMs, we performed super-resolved spinning-disk confocal microscopy and measured the total fluorescence intensity of tyrosinated and acetylated MTs in different regions of DIV10 AcD and nonAcD neurons. Quantification showed no differences in the abundance of acetylated MTs between the stem dendrite, the proximal part of the axon, the proximal part of the AcD, and the proximal part

of somatic dendrites in AcD neurons (Fig. 2, G and H). Also, no significant differences were found between neurites in nonAcD neurons (Fig. 2, G and H). Conversely, the presence of tyrosinated MTs was higher in all somatic dendrites of AcD and nonAcD neurons, and tyrosinated MTs were especially enriched in the stem dendrite of AcD neurons (Fig. 2, G–I). This enrichment of tyrosinated MTs at the stem dendrite of AcD neurons correlates with a higher probability of EB3 comet appearance in this region (Fig. 2 D). Altogether, our data suggest that the dendritic and axonal MTs in AcD neurons are similarly organized as in nonAcD neurons. The stem dendrite of AcD neurons likely inherits axonal MT orientation from its development and is enriched in dynamic MTs.

### The AIS of AcD neurons has similar cytoskeletal organization as nonAcD neurons

The AIS has a distinct cytoskeletal organization that helps neurons to maintain polarity by segregating dendritic and axonal proteins (Leterrier, 2018; Rasband, 2010). We next proceeded with characterizing the cytoskeletal structure of the AcD neuron's AIS. An important feature of the AIS cytoskeleton that has been observed in nonAcD neurons is the presence of MT fascicles mediated by TRIM46 (Harterink et al., 2019; Van Beuningen et al., 2015). Similarly, we found that the TRIM46 signal is clearly present and accumulates at the AIS of AcD neurons (Fig. 2, F and G), indicating that the MT fasciculation remains.

Another feature of the AIS is the presence of submembrane F-actin rings of the MPS (Costa et al., 2020; Leterrier, 2018; Rasband, 2010) and intracellular F-actin patches (Al-Bassam et al., 2012; Arnold and Gallo, 2014; Balasanyan et al., 2017; Watanabe et al., 2012). To visualize F-actin at the AIS, we labeled dissociated neurons with phalloidin and the AIS-specific scaffolding protein βIV-spectrin and performed gSTED imaging. We found that F-actin at the AIS of AcD neurons also formed patches

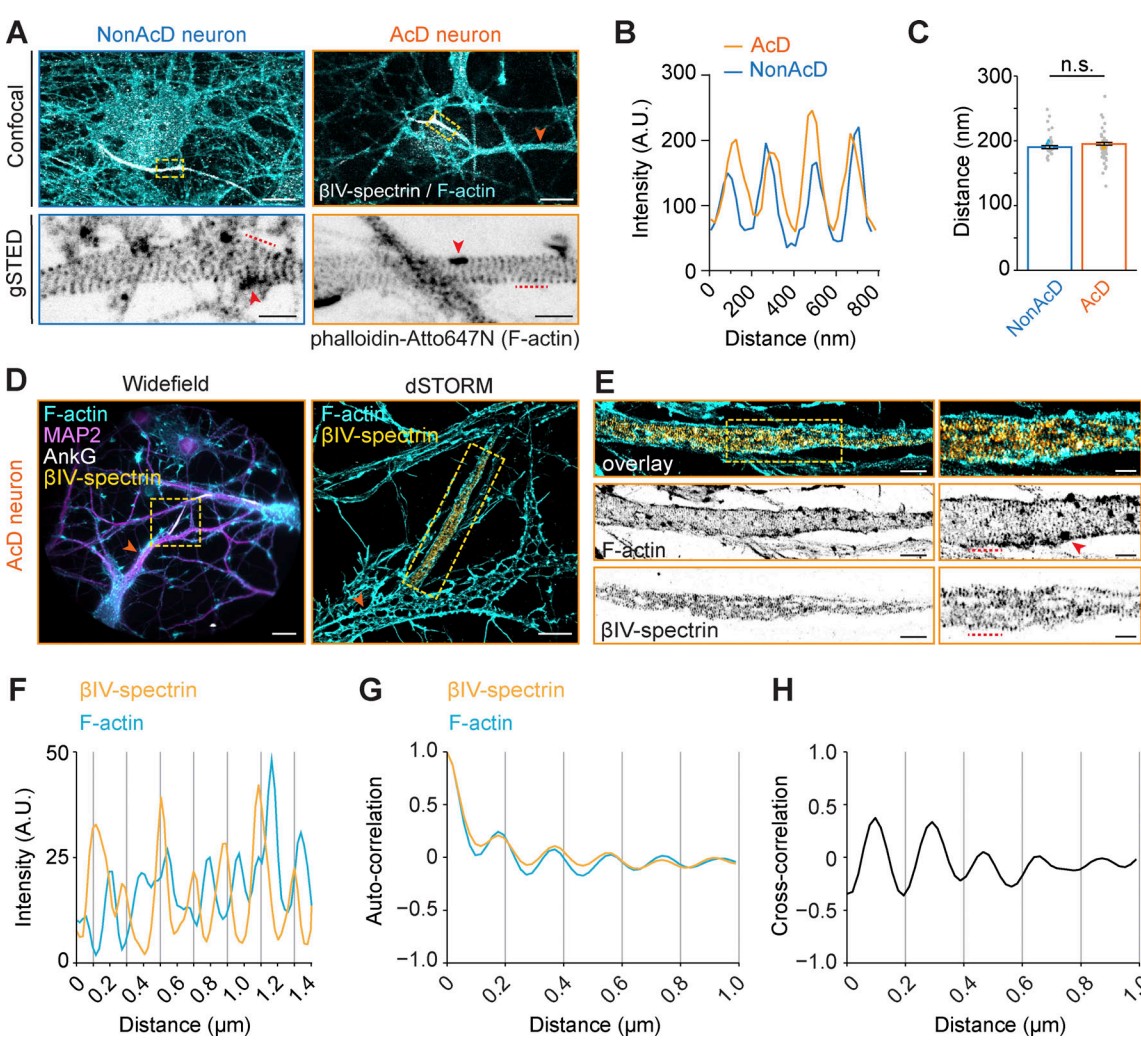

Figure 3. **AcD neurons have similar MPS organization of the AIS as in nonAcD neurons. (A)** Top row: Representative confocal images of DIV14 nonAcD (left) and AcD (right) neurons stained with the AIS marker βIV-spectrin and F-actin probe Phalloidin-Atto647N. Orange arrowhead indicates the stem dendrite of the displayed AcD neuron. Yellow dashed rectangle indicates the AIS. Scale bar is 10 µm. Bottom row: Single plane 2D gSTED image of F-actin in the AIS of nonAcD and AcD neurons (corresponding zoom-ins of yellow dashed rectangle in top row). Scale bar is 1 µm. Red arrowhead indicates F-actin patch. Red dashed line indicates the profile of F-actin rings shown in B. **(B)** Intensity profile of periodic F-actin structures along the longitudinal axis of the AIS in nonAcD and AcD neurons; corresponding to black dashed lines in A bottom row. **(C)** Quantification of distance between F-actin rings at the AIS of nonAcD and AcD neurons. Mean ± SEM, three independent cultures, nonAcD: n = 45 profiles from 12 cells, AcD: n = 67 profiles from 18 cells. Grey dot indicates value of individual cell. Orange and cyan triangle indicates mean of each independent culture. Mann–Whitney test (two-sided): not significant (n.s.) P > 0.05. More detailed statistical information see Data S1. **(D)** Representative widefield (left) and dSTORM (right) images of DIV14 AcD neurons stained with the AIS marker βIV-spectrin and AnkG, F-actin probe Phalloidin-Alexa647+ and MAP2. Scale bar is 20 µm on widefield image and 5 µm on dSTORM image. Orange arrowhead indicates the stem dendrite. Yellow dashed square indicates the AIS. dSTORM image corresponds to the yellow dashed rectangle in widefield image. **(E)** Dual-color 3D dSTORM image of F-actin and βIV-spectrin at the AIS of the displayed AcD neuron. Left panel: corresponding zoom-in of the yellow dashed square on dSTORM image shown in D right panel. Yellow dashed square indicates the zoom-in region shown in right panel. Right panel: corresponding zoom-in of the yellow dashed square in left panel. Scale bar is 5 µm. Red arrowhead indicates F-actin patch. Red dashed line indicates the intensity profile of F-actin rings and βIV-spectrin shown in F. **(F)** Intensity profile of periodical F-actin (cyan) and βIV-spectrin (orange) structures along the longitudinal axis of the AIS in AcD neurons; corresponding to red dashed lines in E right panel. **(G)** Auto-correlation curve of F-actin (cyan) and βIV-spectrin (orange) at the AIS of AcD neurons. Three independent cultures, AcD: n = 39 profiles from eight cells. Space between βIV-spectrin bands is ∼196 nm. Space between F-actin bands is ∼200 nm. **(H)** Cross-correlation curve of F-actin and βIV-spectrin at the AIS of AcD neurons. Three independent cultures, AcD: n = 39 profiles from eight cells.

and periodically arranged ring-like structures (Fig. 3 A). Similar to nonAcD neurons, the distance between F-actin rings in the AcD neuron's AIS was ∼190 nm (Fig. 3, B and C), suggesting that the F-actin in AcD neuron's AIS is similarly organized as in nonAcD neurons.

To get further details on the MPS structure, we then performed dual-color 3D stochastic optical reconstruction

microscopy (dSTORM) of βIV-spectrin and F-actin at the AIS of AcD neurons identified by MAP2 and AnkG staining patterns. We found that the βIV-spectrin also forms ring-like structures at the AIS of AcD neuron with a distance of ∼196 nm between each band and a complementary pattern between F-actin and βIV-spectrin bands (Fig. 3, D–F), in line with previous results in nonAcD neurons (Leterrier et al., 2015; Xu et al., 2013).

Autocorrelation and cross-correlation analysis confirmed that βIV-spectrin and actin bands are periodically arranged (Fig. 3 G) and that the βIV-spectrin bands alternate with actin bands (Fig. 3 H). Collectively, our data suggest that the dendritic axon origin in AcD neurons does not change MT and MPS nanostructure in the AIS.

## The AIS of AcD neurons regulates neuronal polarity by selective permissiveness of axonal cargoes

The AIS helps the axon to retain distinct protein composition by selectively blocking MT-based trafficking of dendritic cargoes (Al-Bassam et al., 2012; Arnold and Gallo, 2014; Balasanyan et al., 2017; Leterrier, 2018; Rasband, 2010; Watanabe et al., 2012). To test the filtering function of the AIS of AcD neurons, we selected well-established markers of dendritic and axonal cargoes. For dendritic cargoes, we live-labeled endogenous TfRs (Burack et al., 2000; Cameron et al., 1991) with the fluorescently conjugated ligand Transferrin-Alexa568. For axonal cargoes, we selected Rab3A which associates with presynaptic vesicles (Niwa et al., 2008). The AIS was live-labeled with an anti-NF-CF640R antibody (Fig. S1 I). Time-lapse imaging revealed that EGFP-Rab3A vesicles in AcD neurons were specifically directed into the axon (Fig. 4, A–C and Video 6 A). Conversely, the TfR vesicles in AcD neurons were mostly moving within the somatodendritic compartments but were halted at the AIS, and nearly no entry of TfR vesicles into the axon was observed (Fig. 4, D–F and Video 7 A).

To further assess the axonal cargo permissiveness and dendritic cargo filtering capacity of the AIS in AcD neurons in more detail, we thoroughly analyzed the trajectories of TfR and EGFP-Rab3A vesicles using a Python-based program—KYMOA. TfR-positive organelles in both AcD and nonAcD neurons were immobile in the AIS, but highly mobile in dendrites (Fig. 4, G and H; and Video 7, A and B). Although a few moving TfR vesicles were visible at the AIS (Fig. 4, G and H), they preferentially moved in the retrograde direction (Fig. 4 I) and showed a higher percentage and number of pausing (Fig. 4 J and Fig. S2 A), longer total pausing duration (Fig. S2 B), and slower velocity (Fig. S2 C) than in the dendritic region. Moreover, these mobile TfR vesicles within the AIS showed shorter run length (Fig. S2D) and run duration in the anterograde direction (Fig. S2 E) compared with the dendrites, indicating that, like the AIS of nonAcD neurons, the AIS of AcD neurons selectively blocks the entry of dendritic cargo into the axonal region.

In contrast to the TfR, quantitative analysis showed that most EGFP-Rab3A vesicles were mobile within the AIS of both AcD and nonAcD neurons (Fig. 4 K; and Video 6, A and B), and no significant difference was found on the average number of mobile vesicles between the AIS of both cell types (Fig. 4 L; and Video 6, A and B). These mobile EGFP-Rab3A vesicles mainly traveled anterogradely across the AIS toward the distal axon (Fig. 4 M) and spent nearly 70% of the total time in processive runs (Fig. 4 N). Only a small fraction of EGFP-Rab3A vesicles underwent directional reversal while traveling anterogradely through the AIS (Fig. S2 F). During the rest of the total time, the anterogradely transported EGFP-Rab3A vesicles in the AIS of both AcD and nonAcD neurons were either passively diffusing

back and forth (Fig. 4 N) or shortly pausing (Fig. 4 N and Fig. S2, G–I). Interestingly, we found that the anterogradely transported EGFP-Rab3A vesicles in the AIS of AcD neurons show slightly longer mean run lengths (Fig. 4 O), similar mean run duration (Fig. 4 P) and, therefore, higher mean run velocity than those in the AIS of nonAcD neurons (Fig. 4 Q). This finding suggests that the Rab3A-positive vesicles are transported slightly faster across the AIS of AcD neurons than nonAcD neurons.

Of note, we did not observe any difference in run length, run duration, or velocity on retrogradely transported EGFP-Rab3A vesicles between the AIS of AcD and nonAcD neurons (Fig. 4, O–Q). As expected, EGFP-Rab3A vesicles in regular dendrites showed very limited mobility (Fig. 4, A–C and K–Q; and Fig. S2, F–I). Intriguingly, we noticed that both the dendritic cargo TfR and the axonal cargo EGFP-Rab3A were actively transported in the stem dendrite of AcD neurons (Fig. 4, A–F; and Videos 6 and 7). The velocity of EGFP-Rab3A and TfR vesicles in the stem dendrite highly resembled that in the AIS and dendrites (Fig. S2 C and Fig. 4 Q), respectively. However, the average run length and run duration for Rab3A vesicles traveling in the anterograde direction (Fig. 4, O and P) was much shorter at the stem dendrite than in the AIS, and the run length of TfR vesicles traveling in the stem dendrite toward anterograde direction was also reduced compared with dendrites (Fig. S2 D). This reduction of run length and duration is likely due to the difference in physical length between these compartments since the stem dendrite is in general shorter than the AIS and the dendrites.

Is the faster anterograde trafficking speed in the AIS of AcD neurons specific to Rab3A-associated presynaptic vesicles, or is it a general feature of all cargoes that travel into the axon? We addressed this question by looking at neuropeptide Y (NPY) as an additional axon-specific cargo as well as lysosomal-associated membrane protein 1 (LAMP-1)-labeled endolysosomes that are indiscriminately transported into both the axon and dendrites (Schlager et al., 2010; van Bommel et al., 2019). We found that both NPY-GFP (Fig. S3, A–C; and Video 8, A and B) and LAMP1-mCherry (Fig. S3, D–F; and Video 9, A and B) positive organelles entered the axon and successfully passed through the AIS. Moreover, the velocity of anterograde and retrograde transport for both LAMP1 and NPY was the same between the AIS of AcD and nonAcD neurons (Fig. S3, G and H). We also measured run length, run duration, number of pausing, and total pausing time for both NPY- (Fig. S3, I–L) and LAMP1 (Fig. S3, M–P)-positive vesicles and found no significant difference, except that LAMP1 showed slightly shorter run length when traveling in the retrograde direction (Fig. S3 M). This data suggests that the faster anterograde trafficking velocity in the AcD neuron's AIS might be specific to presynaptic vesicles associated with Rab3A.

Taken together, our results suggest that the AIS of AcD neurons fully preserves its filtering capacity, and the anterograde transport of axonal cargoes is equally effective and processive as in the AIS of nonAcD neurons.

## The AIS of AcD neurons contains fewer cisternal organelles and inhibitory synapses than the AIS of nonAcD neurons

Next, we focused on mapping the AIS-localized proteins that are critical for neuronal excitability and thus underly the AIS

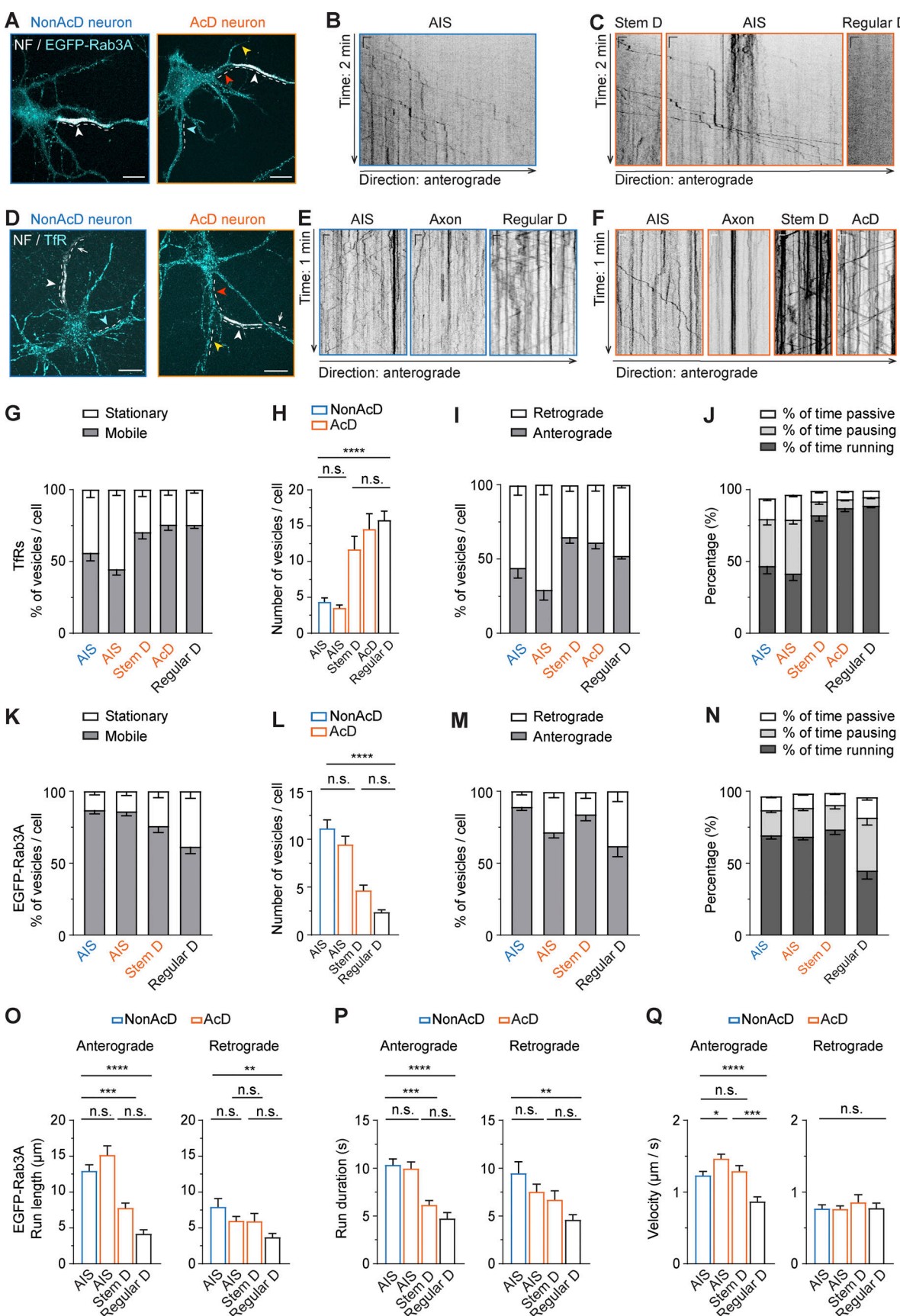

Figure 4. **The AIS of AcD neurons serves as an efficient barrier, preventing entry of dendritic cargo into the axon. (A)** Representative images of nonAcD (left) and AcD (right) neurons expressing a presynaptic vesicle marker EGFP-Rab3A, the AIS is live-labeled with anti-NF-CF640R (NF). White dashed line

indicates the analyzed area; white arrowhead indicates the AIS; red, yellow, and cyan arrowheads indicate the stem dendrite, AcD, and regular dendrite of AcD neuron, respectively. Scale bar is 15 µm. Related to Video 6. **(B)** Representative kymographs showing trajectories of EGFP-Rab3A vesicles entering the AIS of nonAcD neuron shown in A, corresponding to area indicated by white dashed line. Scale is 10 s (vertical) and 2 µm (horizontal). **(C)** Representative kymographs showing trajectories of EGFP-Rab3A vesicles moving in the stem dendrite (Stem D), AIS and regular dendrite (Regular D) of the AcD neuron shown in A, corresponding to area indicated by white dashed line. Scale is 10 s (vertical) and 2 µm (horizontal). **(D)** Representative images of nonAcD (left) and AcD (right) neurons labelled with anti-NF-CF640R (NF) for AIS and Transferrin-Alexa568 for TfRs as dendritic cargo. White dashed line indicates analyzed area; white arrowhead indicates the AIS; cyan arrowhead indicates regular dendrite of nonAcD neuron; yellow and red arrowhead indicates the AcD and stem dendrite of AcD neuron, respectively. Scale bar is 15 µm. Relate to Video 7. **(E)** Representative kymographs showing trajectories of TfR vesicles entering and moving in the AIS, axon and regular dendrite (Regular D) of nonAcD neuron shown in D, corresponding to the area indicated by white dashed line. Scale is 5 s (vertical) and 2 µm (horizontal). **(F)** Representative kymographs showing trajectories of TfR vesicles moving in the AIS, axon, stem dendrite (Stem D) and AcD of the AcD neuron shown in D, corresponding to the area indicated by white dashed line. Scale is 5 s (vertical) and 2 µm (horizontal). **(G and H)** Motility of TfR vesicles in the AIS of nonAcD neuron, the AIS, stem dendrite (Stem D) and AcD of AcD neuron, and the regular dendrite (Regular D) of both nonAcD and AcD neurons. **(G)** Percentage of mobile and stationary TfR vesicles. **(H)** Number of mobile TfR vesicles per cell. Mean ± SEM, five independent cultures, AIS (nonAcD) $n = 25$ cells, AIS (AcD) $n = 23$ cells, Stem D $n = 23$ cells, AcD $n = 19$ cells, Regular D $n = 56$ cells. **(I)** Percentage of TfR vesicles running towards anterograde and retrograde directions in the AIS of nonAcD neuron, the AIS, stem dendrite (Stem D) and AcD of AcD neuron, and the regular dendrite (Regular D) of both nonAcD and AcD neurons. Mean ± SEM, five independent cultures, AIS (nonAcD) $n = 25$ cells, AIS (AcD) $n = 23$ cells, Stem D $n = 23$ cells, AcD $n = 19$ cells, Regular D $n = 56$ cells. **(J)** Percentage of time a mobile TfR vesicle is running, pausing or passively moving during anterograde transport. Mean ± SEM, five independent cultures, AIS (nonAcD) $n = 25$ cells, AIS (AcD) $n = 23$ cells, Stem D $n = 23$ cells, AcD $n = 19$ cells, Regular D $n = 56$ cells. One-way ANOVA with Tukey's multiple comparisons test, no significance (n.s.) $P > 0.05$, **$P < 0.01$, ***$P < 0.001$, ****$P < 0.0001$. **(K and L)** Motility of EGFP-Rab3A vesicles in the AIS of nonAcD neuron, the AIS and stem dendrite (Stem D) of AcD neuron, and the regular dendrite (Regular D) of both nonAcD and AcD neurons. **(K)** Percentage of mobile and stationary EGFP-Rab3A vesicles. **(L)** Number of mobile EGFP-Rab3A vesicles per cell. Mean ± SEM, seven independent cultures, AIS (nonAcD): $n = 42$ cells, AIS (AcD) $n = 42$ cells, Stem D: $n = 28$ cells, Regular D: $n = 27$ cells. **(M)** Percentage of EGFP-Rab3A vesicles running towards anterograde and retrograde directions in the AIS of nonAcD neuron, the AIS and stem dendrite (Stem D) of AcD neuron, and the regular dendrite (Regular D) of both nonAcD and AcD neurons. Mean ± SEM, seven independent cultures, AIS (nonAcD): $n = 42$ cells, AIS (AcD) $n = 42$ cells, Stem D: $n = 28$ cells, Regular D: $n = 27$ cells. **(N)** Percentage of time a mobile Rab3A vesicle that is running, pausing or passively moving during anterograde transport. Mean ± SEM, seven independent cultures, AIS (nonAcD) $n = 42$ cells, AIS (AcD) $n = 42$ cells, Stem D $n = 28$ cells, Regular D $n = 27$ cells. **(O–Q)** Average length (O), duration (P) and velocity (Q) of EGFP-Rab3A vesicles running towards anterograde and retrograde direction within the AIS of nonAcD neuron, the AIS and stem dendrite (Stem D) of AcD neuron, and the regular dendrite (Regular D) of both nonAcD and AcD neurons. Mean ± SEM, seven independent cultures, Anterograde: AIS (nonAcD) $n = 42$ cells, AIS (AcD) $n = 41$ cells, Stem D $n = 28$ cells, Regular D $n = 26$ cells, Retrograde: AIS (nonAcD) $n = 25$ cells, AIS (AcD) $n = 37$ cells, Stem D $n = 17$ cells, Regular D $n = 20$ cells. More detailed statistical information see Data S1.

function in neurotransmission. We first characterized the distribution of voltage-gated sodium channels (VGSCs) that are required for APs initiation at the AIS (Leterrier, 2018; Rasband, 2010). Our data showed that VGSCs are enriched at the AIS of AcD neurons (Fig. 5 A), and the labeling intensity was slightly lower than in the AIS of nonAcD neurons (Fig. 5 E). We then measured the distribution of AnkG and neurofascin, which anchor VGSCs at the AIS (Hedstrom et al., 2007; Leterrier, 2018; Leterrier et al., 2015; Rasband, 2010) and found no significant difference in these proteins between the AIS of the two neuronal cell types (Fig. 5, B, C, F, and G). These results indicate that the amounts of sodium channels, AnkG, and neurofascin are similar between AcD and nonAcD neurons. We also analyzed the endogenous expression of the AIS-specific ECM protein brevican (Hedstrom et al., 2007) in DIV21 AcD neurons (Fig. S4 A). Brevican signal in AcD neurons showed the same localization as in nonAcD neurons and was exclusively present at the beginning of the axon (Fig. S4, A and B), suggesting that a specific ECM is formed around the AIS of AcD neurons.

Another unique feature of the AIS is the presence of cisternal organelles (Benedeczky et al., 1994; Jungenitz et al., 2023; Konietzny et al., 2019; Bas Orth et al., 2007; Sánchez-Ponce et al., 2011). By using synaptopodin (synpo) as a marker, we next measured the density and size of cisternal organelles at the AIS of AcD neurons. In comparison to nonAcD neurons, the density was reduced at the AIS of AcD neurons (Fig. 5, D and H), but their size remained the same (Fig. 5 H). This alteration in cisternal organelle density could potentially impact the dynamic of cytoplasmic Ca$^{2+}$ transients.

Excitatory neurons are known to establish inhibitory axo-axonic synapses at the AIS for the regulation of excitability via innervation of GABAergic neurons (Nathanson et al., 2019). To estimate the GABAergic neuron population in primary hippocampal cultures, we quantified the percentage of inhibitory neurons in at least three independent preparations by using GAD1 as an inhibitory neuron marker and pCaMKII as an excitatory neuron marker (Fig. 6 A). Quantification showed that on average 10% GAD1 positive and 15% pCaMKII negative neurons are present in our cultures system, with a low variation of inhibitory neuron proportion between cultures (Fig. 6 B). We next set out to quantify the amount of inhibitory axo-axonic synapses at the AIS of AcD neurons. We analyzed the density of the inhibitory pre- and postsynaptic markers, vesicular GABA transporter (VGAT), and gephyrin, respectively. We found that the density of both markers at the AIS of AcD neurons was significantly reduced compared with that in nonAcD neurons (Fig. 6, C–E). Co-labeling of gephyrin and VGAT to identify bona fide inhibitory synapses at the AIS also revealed a reduced number of co-localizations in AcD neurons (Fig. 6 F and Video 10) but no difference in the distance between co-localized gephyrin and VGAT (Fig. S4 E). These data suggest that there is a significant difference in the density of inhibitory axo-axonic synapses between the AIS of AcD and nonAcD neurons.

As a control, we also measured the density of excitatory synapses at the AIS of AcD and nonAcD neurons by using vesicular glutamate transporter (VGLUT) and homer-1 as pre- and postsynaptic markers, respectively. Our quantification showed a very low density of both homer-1 and VGLUT at the

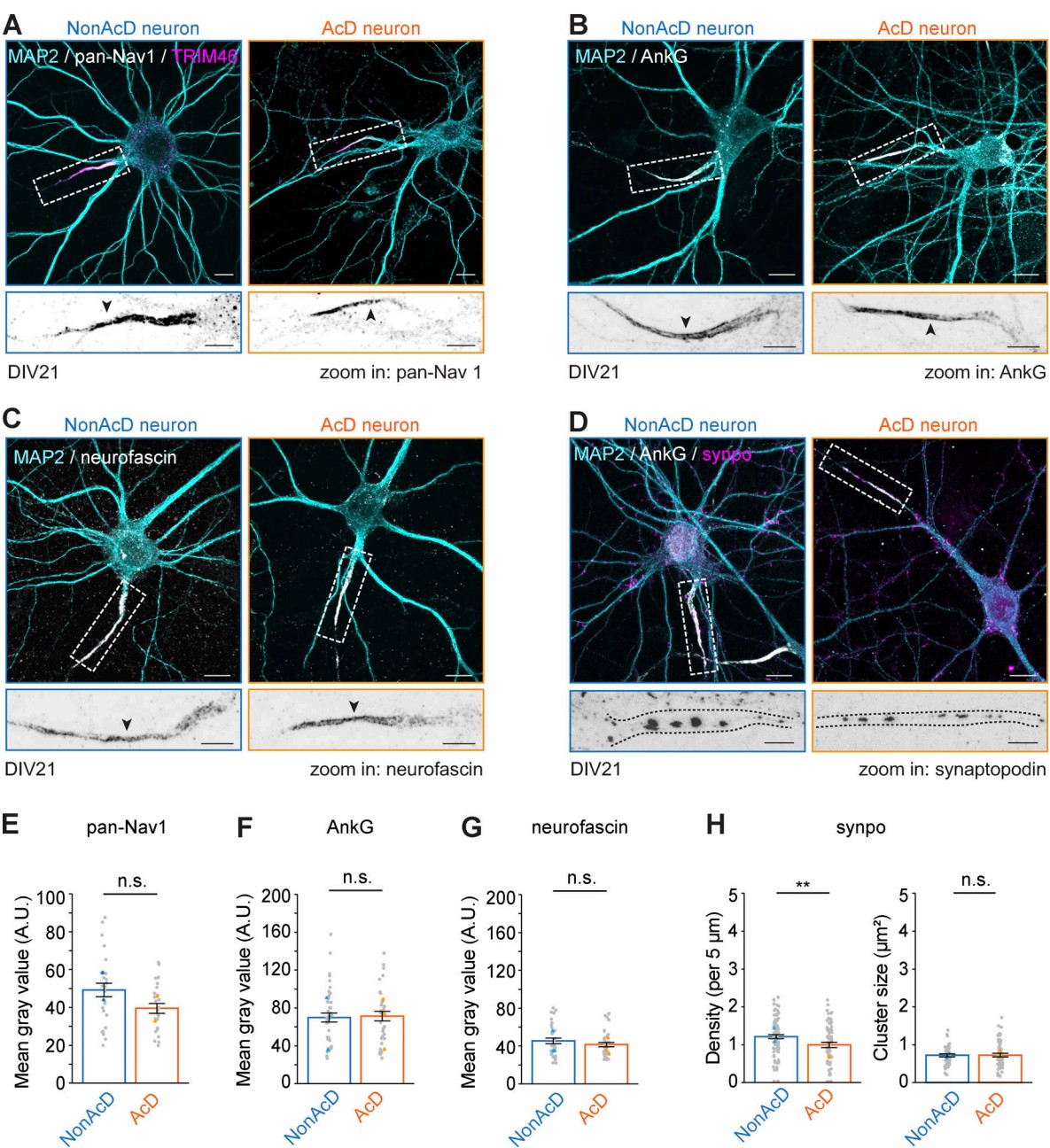

Figure 5. **AcD neurons do not differ in pan-Nav1, AnkG, and neurofascin abundancies at the AIS from nonAcD neurons but contain fewer cisternal organelles. (A)** Top row: Representative images of DIV21 nonAcD (left) and AcD (right) neuron-labelled with MAP2, AIS-specific MAP TRIM46, and pan-Nav1 for sodium channels. White dashed rectangle indicates the AIS. Scale bar is 10 μm. Bottom row: corresponding zoom-ins to white dashed rectangle in upper row. Scale bar is 5 μm. **(B)** Top row: Representative images of DIV21 nonAcD (left) and AcD (right) neuron labelled with MAP2 and AIS specific scaffolding protein AnkG. White dashed rectangle indicates the AIS. Scale bar is 10 μm. bottom row: corresponding zoom-ins to white dashed rectangle in Upper row. Scale bar is 5 μm. **(C)** Top row: Representative images of DIV21 nonAcD (left) and AcD (right) neuron labelled with MAP2 and AIS specific membrane protein neurofascin. White dashed rectangle indicates the AIS. Scale bar is 10 μm. Bottom row: corresponding zoom-ins to white dashed rectangle in Upper row. Scale bar is 5 μm. **(D)** Top row: Representative images of DIV21 nonAcD (left) and AcD (right) neuron labeled with MAP2, AIS-specific scaffolding protein AnkG, and synaptopodin (synpo) for cisternal organelles. White dashed rectangle indicates the AIS. Scale bar is 10 μm. Bottom row: corresponding zoom-ins to white dashed rectangle in upper row. Scale bar is 5 μm. **(E–G)** Quantification of fluorescent intensity at the AIS of nonAcD and AcD neurons for sodium channels (E), AnkG (F), and neurofascin (G). Mean ± SEM, three independent cultures for each AIS protein, pan-Nav1: $n$ (nonAcD) = 15 cells, $n$ (AcD) = 19 cells, AnkG: $n$ (nonAcD) = 41 cells, $n$ (AcD) = 38 cells, neurofascin: $n$ (nonAcD) = 32 cells, $n$ (AcD) = 36 cells. Grey dot indicates value of individual cell. Orange and cyan triangle indicate mean of each independent culture. **(H)** Quantification of synpo cluster density (left) and size (right) at the AIS of nonAcD and AcD neurons. Mean ± SEM, three independent cultures, synpo density: $n$ (nonAcD) = 91 cells, $n$ (AcD) = 67 cells, cluster size: $n$ (nonAcD) = 48 cells, $n$ (AcD) = 54 cells. Grey dot indicates the value of individual cell. Orange and cyan triangles indicate mean of each independent culture. Mann–Whitney test (two-sided): not significant (n.s.) P > 0.05, *P < 0.05, **P < 0.01. More detailed statistical information see Data S1.

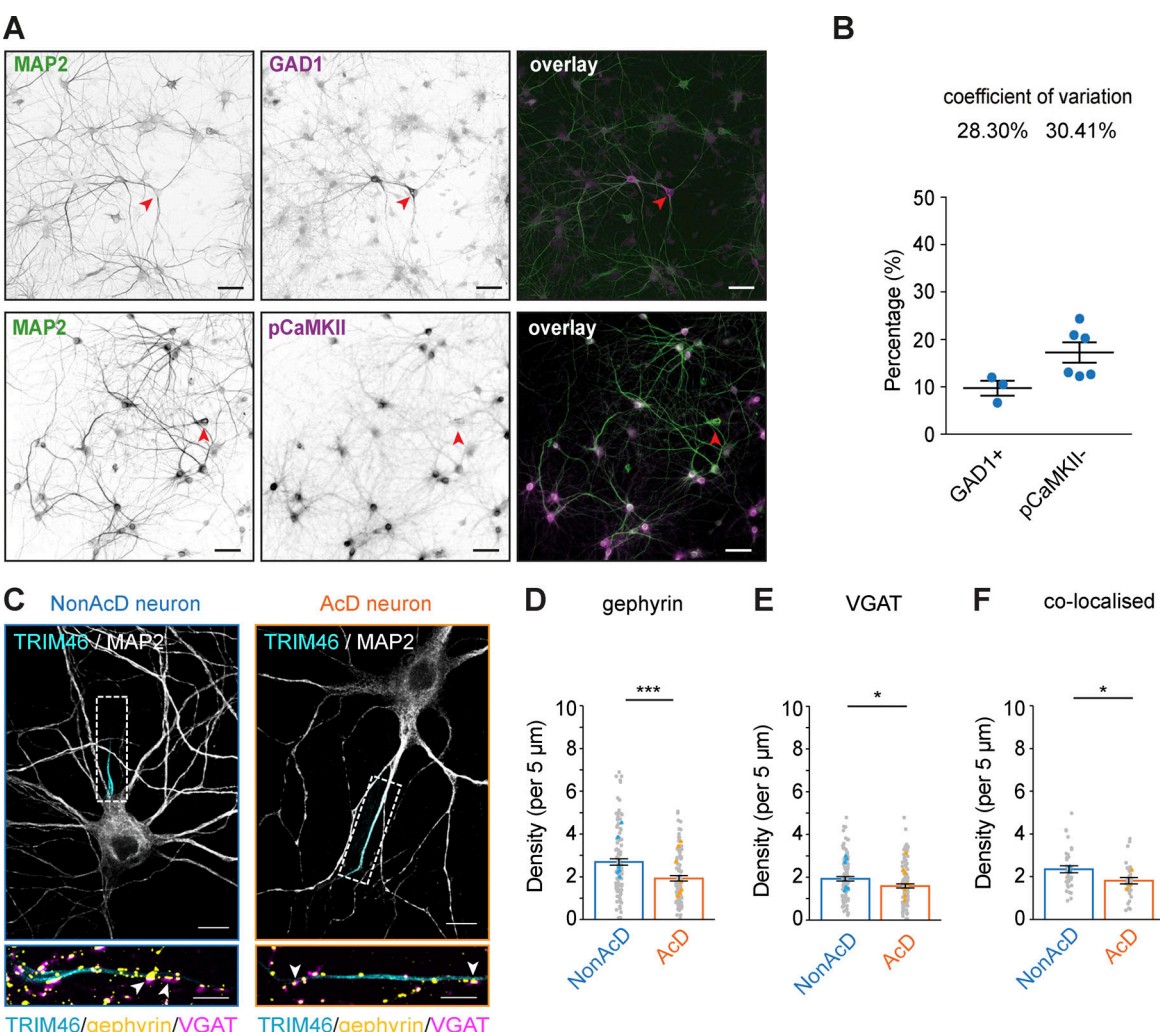

**Figure 6. AcD neurons have less inhibitory synapses at the AIS. (A)** Representative images of DIV14 hippocampal neurons stained with GAD1 for inhibitory interneurons (top row), and pCaMKII (bottom row) for excitatory neurons. Neuronal morphology is highlighted by MAP2. Red arrowhead indicates GAD1 positive neurons (top row) and pCaMKII negative neurons (bottom row). Scale bar is 50 µm. **(B)** Quantification of GAD1 positive (GAD1+) and pCaMKII negative (pCaMKII−) neuron percentage in DIV14 dissociated hippocampal cultures. Mean ± SEM, GAD1: $n$ = 3 independent preparations, pCaMKII: $n$ = 6 independent preparations. **(C)** Top row: Representative images of nonAcD (left) and AcD (right) neurons co-immunostained with markers for pre- and postsynaptic compartments of an inhibitory synapse (VGAT and gephyrin, respectively), for AIS (TRIM46), and for somato-dendritic compartment (MAP2). Scale bar is 10 µm. White dashed rectangle indicates the AIS. Bottom row: corresponding zoom-ins of white dashed rectangular in upper row. Scale bar is 5 µm. White arrowheads indicate co-localized gephyrin and VGAT clusters. Related to Video 10. **(D)** Number of single gephyrin clusters per 5 µm at the AIS of nonAcD and AcD neurons. Mean ± SEM, seven independent cultures, nonAcD: $n$ = 111 cells, AcD: $n$ = 97 cells. Grey dot indicates the value of individual cell. Orange and cyan triangle indicate mean of each independent culture. **(E)** Number of single VGAT clusters per 5 µm at the AIS of nonAcD and AcD neurons. Mean ± SEM, seven independent cultures, nonAcD: $n$ = 102 cells, AcD: $n$ = 104 cells. Grey dot indicates value of individual cell. Orange and cyan triangle indicate mean of each independent culture. **(F)** Number of VGAT clusters co-localize with gephyrin per 5 µm at the AIS of nonAcD and AcD neurons. Mean ± SEM, three independent cultures, nonAcD: $n$ = 34 cells, AcD: $n$ = 32 cells. Grey dot indicates value of individual cell. Orange and cyan triangle indicate mean of each independent culture. Mann–Whitney test: not significant (two-sided): not significant (n.s.) P > 0.05, *P < 0.05, ***P < 0.0001. More detailed statistical information see Data S1.

AIS regardless of axon origin (Fig. S4, C, D, and F), with no difference in this small excitatory synapse number between neuron types.

**The AIS of AcD neurons does not undergo activity-dependent plasticity**
The AIS regulates neuronal homeostasis at different time scales via short-term and chronic plasticity (Evans et al., 2013, 2015, 2017; Grubb and Burrone, 2010; Jahan et al., 2023; Kuba et al., 2006; Susuki and Kuba, 2016; Wefelmeyer et al., 2015; Yamada

and Kuba, 2016). Previous studies have shown that elevating extracellular potassium ($K^+$) concentration in dissociated hippocampal cultures to depolarize the neuronal membrane induces short-term AIS plasticity after 3 h, resulting in a shortening of the AIS (Evans et al., 2015). Chronic AIS plasticity occurs at a later time point (after 48 h), causing a shift of the entire AIS toward the distal axon (Grubb and Burrone, 2010). Both changes in AIS length and position were accompanied by a reduced AP firing rate and have so far only been described in nonAcD neurons (Evans et al., 2013, 2015; Grubb and Burrone,

2010; Wefelmeyer et al., 2015). This motivated us to investigate how homeostasis is regulated in AcD neurons.

To induce short-term and chronic AIS plasticity, we elevated extracellular K+ concentrations in DIV12 dissociated hippocampal neurons to 15 mM by adding extra KCl for 3 and 48 h, respectively (Fig. S5 A). To rule out possible effects caused by the change in osmolality, we replaced KCl with NaCl as a control group (Fig. S5 A). As an additional control, we also silenced the baseline synaptic activity of dissociated neurons with 1 μM tetrodotoxin (TTX) (Fig. S5 A). To read out AIS plasticity, we labeled the AIS with AnkG and measured its intensity profile to define the AIS length and AIS distance (Fig. S5, B and C). Using a previously published method (Evans et al., 2013, 2015; Grubb and Burrone, 2010), we considered the start and the end of the AIS as the location where AnkG fluorescence intensity drops below 40% of the maximum intensity along the profile (Fig. S5, B and C; see Materials and methods for details). The AIS distance is defined as the distance from the start of the axon to the start of the AIS (Fig. S5, B and C).

Notably, in nonAcD neurons, we did not observe the previously reported AIS length reduction after 3 h of KCl treatment (Fig. 7, A and B; and Fig. S5 D), but we indeed found a distal shift of the AIS after 48 h of KCl treatment as the AIS distance increased (Fig. 7, A and B). This finding suggests that the KCl treatment successfully triggered chronic AIS plasticity. Interestingly, the AIS length was not reduced after 3 h of KCl treatment (Fig. 7, C and D), nor did the 48-h KCl treatment shifted the AIS toward the distal axon tip in AcD neurons (Fig. 7, C and D). This indicates that the AIS in AcD neurons does not respond with the same adaptation to a prolonged membrane depolarization as seen in nonAcD neurons.

Both NaCl and TTX treatment showed no effect on the AIS length after 3 h (Fig. 7, B and D) or on the AIS position after 48 h in both AcD and nonAcD neurons (Fig. 7, B and D). Remarkably, we noticed that after 48 h of incubation, TTX specifically increased AIS length in nonAcD neurons (Fig. S5 E) but not in AcD neurons (Fig. S5 F). This finding not only provides evidence of AIS structural remodeling when nonAcD neurons are deprived of activity but further suggests that the AIS of AcD neurons is insensitive to activity changes.

AcD morphology has previously been noted to be more prevalent among GABAergic interneurons (Gonda et al., 2023; Höfflin et al., 2017; Wahle et al., 2022), and interneurons were shown to not undergo activity-dependent AIS plasticity (Chand et al., 2015; Muir and Kittler, 2014). To further validate our results on chronic AIS plasticity in AcD neurons and disentangle the contribution of the inhibitory AcD neuron population, we repeated the chronic stimulation experiment and combined it with the labeling of GABAergic interneurons by GAD1 immunostaining. Quantification showed no significant difference in the AIS distance in GAD1-negative AcD neurons between the KCl and control group (Fig. S5 G). This result confirms that excitatory AcD neurons indeed do not undergo the activity-dependent AIS plasticity.

What are the potential reasons for the lack of plasticity at the AIS of AcD neurons? A previous study (Thome et al., 2014) hinted that the stem dendrite might be involved in homeostatic regulation of AcD neurons. We therefore measured the length of the stem dendrite in AcD neurons after 48 h of incubation with KCl. However, the stem dendrite length is similar between the KCl-treated and control groups (Fig. S5 H), indicating that the stem dendrite of AcD neurons may not compensate for changes in neuronal activity. We also performed a two-way ANOVA analysis to test whether neuronal morphology is linked to the probability of neurons undergoing chronic AIS plasticity (Fig. S5 I). Interestingly, our analysis showed that the AIS is significantly shifted away only in nonAcD neurons treated with KCl but remained similar between other groups (Fig. S5 I), suggesting that AcD morphology has an impact on the occurrence of chronic AIS plasticity.

## Fewer cisternal organelles are found in the AIS after induction of chronic AIS plasticity

We next explored possible molecular mechanisms that regulate AIS plasticity. At first, we reasoned that the AIS-specific ECM could crosslink AnkG with membrane components (Hedstrom et al., 2007) and therefore restrict the ability of the AIS to change its size and position in response to neuronal activity. To test this, we investigated the expression of the AIS-specific ECM protein brevican in dissociated hippocampal neurons at DIV12, the age when AIS plasticity was induced in our study. However, we found almost no brevican signal (Fig. S4 G), suggesting that the ECM is unlikely to be involved at this time point.

The AIS plasticity in excitatory neurons is triggered by Ca2+ signaling (Bender and Trussell, 2009; Yu et al., 2010). A previous study already hinted the potential role of cisternal organelles in rapid AIS plasticity (Jungenitz et al., 2023). We, therefore, wondered if cisternal organelles are also changed upon induction of chronic AIS plasticity. To test this, we assessed the density, size, and distribution of cisternal organelles in the AIS of nonAcD neurons treated with KCl for 48 h. We analyzed untreated neurons at the same age at which chronic AIS plasticity was induced (DIV12 and DIV14) as a control. We found that cisternal organelles were already formed at the AIS of DIV12 neurons (Fig. 8 A). Quantification showed that both the amount and the size of synaptopodin clusters are the same between DIV12 and DIV14 neurons (Fig. 8, A–C), suggesting there are no developmental changes of the cisternal organelles during the time of plasticity induction.

Next, we performed the same analysis of cisternal organelles in neurons treated with KCl for 48 h to induce chronic AIS plasticity (Fig. 8, D and E) and compared them with the control group. We found that both the density and number of synpo clusters at the AIS of KCl-treated neurons are reduced (Fig. 8 F). We also compared the size of synpo clusters and their distribution within the AIS but found no difference in both parameters (Fig. 8, G and H). Collectively, our data suggest that a chronic and sustained increase in neuronal activity indeed reduces the amount of cisternal organelles at the AIS. This result resembles the situation in AcD neurons, which have lesser cisternal organelles (Fig. 5) at the AIS and possibly higher activity rate (Hodapp et al., 2022; Thome et al., 2014).

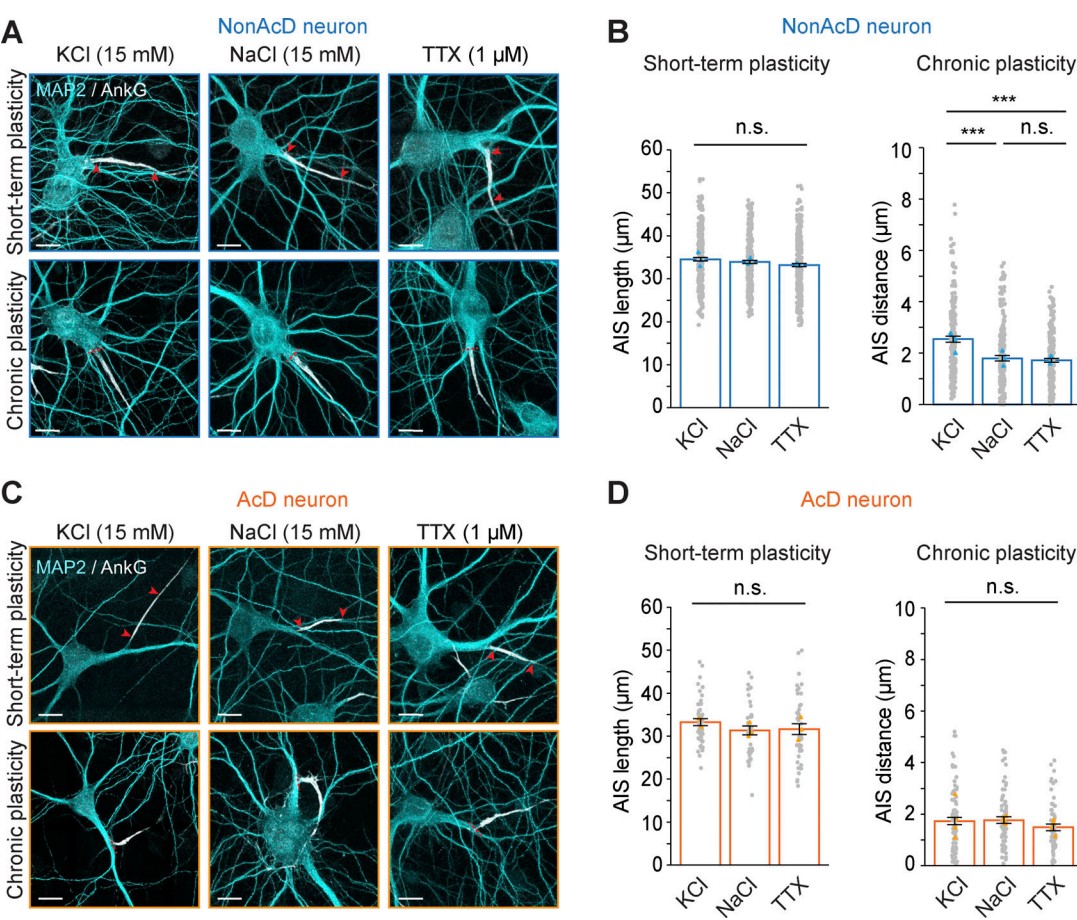

**Figure 7. Short-term and chronic plasticity of the AIS in AcD neurons. (A)** Representative images of the AIS in nonAcD neurons upon induction of short-term (top row) and chronic (bottom row) plasticity. Neurons were treated with KCl for depolarization, NaCl as control and TTX for silencing of activity. Red arrowheads in upper row indicate the start and end of AIS. Red dashed line in bottom row indicates the start of the axon. Scale bar is 10 μm. **(B)** Measurement of AIS length and AIS distance in nonAcD neurons treated with KCl (depolarization), NaCl (control) and TTX (silencing) for 3 and 48 h. Mean ± SEM, three independent cultures, short-term plasticity: *n* (KCl) = 298 cells, *n* (NaCl) = 264 cells, *n* (TTX) = 306 cells, Chronic plasticity: *n* (KCl) = 174 cells, *n* (NaCl) = 184 cells, *n* (TTX) = 192 cells. Grey dot indicates value of individual cell. Orange and cyan triangle indicate mean of each independent culture. **(C)** Representative images of the AIS in AcD neurons upon induction of short-term (top row) and chronic (bottom row) plasticity. Neurons were treated with KCl for depolarization, NaCl as control, and TTX for silencing. Red arrowheads in upper row indicate the start and end of AIS. Red dashed line in bottom row indicates the start of the axon. Scale bar is 10 μm. **(D)** Measurement of AIS length and AIS distance in AcD neurons treated with KCl (depolarization), NaCl (control) and TTX (silencing) for 3 and 48 h. Mean ± SEM, Short-term plasticity: three independent cultures, *n* (KCl) = 48 cells, *n* (NaCl) = 40 cells, *n* (TTX) = 41 cells, Chronic plasticity: four independent cultures, *n* (KCl) = 82 cells, *n* (NaCl) = 72 cells, *n* (TTX) = 63 cells. Grey dot indicates value of individual cell. Orange and cyan triangle indicate mean of each independent culture. One-way ANOVA with Tukey's multiple comparisons test, no significance (n.s.) $P > 0.05$, ***$P < 0.001$. More detailed statistical information see Data S1.

## Discussion

We showed that the development into AcD morphology can occur independently from the in vivo environment, and, in most cases, a single precursor neurite gives rise to both the axon and the AcD. Although multiple strategies exist for axon and AcD differentiation, our data clearly indicated that AcD neurons still follow the classical developmental sequence (Dotti et al., 1988) of first establishing the axon and then the dendrites. The AcD then appears mainly from a collateral formed at the proximal region of an immature axon during early development. We found that the AIS of AcD neurons is positive for classical AIS markers, associates with the AIS-specific ECM, and has similar MPS and MT organization as in nonAcD neurons. Live imaging experiments demonstrated that the AIS of AcD neurons likewise acts as an efficient filter to selectively block the entry of dendritic cargo

into the axon. Interestingly, our data indicated that, in vitro, AcD neurons have fewer inhibitory synapses and cisternal organelles at the AIS compared with nonAcD neurons. These differences in the structural characteristics of the AIS suggest that the functionality of the AIS in regulating the homeostasis of AcD neurons might differ from nonAcD neurons. Indeed, we found that the AIS of excitatory AcD neurons does not undergo homeostatic plasticity upon chronic depolarization that mimics situations of high neuronal activity.

Our quantification showed a portion of ∼20% of AcD neurons in dissociated hippocampal cultures (Fig. S1 E) which is lower than the ∼50% reported previously (Thome et al., 2014). This difference could be explained by several reasons. First, the 50% of AcD neuron population is specific to the hippocampal CA1 region, while in other hippocampal regions, the AcD neuron

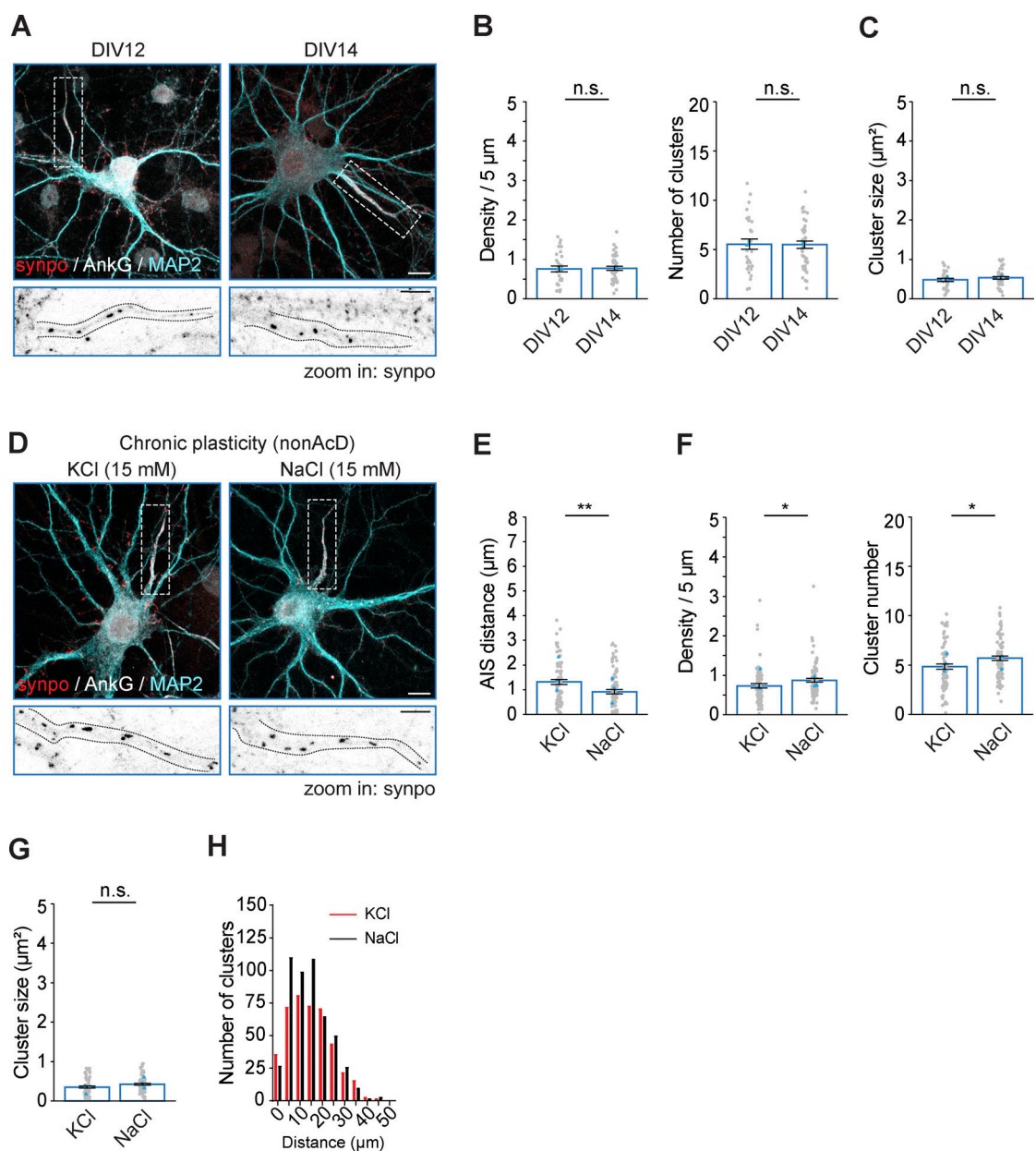

Figure 8. **Increased neuronal activity reduces the number of cisternal organelles in the AIS. (A)** Representative images of DIV12 and DIV14 neurons labelled with synpo antibodies for cisternal organelle, the AIS (AnkG) and the somato-dendritic compartment (MAP2). Scale bar is 10 µm. Inset: Zoom-ins corresponding to white dashed rectangular. Black dashed lines indicate the edge of AIS. Scale bar is 10 µm. **(B)** Quantification of synpo cluster density (left) and number (right) at the AIS of DIV12 and DIV14 neurons. Mean ± SEM, two independent cultures, cluster density: n (DIV12) = 29 cells, n (DIV14) n = 41 cells, cluster number: n (DIV12) = 31 cells, n (DIV14) = 42 cells. Grey dot indicates number of individual cells, cyan triangle indicates mean of each experiment. **(C)** Quantification of synpo cluster size at the AIS of DIV12 and DIV14 neurons. Mean ± SEM, two independent cultures, DIV12: n = 32 cells, DIV14: n = 40 cells. Grey dot indicates number of individual images, cyan triangle indicates mean of each experiment. **(D)** Representative images of DIV14 neurons labelled with synpo, AnkG and MAP2. Neurons were treated with KCl and NaCl from DIV12 for 48 h to induce chronic plasticity. Scale bar is 10 µm. Inset: Zoom-ins corresponding to white dashed rectangular. Black dashed lines indicate the edge of AIS. Scale bar is 10 µm. **(E)** AIS distance from the soma in nonAcD neurons upon induction of chronic plasticity. Mean ± SEM, three independent cultures, n (KCl) = 73 cells, n (NaCl) = 79 cells. Grey dot indicates number of individual cells, cyan triangle indicates mean of each experiment. **(F)** Quantification of synpo cluster density (left) and number (right) at the AIS of DIV14 neurons upon induction of chronic AIS plasticity. Mean ± SEM, three independent cultures, cluster density: n (KCl) = 73 cells, n (NaCl) = 79 cells, cluster number: n (KCl) = 77 cells, n (NaCl) = 84 cells. Grey dot indicates number of individual cells, cyan triangle indicates mean of each experiment. **(G)** Quantification of synpo cluster size at the AIS of DIV14 neurons upon induction of chronic AIS plasticity. Mean ± SEM, three independent cultures, KCl: n = 73 cells, NaCl: n = 85 cells. Grey dot indicates number of individual cells, cyan triangle indicates mean of each experiment. **(H)** Distribution of synpo cluster at the AIS of nonAcD neurons upon induction of chronic plasticity. In total three independent cultures, n (KCl) = 420 clusters from 79 cells, n (NaCl) = 501 clusters from 86 cells. Mann–Whitney test (two-sided): not significant (n.s.) P > 0.05, *P < 0.05, **P < 0.001. More detailed statistical information see Data S1.

percentage is lower (20–30% in CA3 and subiculum) (Thome et al., 2014). As our model system contains neurons from all hippocampal regions, this may average the percentage in our study. Moreover, the 50% AcD neuron population was quantified in adult mice (P28-35), while the age of neurons quantified in our study was much younger (DIV5 and DIV7). This age disparity could also cause a difference in the amount of AcD neurons, as their percentage in younger animals (20–30%) is much lower (Benavides-Piccione et al., 2020; Thome et al., 2014). It should be noted that our data cannot be directly compared with results from brain sections as primary neurons are a more simplified model. Further, mature neurons can convert between nonAcD and AcD morphologies, and this morphological transition depends on network activity (Lehmann et al., 2023, Preprint). Since the connectivity between neurons in vitro differs from in vivo, fewer cells might convert into an AcD morphology at a mature stage.

The compartmentalization of axonal and dendritic proteins is an essential step in the establishment and maintenance of neuronal polarity. Such segregation is achieved by means of polarized cargo trafficking mediated by the coordinated interplay of cargo adapters, motor proteins, the cytoskeleton, and the presence of the AIS (Balasanyan et al., 2017; Kapitein and Hoogenraad, 2011; Leterrier, 2018; Rasband, 2010). By showing that AcD neurons preserve axonal MT arrangement at the stem dendrite and that their AIS is well-suited for filtering out dendritic cargos, we demonstrated how neurons can still achieve targeted protein delivery even in the case of an axon starting from a dendrite. Additionally, we found that AcD neurons are enriched in dynamic MTs at the stem dendrite. In the future, it would be interesting to investigate the origins and consequences of such enrichment and if there are more MT nucleation sites present in the stem dendrite.

Our data suggest that the faster velocity of cargo trafficking in the AIS of AcD neurons is specific to Rab3A-associated synaptic vesicles as other tested cargoes (NPY and LAMP1) traveled through the AIS with similar speed between AcD and nonAcD neurons. One can speculate that Rab3A vesicles may associate with certain adaptor proteins or specific PTMs and MAPs are present in the AIS of the AcD neurons that might facilitate the KIF1A and KIF1Bβ-based transport.

Inhibitory synapses at the AIS of excitatory neurons are important modulators for the activation and silencing of specific neuronal circuits (Nathanson et al., 2019). By using VGAT and gephyrin as markers, we observed lesser inhibitory synapses at AcD neuron's AIS compared with nonAcD neurons. This result may seem contradictory to what Hodapp et al. found previously in slices, i.e., that AcD and nonAcD neurons receive a similar amount of inhibitory input during ripples. However, it should be emphasized that our finding is specific to the AIS, which in fact supports the notion that AcD neurons can evade peri-somatic inhibition during network oscillation, as the AIS may initiate APs more easily by having less inhibitory constraints.

We showed that the AIS of excitatory AcD neurons does not undergo activity-dependent plasticity, suggesting that a different strategy might be employed by AcD neurons to maintain cellular homeostasis. Of note, we used KCl to induce AIS plasticity. Instead of manipulating postsynaptic activity, KCl stimulation only depolarizes the membrane to mimic the situation of neurons being hyperactivated (Rienecker et al., 2020). This stimulation protocol indeed models certain pathological conditions, such as seizures, when extracellular $K^+$ concentration is increased, and suggests that AcD neurons are perhaps less adaptive than nonAcD neurons, and are therefore more susceptible to pathological conditions. Nonetheless, the AIS plasticity in AcD neurons should also be tested in vivo and by specifically manipulating actual postsynaptic activity, as certain molecules specific to postsynaptic activation might be required to trigger AIS plasticity. In line with this hypothesis, short-term AIS plasticity was recently observed in hippocampal principal excitatory neurons in vitro and ex vivo after incubating with N-methyl-D-aspartate (NMDA), a glutamate agonist, to stimulate NMDA receptors at the postsynapses (Fréal et al., 2023), but not with KCl stimulation. It would be interesting to test the same stimulation protocol in hippocampal excitatory AcD neurons.

The regulatory mechanisms underlying AIS plasticity are still ambiguous. To this date, several lines of evidence suggest the significance of $Ca^{2+}$ signaling in this process (Evans et al., 2013; Grubb and Burrone, 2010). Cisternal organelles are particularly interesting in this regard given their putative role for regulating $Ca^{2+}$ transient at the AIS (Benedeczky et al., 1994; Jungenitz et al., 2023; Bas Orth et al., 2007; Sánchez-Ponce et al., 2011). First, a recent ex vivo study showed a reduction of the cisternal organelle number when rapid AIS plasticity in rat hippocampal DGCs is induced (Jungenitz et al., 2023). Here, we could show that after induction of chronic plasticity, less number of cisternal organelles were present in the AIS of nonAcD neurons. This finding also correlates with our observation that fewer cisternal organelles are present in the AIS of AcD neurons. As these AcD neurons are presumably more excitable when the input comes from the AcD, it is possible that they have already reached their limit in terms of AIS length and localization plasticity. Our results suggest that the number of cisternal organelles is more likely to be the consequence than the cause of the diminished AIS plasticity in AcD neurons. In the future, it would be interesting to test not only the direct relationship between cisternal organelles, $Ca^{2+}$ dynamics, and AP generation but also to investigate the mechanistic role of cisternal organelles in AIS plasticity. Ideally, these findings should be validated in animal models, such as synaptopodin knockout mice. In addition to $Ca^{2+}$ signaling, the endocytic machinery might also be involved in regulating the AIS morphology (Eichel et al., 2022; Wernert et al., 2024; Fréal et al., 2023). It has been shown that endocytosis of sodium channels is required to drive rapid AIS plasticity (Fréal et al., 2023). Additionally, several new AIS-specific proteins were discovered via proximity labeling assays (Hamdan et al., 2020). It would be interesting to test whether they were involved in regulating AIS plasticity and to compare their distribution between AcD and nonAcD neurons. Another direction to explore is the link between the AIS-specific ECM and AIS plasticity. Regardless of our data showing the absence of brevican in younger neuronal cultures (DIV12), it is still possible that the ECM plays a role in older neurons, as AIS is plastic throughout the entire lifespan (Grubb and Burrone, 2010;

Gulledge and Bravo, 2016; Jahan et al., 2023; Jungenitz et al., 2023; Kuba et al., 2006; Susuki and Kuba, 2016; Yamada and Kuba, 2016).

Our study opens several interesting new lines of inquiry. How is a branch at the proximal part of the precursor axon in AcD neurons formed, stabilized, and able differentiate into a dendrite? Is it rather a stochastic process, or a genetically encoded program leading to the development of a specific type of glutamatergic neuron? To answer these questions, it will be important to combine live imaging with the interrogation of molecular mechanisms that have been reported to assist neurons in collateral genesis and the differentiation of neurites. It also might be that the formation of AcD neurons is a fully stochastic process that depends on probabilities and speed of neurite growth and retraction. In-depth analysis of longitudinal imaging data combined with mathematical modeling could help to address this point. It would also be of further interest to perform single-cell RNA sequencing to analyze the gene expression profiles of AcD neurons and possibly identify specific markers of this cell type.

As reported by Wahle et al. recently (Wahle et al., 2022), a third category of excitatory neurons, known as shared root neurons, was also found in brain sections across species. The axon of these neurons almost grows together with a basal dendrite and it is therefore hard to identify its axon origin. In our study, we classified this type of neurons as nonAcD, as there is no evidence showing that their behavior is different from non-AcD neurons. However, it is a very interesting direction to explore in the future. Collectively, our study provided new insights into the in vitro development of AcD neurons and demonstrated that there are differences in AIS functionality between AcD and nonAcD neurons.

## Materials and methods

### Animals
Wistar Unilever HsdCpb:WU (Envigo) rats were used in this study. Rats were bred and kept at the animal facility of the University Medical Center Hamburg-Eppendorf, UKE, Hamburg, Germany. Animal experiments were carried out in accordance with the European Communities Council Directive (2010/63/EU) and the Animal Welfare Law of the Federal Republic of Germany (Tierschutzgesetz der Bundesrepublik Deutschland, TierSchG) approved by the city-state Hamburg (Behörde für Gesundheit und Verbraucherschutz, Fachbereich Veterinärwesen, from 21.04.2015, ORG781 and 1035) and the animal care committee of the University Medical Center Hamburg-Eppendorf. For STORM imaging, rat primary hippocampal neurons were prepared in the Neurocyto lab (Marseille, France), according to the guidelines established by the European Animal Care and Use Committee (86/609/CEE) and approved by the local ethics committee (agreement D13-055-8).

### Primary hippocampal neuron preparation and transfections
Primary rat hippocampal neurons were prepared and maintained as described previously (Konietzny et al., 2019). Briefly, hippocampi were extracted from E18 rat embryos and treated with 0.25% trypsin for 15 min at 37°C. Afterward, hippocampi were physically dissociated by pipetting through a 26G needle and filtered to remove large clumps. The cell suspensions were then plated on poly-L-lysine-coated 18-mm glass coverslips or 35-mm glass bottom petri dish at a density of 20,000 cells (extra low density) or 40,000 cells (low density) or 60,000 cells (high density) per 1 ml in DMEM supplemented with 10% fetal calf serum and antibiotics. After 1 hour, the plating medium was replaced with BrainPhys neuronal medium supplemented with SM1 and 0.5 mM glutamine. Cells were grown at 37°C with 5% $CO_2$ and 95% humidity.

For the MT orientation experiment, primary neurons were transfected with EB3-tdTomato at DIV 14 by using lipofectamine 2000. The conditioned neuronal medium was removed and incubated at 37°C with 5% $CO_2$ before transfection. Neurons were then incubated with transfection medium (transfection mixture in BrainPhys medium without supplements) at 37°C with 5% $CO_2$ for 1 h. After incubation, the transfection medium was exchanged back to the conditioned medium, and neurons were imaged 12 h later. For the EGFP-Rab3 trafficking experiment, primary neurons were infected with a rAAV9-syn-EGFP-Rab3 virus (final concentration: 1.38E+10 vg/ml) at DIV5 and imaged at DIV7.

For mCherry-LAMP1 trafficking experiment, primary neurons were infected with rAAV9-syn-LAMP1-mCherry virus (final concentration: 2.7E+10 vg/ml) at DIV7 and Imaged at DIV10. For NPY-mEGFP trafficking experiment, primary neurons were co-transfected with NPY-mEGFP and Tag-RFP constructs at DIV10 using lipofectamine 2000 as described above and were imaged 12–16 h later.

### Constructs and recombinant adeno-associated viruses
Detailed information on constructs and viruses used in this study can be found in the Table S1. The EB3-tdTomato construct was obtained from Addgene (plasmid #50708; Addgene; https://n2t.net/addgene:50708; RRID: Addgene_50708). For the rAAV9-syn-EGFP-Rab3A construct, full-length Rab3A sequence was amplified from the EGFP-Rab3A vector (plasmid #49542; Addgene) using PCR. The amplified sequence was then cloned into a recombinant adeno-associated virus (rAAV) backbone with an EGFP tag following a human synapsin promotor using homologous recombination. The Rab3A sequence was inserted after the EGFP sequence for N-terminal tagging of Rab3A. After verification by sequencing, rAAV9 was produced by the UKE virus facility. The rAAV9-syn-LAMP1-mCherry pDNA construct was cloned as described previously in van Bommel et al. (2019). The AAV production was done by the UKE vector facility. The NPY-mEGFP construct is the same as in Schlager et al. (2010).

### Immunocytochemistry (ICC)
Neurons were fixed in 4% Roti-Histofix (Carl Roth) and 4% sucrose in PBS for 10 min at 37°C. The fixation reagent was then removed and coverslips were washed three times with PBS. Subsequently, neurons were permeabilized in 0.2% Triton X-100 in PBS for 10 min, then washed three times in PBS and blocked for 45 min at RT with blocking buffer (BB: 10% horse serum and 0.1% Triton X-100 in PBS). Incubation with primary

antibodies was performed in BB at 4°C overnight. After three washes in PBS, neurons were incubated with corresponding secondary antibodies in BB for 1.5 h at RT and unbound antibodies were washed out using PBS. To distinguish dendrites from axons, preconjugated MAP2-Alexa488 antibody diluted in BB was then applied to neurons for 1.5 h at RT. Finally, coverslips were washed three times in PBS (10 min intervals), one time in $H_2O$ for 10 s, and then mounted on microscope slides with Mowiol.

**Spinning disc confocal and TIRF microscopy**
For live imaging of EB3-tdTomato, EGFP-Rab3A, and transferrin receptors (TfRs), spinning-disc confocal microscopy was performed on a Nikon Ti-2E controlled by NIS Elements 5.2 software. Illumination was done by 488, 561, and 642 nm excitation lasers from an Omicron laser unit coupled to a Yokogawa CSU-W1 spinning disc unit. Emission was collected through a bandpass filter (442/525/609/700 nm; Semrock) on an Andor iXon ultra 888 EMCCD camera. Use of 100x TIRF objective (ApoTIRF 100x/1.49 oil; Nikon) achieved a pixel size of 130 nm.

For imaging of neuronal development, phase contrast imaging was performed on a Nikon Ti-2E microscope controlled by NIS Elements 5.2 software. The microscope was equipped with an Omicron laser unit coupled to a Yokogawa Borealis-enhanced CSU-W1 spinning disc unit, transmitted illumination, and an Andor iXon ultra 888 EMCCD camera. Samples were illuminated by transmitted light and imaged via a 40X objective (ApoLWD Lambda 40x/1.15 WI; Nikon). The final fluorescent map was acquired on the same microscope through the same objective with laser illumination at a wavelength of 488 and 642 nm under confocal mode. The achieved pixel size for phase contrast and confocal imaging was 329 nm. Both microscopes were equipped with a live imaging system from OKO lab and live imaging was performed at 37°C with 5% $CO_2$ and 95% humidity.

For live imaging of LAMP1-mCherry and NPY-mEGFP, total internal reflection fluorescence microscopy (TIRFM) was performed with a Visitron Spinning-Disc TIRF-FRAP system on a Nikon Eclipse Ti-E controlled by VisiView software (Visitron Systems). Samples were incubated in a stage top incubator (Okolab) with 37°C, 5% $CO_2$, and 90% humidity atmosphere and were kept in focus with the built-in Nikon perfect-focus system. For neurons cultured on 18-mm coverslips, the coverslips were first mounted onto a Ludin chamber (Life Imaging Services) and then placed into the stage-top incubator. The fluorophores were excited by 488, 561, and 640 nm laser lines that are coupled to the microscope via an optic fiber. The samples were imaged with a 100x TIRF oil objective (ApoTIRF 100x/1.49 oil; Nikon). Oblique and TIRF illuminations were obtained with an ILAS2 (Gattaca systems) TIRF system. Multichannel z-stack images and time-lapse images were acquired sequentially using an appropriate filter set with an Orca flash 4.0LT CMOS camera (Hamamatsu). The final pixel size was 65 nm.

For imaging of fixed neurons co-immunostained with gephyrin and VGAT, confocal spinning disc imaging was performed with a Visitron Spinning-Disc system on a Nikon Eclipse Ti-E equipped with the Yokogawa CSU-X1 unit. The system was controlled by VisiView software (Visitron Systems), and Multichannel z-stacks were taken sequentially using an appropriate filter set with an Orca flash 4.0LT CMOS camera (Hamamatsu) or a pco.edge 4.2 bi sCMOS camera (Excelitas PCO GmbH). The fluorophores were excited by 405, 488, 561, and 640 nm laser lines, coupled to the microscope via an optic fiber. The samples were imaged with a 100x TIRF oil objective (ApoTIRF 100×/1.49 oil; Nikon) to achieve a final pixel size of 65 nm. For AcD percentage analysis of DIV21 neurons and pan-Nav, samples were imaged on the same system but with a 60x TIRF oil objective (P-Apo DM 60x/1.40 oil; Nikon), achieving a pixel size of 108 nm. Tile-scans were taken using ScanSlide module with a 10% overlap to acquire large areas. Tile scan images were stitched in FIJI.

For imaging of fixed neurons labelled with AnkG, neurofascin, tyr-α tub, and ac-α tub, confocal spinning disk imaging was performed on a Nikon Ti2-E microscope equipped with Yokogawa CSU-W1 spinning disk unit, solid-state excitation lasers (at 405, 488, 561 and 640 nm wavelength) and a Sora pixel reassignment module. Multichannel z-stacks were taken sequentially using an appropriate filter set with a Fusion BT sCMOS camera (Hamamatsu). Samples were imaged with a 60x TIRF oil objective (ApoTIRF 60×/1.49 oil; Nikon), and 4x SoRa magnification was used to achieve a higher resolution with a final pixel size of 27 nm.

For all live imaging experiments, neurons were kept and imaged in BrainPhys medium with SM1 supplement.

**Laser scanning confocal microscopy and STED imaging**
Fixed and stained primary hippocampal neurons were imaged at a Leica SP8 confocal microscope (Leica microsystems). The microscope was controlled by Leica Application Suite X (LASX) software and equipped with a white light laser. Samples were imaged using a 63x oil objective (63x HCX PL APO/1.40 oil; Leica). Fluorophores were excited at the desired wavelength and signals were detected using HyD detectors. Tile scans of a maximum 2 × 2 mm areas were performed to increase the chance of finding AcD neurons. A single z-stack tile was acquired with a dimension of 1,024 × 1,024 pixels, pixel size of 80 nm, pixel depth of 16 bit, and z-step size of 0.5 μm. Tiles were merged with the LASX function Mosaic Merge with a 10% overlap.

An Abberior gSTED system equipped with a 405–640 nm pulsed laser and a 60x oil objective (P-APO 60x/1.40 oil; Nikon) was used for confocal and gSTED imaging. For excitation, a 640-nm laser was used for Atto647N/Abberior Star 635p, a 561-nm laser was used for Abberior Star 580, and a 488-nm laser was used for Alexa 488. STED resolution was achieved with a 775 nm pulsed depletion laser for Abberior Star 580, Abberior Star 635p, and Atto647N. Emission spectra were collected between 650–720 nm, 605–625 nm, and 500–550 nm. Detector time gates were set to 8 ns for all fluorophores. Images were acquired as single planes with a pixel size of 20 × 20 nm (x and y) and 16-bit pixel depth. The corresponding confocal images were acquired with identical optical settings, and the pixel size of confocal images was 80 nm. The imaging medium for all fixed samples was Mowiol.

## Classification of AcD and nonACD neurons

AcD and nonAcD neurons were classified based on MAP2 and AnkG staining which labels the somato-dendritic compartment and the AIS of the axon, respectively. The distance from the starting point of an axon to the adjacent end of the corresponding cell body was referred to as "axon distance" in this study and used as the main factor for AcD neuron classification (Fig. 1 B and Fig. S1 A). The diameter of the stem dendrite is the second factor for AcD neuron classification (Fig. 1 B and Fig. S1 A). To measure the axon distance, the axon and cell body of a neuron were defined individually based on AnkG and MAP2 signals. A one-pixel wide segmented line was drawn along the longitudinal axis of the stem dendrite to connect the beginning of the axon and the ending edge of the cell body (Fig. 1 B). The length of the segmented line represents axon distance. If the axon origin was not parallel to the ending edge of soma (Fig. S1 A), the axon distance was then considered as the distance from the end of soma to the axis perpendicularly extended from the center of the axon origin (Fig. S1 A). The stem dendrite diameter measurement was based on MAP2 staining. Two lines were drawn at the start and the end of the stem dendrite to measure the width ($d_1$ and $d_2$; Fig. 1 B and Fig. S1 A). The diameter (d) was then defined as the averaged value of $d_1$ and $d_2$ $[d = (d1 + d2/2)]$. Neurons with axon distance longer than 2 μm and larger than stem dendrite diameter were considered AcD neurons. The AcD and nonAcD neurons were classified in the same manner throughout the entire study unless otherwise stated. Neurons with more than one axon were excluded in this study.

## Assessment of AcD neuron population in dissociated cultures and the timeline of AIS formation

Dissociated neurons (60,000 cells/ml) were fixed at DIV3, DIV5, DIV7, DIV12, and DIV21 and immunostained with anti-MAP2-Alexa488 antibody to label the cell body and dendrites. Antibodies against AnkG were used to visualize the AIS. Tile scans of large areas were taken for both MAP2 and AnkG channels with 1% laser power and 5 μm depth in z and 0.7-μm z-step size. For neurons fixed at DIV21, images were taken from a different microscope as described above with laser power of 10% for each wavelength and the same depth in z and z-step size. The total number of neurons at the age of DIV5, DIV7, and DIV21 was counted in all tile scan images. The number of AcD neurons was then counted in the same images and divided by the total number of neurons to calculate the percentage of AcD neurons in the population.

## Time-lapse imaging to visualize neuronal development in dissociated culture

High-density neurons (60,000 cells/ml) were cultured on a 35-mm glass bottom petri dish with 20 × 20 mm grid coordinates. A 3-mm diameter hole was drilled on the lid of the petri dish for replenishment of the conditioned medium. To prevent medium evaporation during the entire imaging period, the conditioned medium was refilled on a daily basis. Additional three 35-mm petri dishes were filled with distilled $H_2O$ and placed in the live imaging chamber without lids to increase atmospheric humidity. The $H_2O$ petri dishes were refilled every 12 h to maintain a constant humidity level within the live imaging chamber.

Developmental sequences were recorded 6 h after plating. Dishes were placed on a spinning disc confocal microscope and imaged for 5 days every 3 h. To increase the number of imaged AcD neurons, a 7 × 7 mm area was scanned spirally starting from the center. On day 5, neurons were fixed and immunostained with preconjugated anti-MAP2-Alexa488 and anti-AnkG antibodies to label dendrites and the AIS, respectively. A single focal plane fluorescent image of both MAP2 and AnkG channels was then taken at the same scanned area with 20% laser power and 200 ms exposure time. The fluorescent image was used as a final map for the selection of AcD and nonAcD neurons, and the developmental sequence of selected neurons was retrieved according to grid coordinates. The formation and growth of AcD neurons and the axon of AcD neurons were analyzed manually by going through the corresponding developmental sequences frame by frame. All analyses were performed in ImageJ (NIH).

## F-actin staining and analysis of periodic membrane actin cytoskeleton of the AIS

For F-actin staining, neurons at a density of 20,000 cells/ml were plated on 18-mm high precision glass coverslips, fixed on DIV14, and immunostained with anti-βIV-Spectrin and preconjugated anti-MAP2-Alexa488 antibody to label the AIS and dendrites, respectively. Following the antibody staining, F-actin was labeled by overnight incubation with phalloidin-Atto647N (1:100 dilution in PBS) at 4°C. Following three washes with PBS, coverslips were mounted using Mowiol and imaged on an Abberior gatedSTED system as described above.

Peak Cal 3.0, a custom-written Python-based script (see Table S1), was used to measure the F-actin periodicity at the AIS. Briefly, AIS regions with little phalloidin-Atto647N background signal were selected manually from the STED image. A 3 × 3 pixel broad segmented line profile was drawn at the selected regions along the longitudinal axis of the AIS, and the corresponding phalloidin-Atto647N intensity was then extracted using Fiji. The F-actin rings are hence represented by the periodical peaks of the phalloidin-Atto647N fluorescent signal along the segmented line. The extracted intensity profile was loaded into Peak Cal 3.0 and smoothed over five pixels to filter the background noise. The indices of phalloidin peaks were then detected using Python built-in function Find Peaks. The distance was calculated by subtracting the indices between two adjacent peaks and then multiplied by 20 nm pixel size.

## STORM imaging for AIS-specific cytoskeleton

To super resolve the AIS-specific MPS, rat hippocampal neurons were cultured following the Banker method, above a feeder glia layer (Kaech and Banker, 2006). Briefly, 12 or 18 mm-diameter round #1.5H coverslips were affixed with paraffin dots as spacers and then treated with poly-L-lysine. Hippocampi from E18 rat pups were dissected and homogenized by trypsin treatment followed by mechanical trituration and seeded on the coverslips at a density of 4,000–8,000 cells/cm² for 3 h in serum-containing plating medium. Coverslips were then transferred, cells down, to Petri dishes containing confluent glia cultures conditioned in B27-supplemented neurobasal medium and cultured in these dishes for up to 4 wk.

For this, neurons were fixed at DIV14 by following a previously published protocol (Jimenez et al., 2020). Briefly, neurons were fixed for 10 min at 37°C with 4% PFA and 4% sucrose dissolved in PEM buffer (80 mM PIPES, 2 mM $MgCl_2$, 5 mM EGTA, pH 6.8). Subsequently, neurons were washed three times with phosphate buffer and blocked for 1 h under room temperature with TpT (0.22% gelatine and 0.1% Triton in phosphate buffer). Neurons were then incubated with primary antibody against MAP2, AnkG, and βIV-spectrin (rabbit, M.N. Rasband) diluted in TpT buffer overnight at 4°C, washed 3x with phosphate buffer, and incubated with secondary antibodies for 1 h at room temperature. After washing 3x with phosphate buffer, neurons were incubated with phalloidin-Alexa647+ diluted in phosphate buffer (1:400 dilution) at room temperature for 1.5 h to label actin. Eventually, samples were kept in phalloidin-Alexa647+ at 4°C before imaging.

To super-resolve the spectrin and actin cytoskeleton at the AcD neuron's AIS, we performed dSTORM imaging. Briefly, samples were secured to a silicone perfusion chamber filled with reducing imaging buffer (Smart Kit Buffer A, enzyme solution, and 2-mercaptoethanol) and sealed with a glass slide. Prepared samples were mounted on an ECLIPSE Ti2 inverted microscope body (Nikon) with a high numerical aperture 100X oil-immersion objective lens (1.49 NA, CFI SR HP Apochromat TIRF 100XC; Nikon). Illumination was provided by two superposed 640 nm continuous wave diode lasers (Oxxius) with a combined power between 300 and 400 mW measured at the back aperture. An ASTER module (Abbelight) scanned the beam in a 70 × 70 µm, flat-top square illumination profile at the sample plane, according to Mau et al. (2021), yielding a power density between 6.12- and 8.16-kW cm$^{-2}$. Fluorophores in long-lived dark states were recovered with a 405 nm continuous wave diode laser in the same beam line at powers <15 mW measured at the back aperture. Fluorescence emission was detected in a split light path optimized for spectral demixing, analogous to that described in https://www.ncbi.nlm.nih.gov/pmc/articles/PMC10545913/, Friedl et al. (2023). Briefly, emission was isolated with a quad-edge dichroic beamsplitter (Di03-R405/488/532/635-t3-25x36; Semrock) and stray light–filtered further on a quad-band bandpass filter (FF01-446/510/581/703-25; Semrock). Astigmatic shaping of the PSF was achieved with a cylindrical lens, and a dichroic mirror (FF699-Fdi01-t3-25x36; Semrock) divided the filtered emission into reflected and transmitted image paths, detected by two water-cooled 16-bit sCMOS cameras (ORCA-Fusion BT; Hamamatsu Photonics). Acquisition sequences of 512 × 512 pixels were recorded for 60,000 frames at an exposure of 20 ms.

### Data processing for spectral demixing dSTORM
Paired single molecule localizations were detected in the reflected and transmitted images using the globLoc fitting algorithm (https://www.nature.com/articles/s41467-022-30719-4, Li et al. [2022]), incorporated as a module in the Super-resolution Microscopy Analysis Platform (SMAP) (https://www.nature.com/articles/s41592-020-0938-1, Ries [2020]). The localization step was performed on a CUDA-enabled NVIDIA GeForce RTX 3090 graphics processor, and localizations were subsequently classified into two channels according to the ratiometric distribution of photon intensity detected in the reflected and transmitted images. Localization coordinates were reconstructed as images using the ImageJ ThunderSTORM plugin (https://www.ncbi.nlm.nih.gov/pmc/articles/PMC4207427/, Ovesný et al. [2014]).

### MT extraction and staining
The staining of tyrosinated and acetylated MTs was performed according to a previously described protocol[33]. In short, very low-density neurons (20,000 cells/ml) were grown on 18-mm high precision glass coverslips till DIV10. Then neurons were pre-extracted for 1 min using MT-Extraction buffer (0.3% Triton X-100, 0.1% glutaraldehyde, 80 mM PIPES, 1 mM EGTA, and 4 mM $MgCl_2$; pH 6.8) and fixed for 10 min using EM grade 4% PFA and 4% sucrose in PBS at 37°C. After three washes in PBS, neurons were immunostained with antibodies against tyrosinated and acetylated α-tubulin for dynamic and stable MTs, respectively. An antibody against TRIM46 was used to locate the AIS. Samples were imaged on an Abberior gatedSTED system as described above. The distribution of stable and dynamic MTs along the width of a region of interest was displayed by extracting and plotting the fluorescent intensities of a 2 × 2 pixel line along the latitudinal axis.

For fluorescent intensity analysis, samples were imaged using a spinning disk confocal microscope with SoRa pixel reassignment module as described above to improve resolution. Images were taken with 10% power and 200 ms exposure time for 488 nm laser (TRIM46), 30% power and 400 ms exposure time for 561 nm laser (acetylated α-tubulin), and 30% power and 400 ms exposure time for 640 nm laser (tyrosinated α-tubulin). Z-stacks were taken with 0.2 µm step size for all channels. Different regions of neurons were outlined with the help of the polygon shape tool in FIJI to extract fluorescent intensity.

### Imaging of MT dynamics
High-density neurons (60,000 cells/ml) were grown on 35-mm glass bottom petri dish and transfected with EB3-TdTomato plasmid at the age of DIV13 and imaged 12 h after transfection using a spinning disk confocal microscope. Shortly before imaging, the conditioned medium was replaced by BrainPhys medium containing neurofascin-CF640R antibody (1:200 dilution) to label the AIS. Dishes were returned to the incubator and kept at 37°C with 5% CO2 for 5 min. Afterward, the conditioned medium was exchanged and neurons were returned to the incubator for 10 min to recover. AcD and nonAcD neurons were then identified based on the neurofascin and EB3-TdTomato channel. For this, images were acquired as z-stacks with 0.7 µm step size, 5% laser power, and 100 ms exposure time. Then, the EB3 channel of selected neurons was recorded as a time-lapse with a frame rate of 1.3 s/frame and 5% laser power for 5 min.

### Axonal and dendritic cargo trafficking assays
For the axonal cargo trafficking assay, high-density DIV5 neurons (60,000 cells/ml) on 35-mm glass bottom petri dishes were infected for 2 days with rAAV9 virus expressing EGFP-Rab3A. At

DIV7, the AIS was labeled at 37°C with 5% $CO_2$ for 5 min by replacing the conditioned medium with BrainPhys medium containing neurofascin-CF640R antibody (1:500 dilution). Neurons were then placed back into the conditioned medium for recovery and recording. AcD and nonAcD neurons were selected based on EGFP-Rab3A and neurofascin-CF640R fluorescence. The z-stack image was taken for both channels with 0.7 µm step size, 5% laser power, and 100 ms exposure time to verify neuronal morphology. To analyze the newly delivered EGFP-Rab3A vesicles and to improve the signal-to-noise ratio, the vesicles residing at the AIS, stem dendrite, and regular dendrite were photo-bleached using the 405 nm laser with 70% laser power for 5 s. New EGFP-Rab3A vesicles traveling through the photo-bleached regions were continuously imaged with five frames per second and 10% laser power for 2 min.

For the dendritic cargo trafficking assay, the AIS of high-density DIV7 neurons (60,000 cells/ml) grown on 35-mm glass bottom petri dishes was labelled as described above. Then, preconjugated Transferrin-Alexa568 (1:1,000 dilution) was used to label the endogenous TfRs by incubation for 15 min at 37°C with 5% $CO_2$ in the BrainPhys medium. Then neurons were returned to the conditioned medium for recovery. AcD and nonAcD neurons were selected based on Transferrin-Alexa568 and neurofascin-CF640R channels. Neuronal morphology was confirmed by z-stack imaging (0.7 µm step size, 5% laser power, and 50-ms exposure time) of both channels. Trafficking of TfRs in the AIS, stem dendrite, AcD, and regular dendrite was recorded with five frames per second and 10% laser power for 2 min.

For LAMP1-mCherry, high-density neurons (60,000 cells/ml) cultured on 35-mm glass bottom petri dishes or on 18-mm glass coverslips were infected at DIV7 for 3 days with rAAV9 virus expressing LAMP1-mCherry. At DIV10–DIV11, the AIS was labeled at 37°C with 5% $CO_2$ for 5 min by replacing the conditioned medium with a BrainPhys medium containing neurofascin-CF640R antibody (1:200 dilution). Neurons were then placed back into the conditioned medium for recovery and recording. AcD and nonAcD neurons were selected based on LAMP1-mCherry and neurofascin-CF640R fluorescence. The z-stack image was taken for both channels with a 0.6-µm step size to verify neuronal morphology (561 nm laser: 10% laser power and 200 ms exposure time for LAMP1-mCherry, 647 nm laser: 40% laser power and 300 ms exposure time for neurofascin). LAMP1-mCherry vesicles traveling through the AIS were continuously imaged with a frame rate of three frames per second, 300 ms exposure time per frame, and 10% laser power for 1–2 min. For neurons showing higher LAMP1-mCherry expression, the AIS was photobleached using the 405 nm laser with 70% laser power for 5 s to remove the residing stationary vesicles. LAMP1-mCherry vesicles traveling through the photobleached region were then recorded with the same imaging parameters.

For NPY-EGFP, high-density neurons (60,000 cells /ml) were co-transfected with NPY-mEGFP and tag-RFP by using lipofectamine2000 at DIV10 and imaged 12 h later. The AIS was labeled as described above. To verify neuronal morphology Z-stack images were taken for all channels with 0.4 µm step size. NPY-mEGFP vesicles traveling the AIS area were continuously imaged with a frame rate of five frames per second or with frame rate of three frames per second, 200 ms exposure time per frame, and 10% laser power for 1.5 min. For neurons showing higher NPY-mEGFP expression, the AIS was photobleached using a 405-nm laser with 70% laser power for 5 s. NPY-mEGFP vesicles traveling through the photobleached region were then recorded with the same imaging parameters.

## Analysis of MT dynamics and cargo trafficking assays

Time-lapse imaging data for EB3-tdTomato, NPY-EGFP, LAMP1-mCherry, EGFP-Rab3, and Transferrin-568 were analyzed using Fiji and self-written python scripts KYMOA 6.0 and KA Post Processing 2.0 (https://github.com/HU-Berlin-Optobiology/AIS-project.git). In short, segmented lines were drawn on time-lapse images at regions of interest (ROIs). The segmented line is along the anterograde direction (e.g., cell body to distal axon), and the width of the segmented line covers the width of the ROIs to include the majority of the signals. Corresponding kymographs were then generated using the FIJI plugin KymoResliceWide. The coordinates of each trajectory on the kymograph were extracted using the freehand tool and the get-coordinate function in FIJI. Subsequently, the coordinates were further processed in KYMOA 6.0 and KA Post Processing 2.0, as described by Konietzny et al. (2024). For EGFP-Rab3, Transferrin-568, NPY-EGFP, and LAMP1-mCherry, we defined vesicles having at least one movement, i.e., frame-to-frame change in the y-coordinates, over five continuous pixels and frames as mobile vesicles, and the corresponding movement was considered as a run. Otherwise, vesicles were classified as stationary. Several parameters were then calculated only for mobile vesicles, such as run length, run speed, run duration, and net displacement. To define the trafficking direction of a mobile vesicle, we took advantage of the net displacement. Mobile vesicles having a displacement above zero were classified as anterogradely transported vesicles, however, below zero were classified as retrogradely transported vesicles. During active transport, vesicles sometimes shortly pause at one spot for a few seconds. We thus defined pausing behavior as a mobile vesicle that stalled at the same position for at least three frames. It is worth noting that we often observed a vesicle repeatedly moving one pixel, stopping for one frame, and moving for one pixel again. Since those vesicles were neither fully stationary nor actively being transported, we specified this type of movement as passive movement. For EB3-tdTomato, the running and pausing threshold was set to two pixels and 2–10 frames, respectively. There was no threshold for passive movement because MT plus ends have unidirectional movements.

## Induction of rapid and chronic AIS plasticity

Induction of AIS plasticity in dissociated primary neuron cultures was performed following previously published protocols (Evans et al., 2015; Grubb and Burrone, 2010). Briefly, high-density dissociated primary neurons (60,000 cells/ml) were depolarized with 15 mM KCl to artificially increase neuronal activity, 15 mM NaCl was used as osmolarity control, and 1 µM Tetrodoxin (TTX) was used to silence the neuronal activity of

the entire culture. For rapid plasticity, neurons were treated for 3 h at DIV12 and then fixed for 10 min at 37°C. For chronic plasticity, neurons were treated from DIV12 to DIV14 for 48 h and then fixed for 10 min at 37°C. Fixed neurons were immunostained with preconjugated anti-MAP2-Alexa488 and anti-AnkG antibodies to label dendrites and AIS, respectively, and imaged using a laser scanning confocal microscope. Tile scan images were taken with 1% laser power for both channels and 5 μm depth in z.

## Analysis of AIS length and distance

The length of the AIS and its distance to the starting point of the axon were measured based on AnkG fluorescent intensity. Maximum projection of confocal z-stack images was used for this analysis. A three-pixel wide segmented line was drawn from the beginning of the axon toward the distal end to cover the entire AIS. The fluorescent intensity of AnkG staining along the segmented line profile was extracted in FIJI and processed by AIS Pack 4.0, a self-written Python script (https://github.com/HU-Berlin-Optobiology/AIS-project.git). The AnkG intensity profile was smoothed over 4 μm to filter out background noise, and the peak of the intensity profile was detected. To define the start and the end of the AIS, the algorithm iterates through intensity values from the peak to the left and right sides until the value is below 40% of the peak value. Consequently, the left edge of the AnkG intensity profile is the start of the AIS and the right edge is the end. Indices of the left and right edges are subtracted to calculate AIS length, and the AIS distance equals the left edge of the AnkG intensity profile.

## Immunostaining of inhibitory and excitatory synapses and cisternal organelle

Low-density DIV21 neurons (40,000 cells/ml) were fixed for 10 min at 37°C. For inhibitory and excitatory synapse stainings, fixed neurons were stained either with antibodies against VGAT and gephyrin for pre- and postsynaptic sites of inhibitory synapses or with antibodies against VGLUT and homer-1 to visualize the pre- and postsynaptic compartments of excitatory synapses. The AIS was stained either with AnkG or TRIM46 antibody depending on antibody species compatibility. MAP2-Alexa488 antibody was used to visualize cell bodies and dendrites. For the cisternal organelles, the AIS, cell body, and dendrite of fixed neurons were stained in the same way as above. Cisternal organelles were immunostained by an antibody against synaptopodin. All samples were imaged using a laser scanning confocal microscope, and a tile scan image was taken with 0.5–1% laser power and 5 μm depth in z (z step size: 0.5 μm). The number of synaptopodin puncta as well as the number of inhibitory and excitatory synapses were quantified using FIJI. Maximum projection of tile scan images was used for the analysis. Synaptopodin, VGAT, gephyrin, VGLUT, and homer-1 puncta within the AIS region were counted using the multipoint tool and normalized to the length of measured AIS.

For co-localization of pre- and postinhibitory synapses, fixed low-density (40,000 cells/ml) DIV21 neurons were co-immunostained with antibodies against VGAT and gephyrin. The AIS was labeled with an antibody against TRIM46, and the MAP2-Alexa488 antibody was used to indicate the somatodendritic compartment. Samples were imaged using a spinning disk confocal microscope from Visitron Systems as described above with 40% laser power for 405 nm laser (TRIM46), 15% laser power for 488 nm laser (MAP2), 15% laser power for 561 nm laser (gephyrin), and 15% laser power for 640 nm laser (VGAT). The exposure time was 300 ms for all wavelengths and z-step size was 0.2 μm. Representative images for AcD and nonAcD neurons were 3D reconstructed using Imaris.

The analysis of co-localized gephyrin and VGAT was performed in FIJI on a maximum projection image. To define co-localized gephyrin and VGAT clusters, we first defined an ROI at the AIS by using the Polygon tool and individually detected the particles in each channel at the ROI by using ParticleAnalyser in FIJI. Next, we restricted the maximum distance between the center of co-localized gephyrin and VGAT particle as 8 × 8 pixels (x and y). To calculate density, the number of co-localized gephyrin-VGAT particles was divided by the length of the measured AIS.

## Immunostaining of AIS specific ECMs and sodium channels

Low-density DIV21 neurons (40,000 cells/ml) were fixed for 10 min at 37°C. For AIS-specific ECMs, fixed neurons were stained with a preconjugated antibody against brevican. For sodium channels, fixed neurons were stained with a polyclonal antibody against alpha subunits of VGSCs (pan-Nav1). MAP2-Alexa488 antibody was used to visualize cell bodies and dendrites. An antibody against TRIM46 was used to identify the AIS for neurons stained with pan-Nav.

Samples stained with Brevican were imaged using a laser scanning confocal microscope, and a tile scan image was taken with 5–10% laser power and 5–7 μm depth in z (z step size: 0.5 μm). Samples stained with pan-Nav1 were imaged using spinning disk confocal microscope from Visitron system as described above, with laser power of 100% for 640 nm laser (pan-Nav1), 100% for 561 nm laser (TRIM46), and 60% for 488 nm laser (MAP2). The fluorescent intensity of pan-Nav1 was quantified using FIJI. Maximum projection of the stack images was used for the analysis. The AIS region was outlined with the polygon shape tool and average intensities of pan-Nav1 were extracted from the outlined shape and normalized to the background of each image.

## Immunostaining of AIS-specific scaffolding and membrane proteins

Low-density DIV21 neurons (40,000 cells/ml) were fixed for 10 min at 37°C. Neurons were then immunostained with antibodies against AIS-specific scaffolding proteins AnkG or the membrane protein neurofascin. Samples were then imaged on a spinning disk confocal microscope equipped with SoRa pixel reassignment module with 30% power and 400 ms exposure time for 488 nm laser (MAP2) and 40% power and 400 ms exposure time for 561 nm laser (neurofascin or AnkG). The intensity of AnkG and neurofascin was measured the same way as pan-Nav1.

## Analysis of inhibitory neuron density

DIV14-DIV15 neurons (20,000–30,000 cells/ml) were fixed at 37°C with 4% PFA/4% sucrose in PBS for GAD1 and at RT for

pCaMKII. The MAP2-Alexa488 antibody was used to visualize cell bodies and dendrites. GAD1 samples were imaged using a laser scanning microscope as described above with 1–5% laser power. Tile scans were taken to cover large areas. For samples stained against pCaMKII, images were taken at a spinning disk confocal microscope from Visitron as described above but with a 20x air objective. All images were processed in FIJI. Cell bodies were selected with circular ROIs in the MAP2 channel and the mean intensity was measured in both channels. A threshold was determined to select neurons positive for pCaMKII or GAD1 and the number of positive neurons was divided by the total number of MAP2 positive neurons. For GAD1, somata were additionally detected in the GAD1 channel due to their often lower MAP2 expression in the soma. Quantification of 399 fields of view (fov, $665.6 \times 665.6$ μm²; $N = 6$ independent cultures) was done for pCaMKII samples. Quantification of 12 tile scans ($1,116 \times 1,115$ μm²; $N = 2$ independent cultures) was performed for GAD1 samples.

### Analysis of cisternal organelle size and distribution in the AIS

The cisternal organelle size was measured using a self-written FIJI plugin synpo_det (https://github.com/HU-Berlin-Optobiology/AIS-project). Briefly, the AIS was manually outlined based on AnkG and MAP2 immunofluorescence using the polygon shape tool in FIJI. Single-channel images containing Synpo staining as a cisternal organelle marker were duplicated and thresholded. The thresholded image was then processed with fine-edges filter to highlight Synpo clusters within the AIS. The area size of the outlined Synpo clusters was then measured to represent Synpo cluster size.

To analyze the distribution of cisternal organelles within the AIS, a segmented line (three-pixel width) was first drawn based on AnkG and MAP2 staining along the central axis of the AIS in FIJI. The line extended from the start of the axon in the distal direction to cover the entire AIS. Subsequently, the coordinates of the pixels along the segmented line were extracted as a reference. Next, the coordinate of each Synpo cluster within the AIS was extracted in FIJI. The Synpo cluster coordinates were then traced back to the reference coordinates extracted from the segmented line using a self-written Python program "AIS_synpo_cluster_analysis." The distance of each traced Synpo cluster coordinate to the beginning of the reference line was then measured to represent the location of cisternal organelles in the AIS.

### Statistical analysis and image representation

Statistical analysis was performed on Prism (version 7.03) and R. Data are represented as percentages or as mean ± SEM, and outliers were removed by calculating the interquartile range (IQR). Individual channels in multicolor images are contrasted for better representation, with set minimum and maximum identical if groups need to be compared. No other modifications were done unless otherwise stated. All analysis was done on raw images.

### Reagents and resources

Detailed information on reagents and resources is documented in Table S1.

### Online supplemental material

Fig. S1 shows AcD neuron classification, axon distance, and stem dendrite diameter of AcD neurons, gallery of images containing nonAcD and AcD neurons, AIS diameter of AcD neurons, and percentage of AcD neurons. Fig. S2 shows additional information on trafficking of TfR and Rab3A vesicles in AcD neurons. Fig. S3 shows the analysis of trafficking for NPY and LAMP1 vesicles within the AIS of AcD and nonAcD neurons. Fig. S4 shows images of neurons immunostained with AIS-specific ECM protein brevican, markers for pre- and postexcitatory synapses, and quantifications. Fig. S5 shows the schematic of the AIS plasticity experiment and analysis, length of the AIS upon induction of AIS plasticity, analysis of AIS plasticity, and chronic AIS plasticity in GAD1-negative AcD neurons. Table S1 documents detailed information on reagents and resources used in this study. Video 1 shows in vitro developmental sequence of nonAcD and AcD neurons—example 1. Video 2 shows in vitro developmental sequence of nonAcD and AcD neurons—example 2. Video 3 shows in vitro developmental sequence of nonAcD and AcD neurons—example 3. Video 4 shows time-lapse imaging of EB3-tdTomato in AcD neurons. Video 5 shows time-lapse imaging of EB3-tdTomato in nonAcD neurons. Video 6 shows trafficking of EGFP-Rab3A and TfR vesicles in AcD neurons. Video 7 shows trafficking of EGFP-Rab3A and TfR vesicles in nonAcD neurons. Video 8 shows trafficking of NPY-EGFP and LAMP1-mCherry in AcD neurons. Video 9 shows trafficking of NPY-EGFP and LAMP1-mCherry in nonAcD neurons. Video 10 shows 3D-reconstruction of neurons immunostained with gephyrin, VGAT, TRIM46 and MAP2. Data S1 contains detailed information for statistics.

### Data availability

The data are available from the corresponding author upon reasonable request.

## Acknowledgments

We would like to thank Bianca Slivinschi for experimental assistance and help with data analysis, Zhengyue Zhang (MUNI-CEITEC, Brno, Czech Republic) for help with Python scripting, and Tomas Fanutza (ZMNH, Hamburg, Germany) for help with sample preparation. Julia Sandberg and Simone Traeger (CSSB, Hamburg, Germany) as well as Nathalie Hertrich and Lisa Mallis (HU, Berlin, Germany) for technical support, Dr. Antonio Virgilio Failla (UMIF, Hamburg, Germany) for help with use of the Abberior STED confocal microscope, Erich Weisheim for help with image processing, the Kreutz group (ZMNH, Hamburg, Germany) for help with preparation or primary neurons, and Fanny Boroni-Rueda and Florence Pelletier (INP, Marseille, France) for STORM sample preparation. We acknowledge Dr. Anja Konietzny (ZMNH, Hamburg, Germany) for valuable feedback on the manuscript and Dr. Ingke Braren (UKE Virus Facility) for rAAV production. We thank the INP NCIS imaging facility and Nikon Center of Excellence for Neuro-NanoImaging for service and expertise, with funding from Excellence Initiative of Aix-Marseille University, A*MIDEX, a French "Investissements d'Avenir" program (AMX-19-IET-002) through the Marseille Imaging and NeuroMarseille Institutes.

This research is supported by the Landesforschungsförderung Hamburg (LFF to M. Mikhaylova and K. Grünewald), by the German research foundation (Excellence Strategy – EXC-2049–390688087 to M. Mikhaylova, DFG CRC1315 project A03 to M. Mikhaylova, DFG FOR5228 to M. Mikhaylova and DFG FOR2419 to M. Mikhaylova and K. Grünewald), and by European Molecular Biology Organization scientific exchange grants (SEF fellowship 10691 to Y. Han). Open Access funding provided by Humboldt-Universität zu Berlin - Universitätsbibliothek.

Author contributions: Y. Han: Conceptualization, Data curation, Formal analysis, Funding acquisition, Investigation, Methodology, Project administration, Resources, Software, Supervision, Validation, Visualization, Writing - original draft, Writing - review & editing, D. Hacker: Formal analysis, Investigation, Software, Visualization, Writing - review & editing, B.C. Donders: Data curation, Formal analysis, Investigation, Validation, Visualization, Writing - review & editing, C. Parperis: Formal analysis, Investigation, Resources, Writing - review & editing, R. Thuenauer: Investigation, Methodology, Resources, Writing - original draft, Writing - review & editing, C. Leterrier: Resources, Supervision, Writing - review & editing, K. Grünewald: Conceptualization, Funding acquisition, Methodology, Project administration, Resources, Supervision, Writing - review & editing, M. Mikhaylova: Conceptualization, Data curation, Funding acquisition, Project administration, Resources, Supervision, Writing - original draft, Writing - review & editing.

Disclosures: The authors declare no competing interests exist.

Submitted: 26 March 2024

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

# Supplemental material

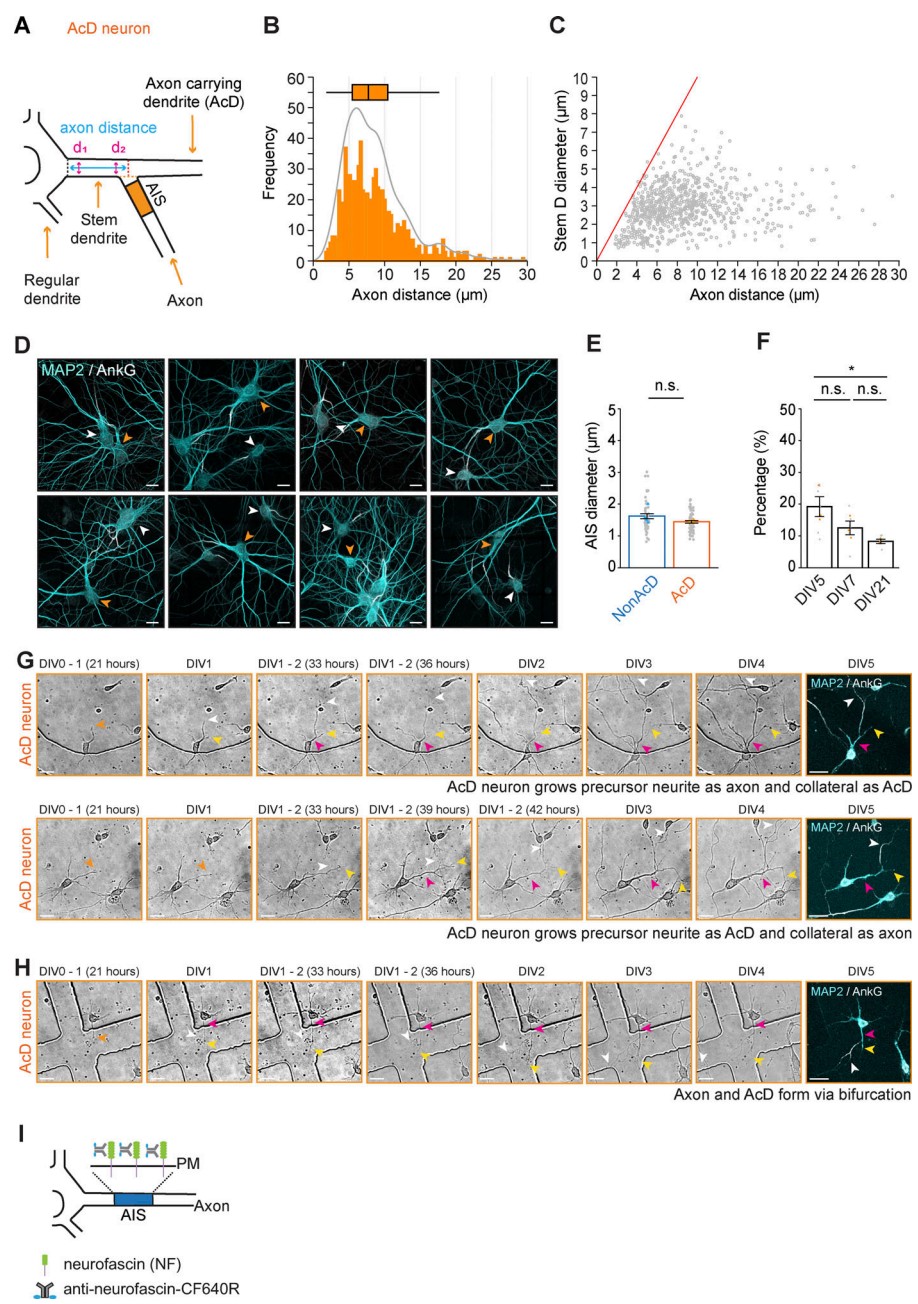

Figure S1. **AcD neuron identification and examples of atypical sequence of events during AcD neuron development. (A)** Schematic of AcD neuron where the point of axon start and end of soma are not parallel. Black dashed line indicates the edge of soma; orange dashed line indicates the border of axon start; red dashed line indicates the axis perpendicularly extended from the center of axon start. Related to Fig. 1 B. **(B)** Histogram of axon distance of AcD neurons. The median of axon distance is 7.75 μm (indicated by box plot). n = 800 cells. **(C)** Scatter plot of AcD neuron axon distance versus stem dendrite (Stem D) diameter. n = 800 cells. **(D)** Representative confocal images of nonAcD neurons and AcD neurons with different axon distance. White arrowhead indicates nonAcD neurons, orange arrowhead indicates AcD neurons. Scale bar is 20 μm. **(E)** AIS diameter measurement. Mean ± SEM, 4 independent cultures, nonAcD: n = 46 cells, AcD: n = 62 cells. Grey dot indicates value of individual cell. Cyan and orange triangles indicate mean of each experiment. Mann–Whitney test (two-sided): not significant (n.s.) P > 0.05. **(F)** Quantification of AcD neuron percentage in dissociated culture at different age. Grey dot indicates the percentage of individual images, orange triangle indicates mean of each experiment. Mean ± SEM, three independent cultures for each age, DIV5: n = 6 coverslips, DIV7: n = 6 coverslips, DIV21: n = 6 coverslips. One-way ANOVA with Tukey's multiple comparisons test, no significance (n.s.) P > 0.05, *P < 0.05. **(G)** Top row: Representative images from a time-lapse recording of AcD neuron developing a precursor neurite as axon and collateral as AcD (majority of cases). Related to Fig. 1 C and Video 2. Bottom row: Representative images from time-lapse recording of AcD neuron growing a precursor neurite as AcD and collateral as axon (atypical sequence). Related to Video 3 A. Orange arrowhead indicates the precursor neurite of AcD neuron. White arrowhead indicates axon. Yellow and pink arrowhead indicates AcD and stem dendrite, respectively. Dendrites and the AIS is labeled by poststaining with MAP2 and AnkG, respectively. Scale bar is 20 μm. **(H)** Representative images from a time-lapse recording of an AcD neuron developing an axon and AcD via bifurcation of the precursor neurite's growth cone. Related to Fig. 1 D and Video 3 B. Orange arrowhead indicates precursor neurite. White arrowhead indicates axon. Yellow and pink arrowhead indicates AcD and stem dendrite respectively. Cell body with dendrites and the AIS are labeled with MAP2 and AnkG antibody, respectively. Scale bar is 20 μm. **(I)** Schematic of neurofascin antibody live-labeling assay.

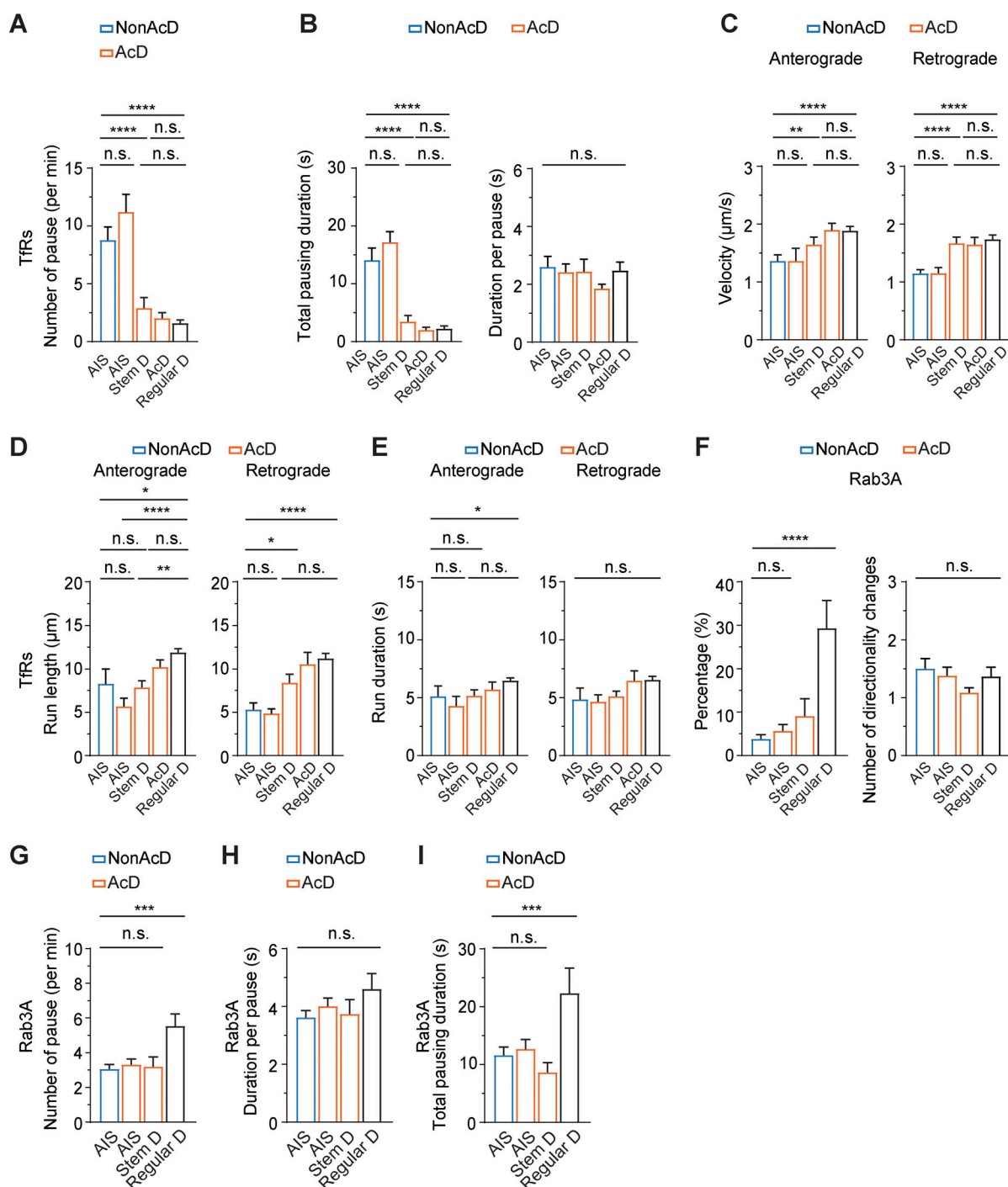

Figure S2.  **Trafficking of axonal and dendritic cargoes in nonAcD and AcD neurons. (A)** Average number of pause of TfR vesicles travelling in the indicated regions of nonAcD and AcD neurons (regardless of directions). Mean ± SEM, five independent cultures, AIS (nonAcD) *n* = 25 cells, AIS (AcD) *n* = 23 cells, Stem D *n* = 23 cells, AcD *n* = 19 cells, Regular D *n* = 56 cells. **(B)** Total pausing duration (left) and time of each pause (right) of TfR vesicles travelling in the indicated regions of nonAcD and AcD neuron (regardless of directions). Mean ± SEM, five independent cultures, AIS (nonAcD) *n* = 25 cells, AIS (AcD) *n* = 23 cells, Stem D *n* = 23 cells, AcD *n* = 19 cells, Regular D *n* = 56 cells. **(C–E)** Velocity (C), run length (D) and run duration (E) of TfR vesicles running towards anterograde and retrograde directions in the AIS of nonAcD neuron, the AIS, stem dendrite (Stem D) and AcD of AcD neuron, and the regular dendrite (Regular D) of both nonAcD and AcD neuron. Mean ± SEM, five independent cultures, Anterograde: AIS (nonAcD): *n* = 21 cells, AIS (AcD) *n* = 15 cells, Stem D: *n* = 23 cells, AcD: *n* = 19 cells, Regular D: *n* = 54 cells, Retrograde: AIS (nonAcD) *n* = 23 cells, AIS (AcD) *n* = 23 cells, Stem D *n* = 20 cells, AcD *n* = 18 cells, Regular D *n* = 56 cells. **(F)** Percentage of EGFP-Rab3A vesicles change directions during anterograde transport (left) and the average number of direction changes (right). Mean ± SEM, seven independent cultures, AIS (nonAcD) *n* = 42 cells, AIS (AcD) *n* = 42 cells, Stem D *n* = 28 cells, Regular D *n* = 27 cells. **(G–I)** Average number of pause and pausing duration of EGFP-Rab3A vesicles travelling within the indicated region (regardless of direction). Average number of pauses per minute (G), average duration of each pause (H) and total pausing duration of EGFP-Rab3A vesicles during anterograde trafficking (I). Mean ± SEM, seven independent cultures, AIS (nonAcD) *n* = 42 cells, AIS (AcD) *n* = 42 cells, Stem D *n* = 28 cells, Regular D *n* = 27 cells. One-way ANOVA with Tukey's multiple comparisons test, no significance (n.s.) P > 0.05, ****P < 0.0001.

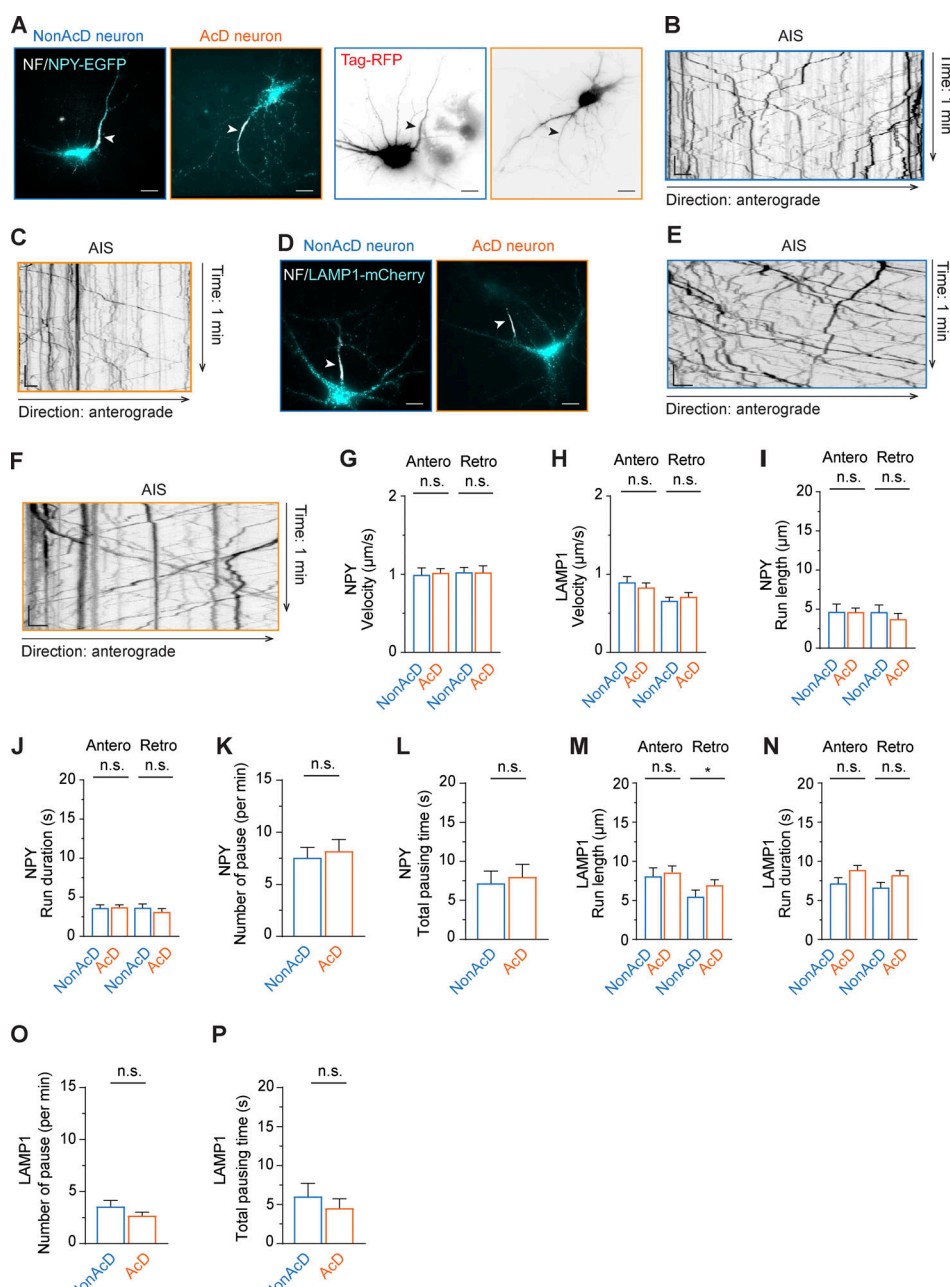

**Figure S3.** **Trafficking of LAMP1 and NPY at the AIS of nonAcD and AcD neurons. (A)** Representative images of nonAcD and AcD neurons co-transfected with NPY-EGFP (left) to visualize dense core vesicles as additional axonal cargo, and Tag-RFP (right) to visualize neuronal morphology. The AIS is live-labeled with anti-NF-CF640R (NF). White arrowhead on the left panel and black arrowhead on the right panel indicates the AIS. Scale bar is 15 µm. Related to Video 8. **(B and C)** Representative kymographs showing trajectories of NPY-EGFP vesicles entering the AIS of nonAcD (B) and AcD (C) neuron shown in A. Scale is 10 s (vertical) and 2 µm (horizontal). **(D)** Representative images of nonAcD and AcD neurons transfected with LAMP1-mCherry to visualize lysosomes as neutral cargo. The AIS is live-labelled with anti-NF-CF640R (NF). White arrowhead indicates the AIS. Scale bar is 15 µm. Related to Video 9. **(E and F)** Representative kymographs showing trajectories of LAMP1-mCherry vesicles entering the AIS of nonAcD (E) and AcD (F) neuron shown in D. Scale is 10 s (vertical) and 2 µm (horizontal). **(G)** Velocity of NPY-EGFP vesicles running towards anterograde and retrograde directions in the AIS of nonAcD and AcD neurons. Mean ± SEM, four independent cultures, nonAcD neurons: n = 12 cells, AcD neurons: n = 12 cells. **(H)** Velocity of LAMP1-mCherry vesicles running towards anterograde and retrograde directions in the AIS of nonAcD and AcD neurons. Mean ± SEM, four independent cultures, nonAcD neurons: n = 20 cells, AcD neurons: n = 17 cells. **(I and J)** Run length (I) and run duration (J) of NPY-EGFP vesicles running towards anterograde and retrograde directions in the AIS of nonAcD and AcD neurons. Mean ± SEM, four independent cultures, nonAcD neurons: n = 12 cells, AcD neurons: n = 12 cells. **(K and L)** Number of pause and total pausing duration of NPY-EGFP vesicles travelling within the AIS of nonAcD and AcD neurons (regardless of directions). Mean ± SEM, four independent cultures, nonAcD neurons: n = 12 cells, AcD: n = 12 cells. **(M and N)** Run length (M) and run duration (N) of LAMP1-mCherry vesicles running towards anterograde and retrograde directions in the AIS of nonAcD and AcD neurons. Mean ± SEM, four independent cultures, nonAcD neurons: n = 20 cells, AcD neurons: n = 17 cells. **(O and P)** Number of pause (O) and total pausing duration (P) of LAMP1-mCherry vesicles travelling within the AIS of nonAcD and AcD neurons (regardless of directions). Mean ± SEM, four independent cultures, nonAcD neurons: n = 20 cells, AcD neurons: n = 17 cells. Mann–Whitney test (two-sided): not significant (n.s.) P > 0.05, *P < 0.05.

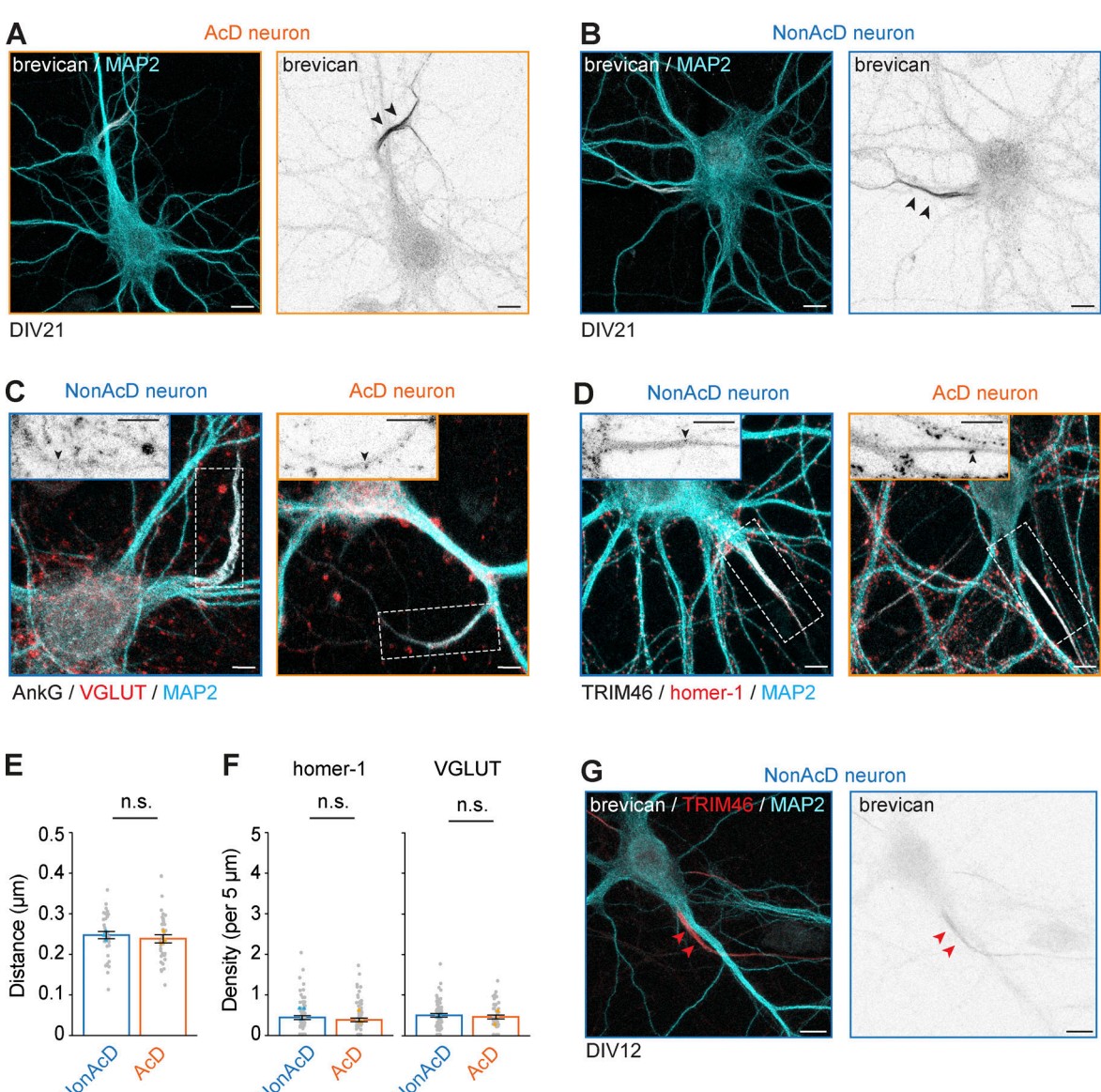

Figure S4.  **Specific ECM and excitatory synapses at the AIS of AcD and nonAcD neurons. (A and B)** Representative images of AIS-specific ECMs labeled by brevican staining at the AIS of DIV21 nonAcD and AcD neurons. Black arrowheads indicate brevican at the AIS. Scale bar is 10 µm. **(C and D)** Representative images of nonAcD and AcD neurons stained with markers for pre- and postsynaptic compartments of an excitatory synapse (VGLUT and homer-1, respectively), for AIS (TRIM46 or AnkG), and for somato-dendritic compartment (MAP2). **(C)** VGLUT **(D)** homer-1. Scale bar is 10 µm. Inset: corresponding zoom-ins of white dashed rectangular. Scale bar is 5 µm. Black arrowheads indicate corresponding markers for pre- and postexcitatory synapse. **(E)** Distance between co-localized gephyrin and VGAT clusters, related to Fig. 6 F. Mean ± SEM, three independent cultures, nonAcD: *n* = 34 cells, AcD: *n* = 32 cells. Grey dots indicate value of individual cell. Orange and cyan triangle indicate mean of each independent culture. **(F)** Number of homer-1 (left) and VGLUT (right) puncta per 5 µm at the AIS of nonAcD and AcD neurons. Mean ± SEM, three independent cultures for Homer-1, four independent experiments for VGLUT, Homer-1: *n* (nonAcD) = 74 cells, *n* (AcD) = 78 cells, VGLUT: *n* (nonAcD) = 67 cells, *n* (AcD) = 50 cells. Grey dots indicate value of individual cell. Orange and cyan triangle indicate mean of each independent culture. **(G)** Representative images of AIS-specific ECMs labeled by brevican staining at the AIS of DIV12 nonAcD. Red arrowheads indicate brevican at the AIS. Scale bar is 10 µm. Mann–Whitney test (two-sided): not significant (n.s.) P > 0.05.

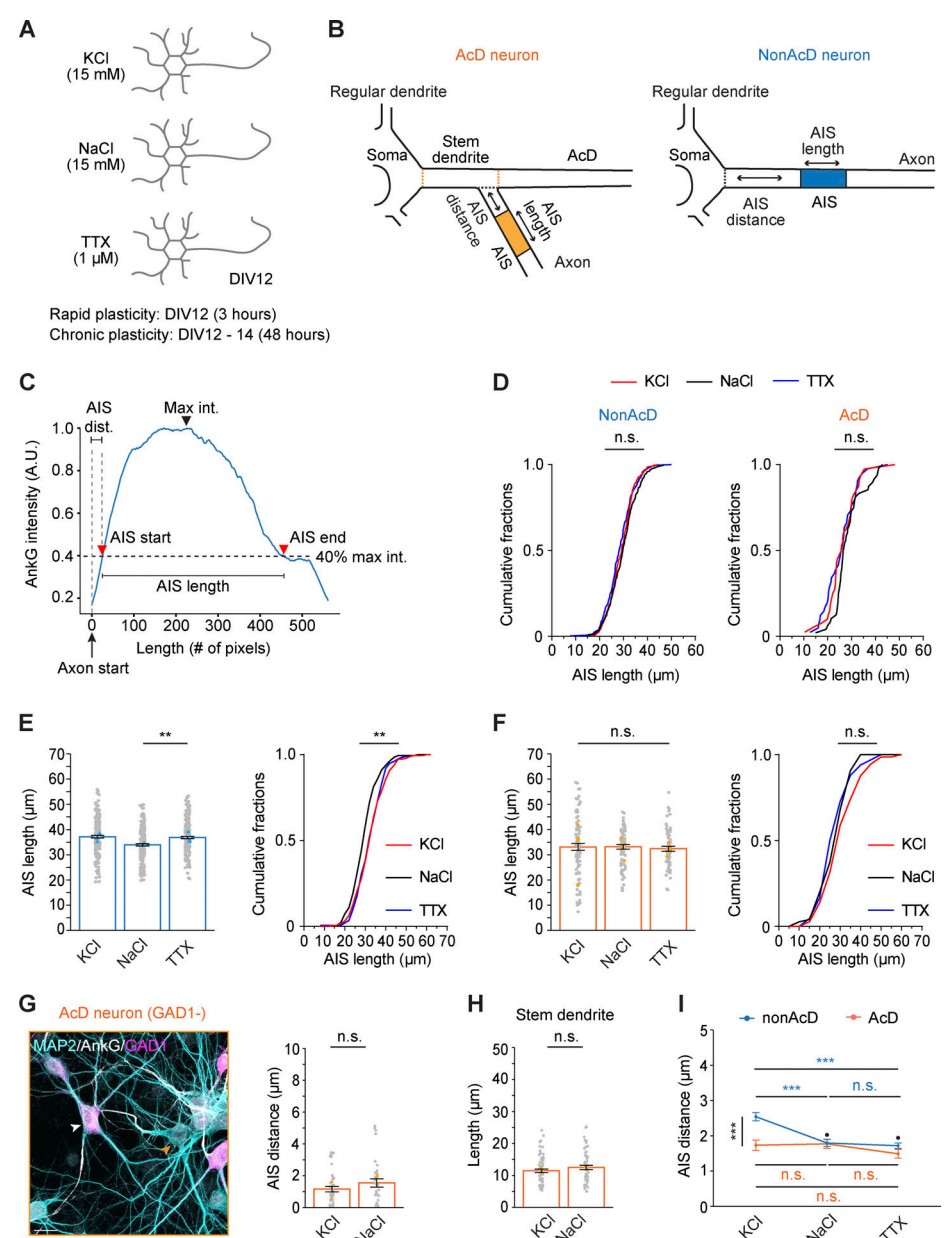

**Figure S5. Definition of AIS length and distance from the cell body or axon carrying dendrite and illustration of the workflow to induce AIS plasticity. (A)** Schematic of AIS short-term and chronic plasticity experiment. **(B)** Definition of AIS length and AIS distance in nonAcD and AcD neurons as read out of AIS plasticity. Orange dashed line indicates the border of stem dendrite of AcD neurons; black dashed line indicates the border of axon origin. **(C)** Illustration of intensity-based AIS length and AIS distance (AIS dist.) measurement. **(D)** Cumulative of AIS length in nonAcD and AcD neuron upon induction of short-term plasticity. Mean ± SEM, three independent cultures, nonAcD: $n$ (KCl) = 295 cells, $n$ (NaCl) = 261 cells, $n$ (TTX) = 303 cells, AcD: $n$ (KCl) = 47 cells, $n$ (NaCl) = 38 cells, $n$ (TTX) = 41 cells. One-way ANOVA with Tukey's multiple comparisons test, no significance (n.s.) P > 0.05. **(E and F)** Measurement of AIS length in nonAcD (E) and AcD (F) neurons upon induction of chronic plasticity. Mean ± SEM, four independent cultures, nonAcD: $n$ (KCl) = 185 cells, $n$ (NaCl) = 186 cells, $n$ (TTX) = 199 cells, AcD: $n$ (KCl) = 86 cells, $n$ (NaCl) = 75 cells, $n$ (TTX) = 65 cells. Grey dot indicates number of individual cells, cyan (E) and orange (F) triangle indicates mean of each experiment. One-way ANOVA with Tukey's multiple comparisons test, no significance (n.s.) P > 0.05, *P < 0.05, ***P < 0.001. **(G)** Left panel: representative image of GAD1 negative (GAD1−) AcD neuron and GAD1 positive inhibitory interneuron. White arrowhead indicates GAD1 positive interneuron, orange arrowhead indicates GAD1 negative (GAD1−) AcD neuron. Scale bar is 20 μm. Right panel: Measurement of AIS distance in GAD1 negative AcD neurons upon chronic plasticity induction. Mean ± SEM, four independent cultures, KCl: $n$ = 42 cells, NaCl: $n$ = 34 cells. Grey dot indicates number of individual cells, orange triangle indicates mean of each experiment. Mann–Whitney test (two-sided): not significant (n.s.) P > 0.05. **(H)** Measurement of stem dendrite length in AcD neuron upon induction of chronic plasticity. Mean ± SEM, three independent cultures, AcD: $n$ (KCl) = 66 cells, $n$ (NaCl) = 55 cells. Grey dot indicates number of individual cells, orange triangle indicates mean of each experiment. Mann–Whitney test (two-sided), no significance (n.s.) P > 0.05. **(I)** Two-way ANOVA (type II) analysis of nonAcD and AcD neurons treated with KCl for 48 h to induce chronic plasticity, NaCl and TTX as control. Mean ± SEM, nonAcD neuron: three independent cultures, $n$ (KCl) = 174 cells, $n$ (NaCl) = 184 cells, $n$ (TTX) = 192 cells, AcD neuron: four independent cultures, $n$ (KCl) = 82 cells, $n$ (NaCl) = 72 cells, $n$ (TTX) = 63 cells. Blue * and "n.s." indicates significance for nonAcD neurons between treatment, dark orange * and "n.s." indicates significance for AcD neurons between treatment, black * and "." Indicates significance of interaction between morphology and treatment, no significance (n.s. or ".") P > 0.05, ***P < 0.001.

Video 1.  **Differentiation of nonAcD neurons.** Related to Fig. 1 C (top row). White arrowhead indicates the axon. Frame rate of this video is 15 frames/s. Scale bar is 50 μm.

Video 2.  **AcD neurons develop AcD via collateralization of precursor neurite.** Related to Fig. 1 C (bottom row) and Fig. S1 G (top row). Orange arrowhead indicates the precursor neurite. White arrowhead indicates the axon. Yellow and pink arrowhead indicates the AcD and stem dendrite, respectively. White arrow indicates the AIS region. Black arrow indicates the cell body of AcD neuron being recorded. Frame rate of this video is 15 frames/s. Scale bar is 50 μm.

Video 3.  **AcD neuron develop axon from a collateral at the precursor neurite (A) and AcD neuron develop axon and AcD via bifurcation (B) of the precursor neurite.** Related to Fig. S1 G (bottom row) and Fig. S1 H, respectively. Orange arrowhead indicates the precursor neurite. White arrowhead indicates the axon. Yellow and pink arrowhead indicates the AcD and stem dendrite, respectively. White arrow in Video 3 A indicates the AIS region. Black arrow indicates the cell body of AcD neuron being recorded. Frame rate of both videos is 15 frames/s. Scale bar is 50 μm.

Video 4.  **MT orientation in the axon, stem dendrite and AcD (A) and in the AIS region and somatic dendrite (B) of an AcD neuron.** Related to Fig. 2 A (bottom row) and Fig. 2 B. **(A)** Black arrow indicates the axon. Pink arrowhead indicates the stem dendrite. Yellow arrowhead indicates the AcD. **(B)** Black arrowhead indicates the AIS. Cyan arrowhead indicates the regular somatic dendrite (Regular D). Frame rate of this video is 30 frames/s. Scale bar is 15 μm.

Video 5.  **MT orientation in nonAcD neuron.** Related to Fig. 2 A (top row). Black arrow indicates the axon. Black arrowhead indicates the AIS. Cyan arrowhead indicates the regular somatic dendrite (Regular D). Frame rate of this video is 30 frames/s. Scale bar is 15 μm.

Video 6.  **EGFP-Rab3A trafficking in AcD neuron (A) and nonAcD neuron (B).** Related to Fig. 4, A–C. Black arrowhead indicates the AIS. Pink arrowhead indicates the stem dendrite. Cyan arrowhead indicates the regular somatic dendrite (Regular D). Frame rate of this video is 30 frames/s. Scale bar is 10 μm.

Video 7.  **TfR trafficking in AcD neuron (A) and nonAcD neuron (B).** Related to Fig. 4, D–F. Black arrowheads indicate the AIS. Pink arrowhead indicates the stem dendrite. Yellow arrowhead indicates the axon carrying dendrite (AcD). Cyan arrowhead indicates the regular somatic dendrite (Regular D). Black arrow indicates the axon. Frame rate of this video is 30 frames/s. Scale bar is 15 μm.

Video 8.  **NPY-EGFP trafficking in nonAcD neuron (A) and AcD neuron (B).** Related to Fig. S3, A–C. Black arrowheads indicate the AIS. Pink arrowhead indicates the stem dendrite. Frame rate of this video is 30 frames/s. Scale bar is 10 μm.

Video 9.  **LAMP1-mCherry trafficking in AcD neuron (A) and nonAcD neuron (B).** Related to Fig. S3, D–F. Black arrowheads indicate the AIS. Pink arrowhead indicates the stem dendrite. Frame rate of this video is 30 frames/s. Scale bar is 10 μm.

Video 10.  **3D reconstruction of the AIS region in AcD neuron (A) and nonAcD neuron (B) labeled with MAP2 (grey), TRIM46 (cyan), gephyrin (yellow) for postinhibitory synapse, and VGAT (magenta) for preinhibitory synapse.** Related to Fig. 6. Frame rate of this video is 15 frames/s. Scale bar is 5 μm.

**Provided online are Table S1 and Data S1. Table S1 shows reagents and resources. Data S1 contains detailed information for statistics.**

