## [Peer Review File · The Journal of Cell Biology]

Unveiling the cell biology of hippocampal neurons with dendritic axon origin

Yuhao Han, Daniela Hacker, Bronte Donders, Christopher Parperis, Roland Thuenauer, Christophe Leterrier, Kay Grünewald, and Marina Mikhaylova

Corresponding Author(s): Marina Mikhaylova, Humboldt-Universität zu Berlin

Review Timeline:

Submission Date:	2024-03-26
Editorial Decision:	2024-04-01
Revision Received:	2024-08-01
Editorial Decision:	2024-08-30
Revision Received:	2024-09-19

Monitoring Editor: Cassandra Ori-McKenney

Scientific Editor: Andrea Marat

Transaction Report:

DOI: <https://doi.org/10.1083/jcb.202403141>

Revision 0

Review #1 - Both, Martin (mboth@physiologie.uni-heidelberg.de)

1. Evidence, reproducibility and clarity:

Evidence, reproducibility and clarity (Required)

****Summary:****

The manuscript by Han et al. describes the structural and functional differences between pyramidal cells in which the axon emanates from a basal dendrite (axon-carrying dendrite cell, AcD cell) and cells with a 'canonical', i.e. somatic origin of the axon (nonAcD cells). They investigate how pyramidal neurons develop into AcD or nonAcD cells during cell development and characterize the cytoskeletal architecture in the two cell classes. Additionally, they examine whether and how axon initial segments, the most important structure for action potential generation, change upon varying activity of the neuron.

The major claims of the paper are:

i) The formation into an AcD or nonAcD cell is intrinsically encoded by a developmental program.

ii) The cytoskeletal structure of AcD and nonAcD cells is similar. However, the stem dendrite inherent only to AcD cells is structurally more similar to an axon than to a dendrite

iii) Axon initial segments of AcD cells contain less cisternal organelles and show less homeostatic plasticity

The authors make use of primary cell cultures from rat hippocampus which are a standard model to investigate developmental questions of single cells and neuronal networks. The manuscript is well structured and in general reads well and the data and conclusions are convincing. I have only a few major comments.

****Major comments:****

The authors cite that acetylated and tyrosinated microtubules have different spatial and compartmental distribution in dendrites and axons and investigate the distribution in the AIS of nonAcD cells and AcD cells, as well as the stem dendrites. However, they just show one example of two different cells (Figure 2D and E) without any statistical analysis. Either, they should remove this part or provide a thorough quantification.

The authors use EGFP-Rab3A vesicle to investigate anterograde transport at the axon and dendrites. They find a slightly faster transport of these vesicles at the AIS of AcD cells and conclude the axonal cargos in general are transported faster across the AIS in AcD cells. In my opinion, this generalization based on one type of vesicle is too far-fetched.

As stated above, the manuscript is well structured and generally reads well. However, throughout the text there are always small mistakes that should be corrected by careful proofreading.

Examples are

Page 6, last paragraph: ...AcD neurons generated [a] collateral...

Page 24, last paragraph: ... line was then drew [drawn] along ...

Page 24, last paragraph: Neurons with ... was consider [were considered] as ...

Page 25, first paragraph: Antibodies ... was [were]

Page 41, (E) Percentage of AcD neurons [that] generate [a] collateral or bifurcate

****Minor comments:****

In the introduction, the authors describe how synaptic inputs are received at the dendrites and propagated to the soma in the form of membrane depolarizations. They should add 'excitatory' to synaptic inputs or also describe the impact of inhibitory synaptic inputs at the dendrites.

In my opinion, Figure 2 could be presented in a slightly better way. The lower part of panel A better fits to panel B, which is next to the upper part of panel A. I understand that the authors systematically present their data first for nonAcD cells and then for AcD cells. However, in this special case it is a little bit more difficult to read the current figure in that order.

The results displayed in Figure 4 are presented in a slightly confusing order. The authors jump from 4D to 4G, then to 4I and 4E, 4H, 4F. Similarly, 4M and N are addressed before 4O and P to finally get to 4K and L. It would be beneficial to present and address the data in a stringent way.

2. Significance:

Significance (Required)

General assessment:

This study addresses a very important and timely question about structural and functional cell diversity of cortical pyramidal neurons. The specific function of AcD cells is currently mostly unknown, which is astonishing given their abundance of 15-50% of pyramidal neurons in cortical structures.

Advance:

This study presents a significant step forward in comprehending the structural and functional relationship of signal computation in single neurons.

Audience:

The study will be important for a wide readership working on very different levels including cellular, network, and behavioral neuroscience.

3. How much time do you estimate the authors will need to complete the suggested revisions:

Estimated time to Complete Revisions (Required)

(Decision Recommendation)

Less than 1 month

4. Review Commons values the work of reviewers and encourages them to get credit for their work. Select 'Yes' below to register your reviewing activity at Web of Science Reviewer Recognition Service (formerly Publons); note that the content of your review will not be visible on Web of Science.

Yes

Review #2 - Rasband, Matthew N (rasband@bcm.edu)

1. Evidence, reproducibility and clarity:

Evidence, reproducibility and clarity (Required)

The manuscript by Han et al. investigates the properties of AIS along axon carrying dendrites (AcDs). These are enigmatic structures with at present poorly defined features. Han et al. work to further characterize the nature of these AIS. Overall, the data are mostly compelling and the reveal that AIS at AcDs are mostly like those of AIS arising from the cell body. Many features were examined and shown not to differ. There were a few instances where the authors claim differences, and this reviewer is not convinced - see comments below. Overall, I think with bit more careful examination of the main differences this could be a nice descriptive paper reporting features of AIS along AcDs.

****Major questions:****

1. The authors suggest that there is reduced Na⁺ channel density at AcD AIS compared to other AIS arising from the cell body. This is not convincing. Immunostaining for Na⁺ channels is notoriously difficult and sensitive to fixation since the epitopes of the anti-Pan Nav antibodies are highly sensitive to fixation. In addition, this is based on immunofluorescence intensity quantification. Since the mechanism of localization is through binding to AnkG, the authors should also measure other AIS proteins like AnkG, b4 spectrin, and Nfasc. Do these change? If all uniformly change I would be much more inclined to accept the conclusion. If they do not change, it still doesn't rule out the concern about fixation conditions and slight differences in the cultures. The authors indicate there is about a 40% reduction in fluorescence intensity. That is quite large. This big difference should also be confirmed in brain sections.
2. The analysis of inhibitory synapse differences at the AIS are also not compelling - this is a limitation of the culture system. The authors have no control over the density of inhibitory neurons in the culture well. This interaction is not intrinsic to the AcD neuron, but rather a feature of neuron-

neuron interactions which should only be modeled in the animal.

3. Finally, this reviewer is also skeptical of the chronic plasticity changes in AIS 'distance.' The authors claim their results are consistent with prior reports, but they see about a 1.5 um shift. Prior studies (Grubb et al.) report 15-17 um change - a full order of magnitude larger than what is reported here. The authors also show no differences in other previously described changes at the AIS. Together with the other results showing AcD neurons and non- AcD neuron AIS are mostly the same, the conclusion that one behaves differently is not compelling with the tiny shift reported.

4. Finally, the major limitation of this study is that it is performed in vitro. Surprisingly, the authors actually argue this is a feature of their system. While it is true some of the questions can be addressed perfectly well in vitro, many cannot. In the first paragraph of the results the authors state an advantage of their system is that there are no microenvironments to influence the development of the AcDs. I'm afraid I view this as a drawback. The authors suggest this is an opportunity to examine intrinsic mechanisms of development - true, but it also foregoes the opportunity to determine if the outcomes are different from what occurs in vivo. To this point, the authors report that only 15-20% of the population of hippocampal neurons in culture are AcD neurons. But in their introduction they cite other literature indicating 50% of hippocampal neurons in vivo are AcD neurons - this suggests that the environment of the hippocampus in vivo influences whether a neuron becomes an AcD neuron or not.

5. I appreciated the balanced discussion of whether this is a stochastic or genetically programmed process. This could have been emphasized earlier in the results since the authors invoke the concept that "...their development must be driven by genetically encoded factors rather than specific...". The authors have not shown this and cannot show it in this system. Indeed, as stated in point 4 above, I think their data argue against a simple genetic program.

2. Significance:

Significance (Required)

interesting subject, timely as features of AIS are of great interest now - especially as a relatively new form of neuronal plasticity. Highly descriptive paper, but emphasizes in this reviewer's opinion that AcD neurons and non AcD neurons have AIS that are essentially the same.

3. How much time do you estimate the authors will need to complete the suggested revisions:

Estimated time to Complete Revisions (Required)

(Decision Recommendation)

Between 1 and 3 months

4. Review Commons values the work of reviewers and encourages them to get credit for their work. Select 'Yes' below to register your reviewing activity at Web of Science Reviewer Recognition Service (formerly Publons); note that the content of your review will not be visible on Web of Science.

No

Review #3 - Kole, Maarten H.P. (m.kole@nin.knaw.nl)

1. Evidence, reproducibility and clarity:

Evidence, reproducibility and clarity (Required)

In this manuscript Han and colleagues report about structural and functional studies on the development of axons originating from dendrites. Leveraging the primary hippocampal neuron preparation, they investigated fundamental cell biological questions including microtubule organization, cargo transport and the early neurite development. I am impressed by the timelapse movies with live AIS labels providing, to the best of my knowledge, the first glance into the development of an axon emerging from a dendrite. The study is technically very good, a pleasure to read, and the results are well described. While conclusions about structure are well supported by their data, the claims about 'function' are weak and speculative. I have listed some issues and by improving clarity the study could become a valuable resource for the field.

1. The authors classify neurons into axon-carrying dendrite (AcD) and non-AcD neurons by measuring the stem dendrite length ($> 3 \mu\text{m}$). I could not find the validity for this cut-off. The non-AcD neurons in Fig. 6B appear more AcD to this reviewer, and, in addition, other researchers have proposed a third category of 'shared root' neurons (doi: 10.7554/eLife.76101). For purposes of reproducibility and transparency, please provide first a comprehensive overview of the entire population of morphologies (i.e. all cells in control conditions). The distances from the soma could be plotted in histogram (etc.) and authors may want to think about independent supporting evidence for the cut-off to classify AcD and non-AcD neurons.
2. Related to point #1 the primary hippocampal neuron system is excellent for cell biological questions but comes with the drawback of imaginative morphologies including neurons with multiple axons and AISs. It is not mentioned here but literature indicates up to 20% of neurons have two axons (e.g. doi: 10.1007/s12264-017-0169-3, 10.1083/jcb.200707042). How did the authors classify the double axon cells? Since the main hypothesis is the existence of an intrinsic program for AcD neurons (p. 5 top), the two axons from one neuron should develop similarly. The authors can easily test this with the data.
3. Some interpretations about function are not correct and the authors should reconsider these. A role of cisternal organelles on neuronal excitability remains to be demonstrated (and see doi.org/10.1002/cne.21445 showing there is none). In addition, the statement that lower fluorescence intensity of Pan-Nav1 is indicating reduced excitability is flawed. Antibody staining does not scale linearly with voltage-gated sodium channel density and since the AIS of AcD neurons is further from the soma it is most likely smaller in diameter which may account for apparent fluorescent differences. For biophysical reasons (for details I refer to 10.3389/fncel.2019.00570, 10.1016/j.conb.2018.02.016 and 10.7554/eLife.53432) smaller diameter axons will be easier to depolarize by depolarizing voltage-gated channels or excitatory synapses. Finally, in AcD neurons the AIS distance from the soma poses all sorts of interesting cable properties with the soma and the local dendritic membrane and the electrotonic properties alone suffice to make these neurons more excitable.
4. Comparing AcD and non-AcD neurons for AIS plasticity is an excellent idea but the present

statistical design is not suitable for answering this question. The authors should directly compare non-AcD and AcD neurons within a two-way ANOVA design, asking the question whether the independent variable axon type is significantly different and interacts with plasticity. Related points: 'AIS distance' in Figure 7 seems to refer to something else than distance from soma (Figure 1). Please clarify. What were the absolute distances from the soma for the AcD neurons and was this dependent on treatment?

****Minor comments****

At p. 7 is stated that "The percentage of none-AcD forming collaterals at DIV1 is much lower than for AcD neurons" but statistical support is lacking. The conclusion in the next line is that "AcD neurons follow consensus development". That is puzzling given the difference just mentioned before. Please clarify. A study not cited in this manuscript showed distinct dendritic morphologies (doi: 10.1073/pnas.1607548113) and AcD interneurons are different for their axonal arborization (doi: 10.1242/dev.202305). Differences in growth of branch arborization could hint to subtypes. Are the AcD and non-AcD neurons different in their adult morphology? A detailed account of the axonal and dendritic trees would strengthen the data.

Some key references are not included here, and a number of these are mentioned above. In the context of the detailed MT and Rab3A vesicle and cargo transport studies, please acknowledge some of the pioneering work of Alan Peters revealing the ultrastructure of axons emerging from dendrites. See Figs. 5-7 in Peters, Proskauer and Kaiserman-Abramof IR., J Cell Biol 39:604 (1968).

What is the identity of the neurons? It makes a difference if the cells are interneurons or pyramidal neurons, CA1 or CA3-like.

For plasticity experiments the authors uses cells as independent measurements, but this is inflating the power. How many cultures were used?

****Referees cross-commenting****

When reading the other reviews I feel they are constructive and providing sufficient conceptual and technical insight to prepare a revision. Although some concerns are overlapping, with 4 independent review reports perhaps not all issues can be addressed within the estimated time frame of 3 months.

2. Significance:

Significance (Required)

That axons originate from dendrites dates back to the 19th century drawings of Ramón y Cajal but today most textbook and schematic drawings of the neuron still show polarized axons and dendrites both emerging from the soma. Since a few years this specific morphological arrangement begins to receive attention and many fundamental cell biological questions remain to be answered. Leveraging the primary hippocampal neuron preparation, the authors use technically clever

experiments to generate new insight into the microtubule organization, cargo transport and the early neurite development. The live imaging of fluorescent labelled axon initial segments is elegant, and an important conclusion is that the stem process, carrying dendrite and axon, grows at a later stage in development. Limitations of the primary neurons should be discussed, however, and the functional consequences of positioning axons on dendrites are not as simple as described by the authors. The study could become a valuable resource for those working in basic research, providing new technical directions.

3. How much time do you estimate the authors will need to complete the suggested revisions:

Estimated time to Complete Revisions (Required)

(Decision Recommendation)

Between 1 and 3 months

4. Review Commons values the work of reviewers and encourages them to get credit for their work. Select 'Yes' below to register your reviewing activity at Web of Science Reviewer Recognition Service (formerly Publons); note that the content of your review will not be visible on Web of Science.

No

Review #4 - Engelhardt, Maren (maren.engelhardt@jku.at)

1. Evidence, reproducibility and clarity:

Evidence, reproducibility and clarity (Required)

****Summary****

Han and colleagues present cell biological data on the development of axon-carrying dendrite (AcD) neurons - cells, in which the axon emerges from a dendrite and not as is more common, from the soma. This particular morphological feature is not new; Cajal already described AcD neurons and since then, their existence was demonstrated in a number of CNS and PNS neurons and across species. However, more recent work from the rodent hippocampus ex vivo and in vivo has pointed towards an interesting functional consequence of AcD morphology, in that these cells can circumvent perisomatic inhibition during network oscillations. Han and colleagues therefore investigate one of the core questions that arises from these previous studies: How is the AcD configuration achieved during development?

The authors utilize an in vitro model (isolated hippocampal neurons from E18 rat) and investigate core parameters of axonal maturation, especially the cytoskeleton, membrane-associated proteins and intra-axonal calcium stores of the axon initial segment (AIS), where action potentials are generated and which contributes to neuronal polarity at different days in vitro (= maturation states). Using a combination of immunofluorescence, confocal, spinning disk and STED microscopy, and

plasticity protocols, the authors present evidence indicating that during development in vitro, AcD neurons follow an intrinsically encoded developmental program, the AIS in nonAcD and AcD cells has comparable cytoskeletal features and retains cellular polarity in both configurations. Using culture conditions to elicit AIS plasticity, the authors then find differences in that AcD neurons do not seem to undergo AIS plasticity and generally show a reduced number of intra-axonal calcium stores. Also, AcD neurons are shown to have fewer axo-axonic synapses at the AIS than nonAcD neurons.

****Major comments****

This study aims at investigating an important question in AcD biology, and uses an easily-accessible model system (E18 rat derived hippocampal neurons in vitro). In this, the study follows previous work in vitro, and nicely reproduces some data, which is a strength of this current study in my opinion. That said, the system is, by nature, artificial and the emergence of axons in vitro often deviates from data obtained in vivo.

Throughout the manuscript, the authors often draw clear-cut conclusions which require a far more critical reflection of what their model can actually accomplish. Thus, a number of statements are not supported by the data (see below). The presentation of the data in the Supplements needs to reflect data distribution, which they currently do not. Likewise, showing S.E.M. instead of S.D. needs to be looked at critically. Otherwise, the data and methods are presented in such a way that they can be reproduced. The quality of the micrographs and videos is excellent and convey the main messages of the study in a very accessible way. I do not see the need for additional experiments, but would ask the authors to critically look at the following issues:

1. A general limitation of this study is the low N for some critical experiments. In several experiments, individual cells become an N, therefore boosting the power of the analysis when in reality, due to the known heterogeneity of AIS length, position, and general cell morphology in vitro, the aim should be to compare means across animals / preparations, each consisting of a comparable number of individual cells. This is especially important for the analyses of COs, axo-axonic synapses and channel expression at the AIS.
2. Such critical parameters as e.g. synaptic innervation at the AIS are investigated in a way that does not support the clear statements given, e.g. "The AIS of AcD neurons receives fewer inhibitory inputs" (Highlights statement) or "AcD neurons have less inhibitory synapses at the AIS" (header of Fig. 6). The overall number of analyzed cells is low (3 and 4 preparations, respectively and approximately 50-cells for each marker). The combination of a pre- and postsynaptic marker for inhibitory / excitatory neurons is a solid decision, but the analysis is not done based on the close approximation of these markers, in 3D, along an AIS, but rather in maxIPs and without any regard of whether pre-and postsynaptic markers are actually close to each other or not. The expression of these markers alone just points towards the epitopes being expressed, but are they localized to each other in such a manner that they could form bona fida synapses? The methods are not totally clear on the image depth (tile scans with 5 μm in z will not provide the detail of information to resolve synapses, so how did the authors address the subcellular analysis here and for the CO and VGSCs?). And generally, were Nyquist conditions taken into consideration throughout the study?

This can be clarified in text and does not require additional experiments.

2. The chapter on AIS plasticity is certainly an interesting addition to the study, but is a bit superficial, yet reaches strong conclusions ("More importantly, it further indicates that the AIS of AcD neurons is insensitive to activity changes"). This is based on unphysiological concentrations of KCl, and certainly not on network manipulation that truly tests synaptic activity. It also comes back to the 1st point above. A suggestion would be to edit the conclusion.

3. The rationale behind looking at the cisternal organelle (CO) in this study is outlined in the Introduction, where the authors state that "... and is responsible for calcium-handling". What is "calcium-handling" and where is the evidence cited? Furthermore, in the Results, they state that "...both compounds (VGSCs and COs) are critical for the AIS to regulate neuronal excitability". While this is the case for VGSCs, there is no conclusive evidence in the literature whether or not the CO is "critical" for neuronal excitability. In fact, a number of neurons have no CO in the AIS (as much as 50% of all AIS in mouse primary visual cortex for example do not express synpo at the AIS at all, Schlüter et al., 2017). The CO can therefore not be as critical for AP initiation as the authors state. Furthermore, the authors state that "AIS plasticity in excitatory neurons is triggered by calcium signaling". While certainly shown and adequately cited here, other factors (independent of calcium) can also play a role, therefore this statement is a bit absolute and should be edited accordingly.

4. The Introduction ends with the rationale of the study, namely that the authors seek to "...provide a detailed characterization of the AIS, including its structural and functional properties...". Structure is investigated, but function is limited to the barrier function of the AIS. Since the authors provide no electrophysiology that would really dissect AIS function, I suggest to rephrase this part and focus on transport.

5. The Discussion is more a list of future pans than a context to current data. The authors could move some of the new questions they identify into an "outlook" section at the end? Also, again have a critical look at the literature that is cited and which statements are accurate. For example, the 2nd phrase in the Discussion states that it was shown that AcD neurons have a "role in memory consolidation", referenced to Hodapp et al., 2022. However, that paper does not provide direct evidence of such a role for AcD neurons. The statement "Collectively, our data provide new insights into the development of AcD neurons and demonstrate that there are differences in AIS functionality between AcD and nonAcD neurons", is not correct. AIS function was not investigated outside of the axonal barrier, and here, the AcD and nonAcD cells do not differ. Also, although the Discussion is geared towards excitatory / glutamatergic neurons, it has been shown by others that interneurons show an even stronger trend to exhibit AcD morphology (work by the Wahle lab and others). This is not clear from the current text (also compare "...AcD neurons being a different subtype if pyramidal neuron"). Further original publications should be included in the paragraph highlighting patch-clamp recordings (see above). In the same context, the statement "...showed that rapid AID plasticity occurs mainly in hippocampal dentate gyrus cells but not in principal excitatory neurons" is not accurate (see Kim, Kuba, Jamann and others).

Generally, the Introduction and Discussion would benefit from a very clear distinction between studies done in vitro versus those done ex vivo or in vivo. This needs to be stated in the Abstract as well.

Methods: For the imaging of synapses, the CO and VGSCs, it is not clear to me from the methods whether Nyquist conditions were applied to produce data that can support the quantification of

nanoscale structures.

Basing the analysis and interpretation of channel expression on fluorescence intensity profiles is problematic (variance in staining quality from samples to sample, lack of an internal standard). This should be noted in the text.

In the text, the first two references given for "Induction of plasticity" do not reference the correct papers.

Finally, the text is lacking a discussion of limitations of the study, especially from a methodological point of view. In the Abstract/Summary already, the authors could point out that this is a pure in vitro study. Interestingly, to this day, AIS relocation during plasticity events has only been shown in cell culture systems, and not in vivo. Therefore, this needs to be put into context here - the chosen system is great for the type of imaging approach presented here, but may look at a type of AIS plasticity that is not seen in vivo.

****Minor comments****

1. How does intrinsic neuronal activity play into developmental programs in vitro? Electrical activity in maturing neurons is a major part of how networks are shaped, and cells differentiate. This is not genetically encoded per se, but has been shown to be a major driving force of neuronal development in vivo. Is this reflected in the culture setting in any way? And have the authors considered testing early changes in activity patterns in their cultures to see whether AcDs and nonAcDs develop in similar percentages? To clarify, I am not asking for additional experiments.
2. The authors may want to add a bit of a technical discussion on the choice of KCl and TTX as triggers for plasticity, especially at the non-physiological concentrations offered here and elsewhere (15 mM KCl).
3. Some key statements would benefit from citing the appropriate original literature (some examples would be the original work by Kole, Bender and Brette on the role of the AIS in AP initiation; original work by D'Este and Letterier on the dendritic and axonal scaffold using nanoscopy; work by Kim, Kuba and Jamann on AIS plasticity in vitro and in vivo that is critical for a more informed discussion of AIS plasticity here, and others)
4. In the Introduction, the authors word their text explicitly for excitatory neurons. However, AIS plasticity has also been observed in interneurons (work by the Grubb lab for example), and axo-axonic synapses are in fact not all inhibitory - this is an important factor to consider given the embryonic state of the culture material. Does the DIV maturation reflect how axo-axonic synapses "switch" from excitatory to inhibitory in vivo (also see work of the Burrone lab)? Can the conclusions from the paper really be drawn based on this type of system?
4. The second header in Results is not clearly formulated. What is meant by "consensus developmental sequence"?
5. The authors state that "less COs account for higher intrinsic excitability". Why is that the case?
6. Last but not least, some very recent studies on AcD biology (Stevens, Thome, Lehmann, Wahle) is available online also on preprint servers and may provide additional support for the current study.

****Referees cross-commenting****

In my opinion, the comments by the other three reviewers are clear, insightful and supportive of / complementary to my own. There are some strong leads within the revisions that the authors hopefully find helpful in the preparation of their final manuscript. I have no doubt that the final publication will be viewed as an important and significant finding in the field of axon onset biology.

2. Significance:

Significance (Required)

The study by Han and colleagues addresses a timely and relevant question and provides excellent quality imaging data (fixed cells and live imaging), as well as convincing superresolution. The authors also provide a solid methods section that will aid others in repeating these experiments. The central question of how AcD neurons develop is of great interest and the study highlights novel findings especially regarding the detailed analysis of axonal and dendritic transport features in AcD cells. The authors also point out a number of questions that arise from their data and that can provide helpful insight for other researchers in the field.

The study has limitations in that it is a pure in vitro study, and some data are based on a low sample number as well as superficial expression analysis (synapses, channels). A number of conclusions made by the authors are therefore not really supported by the current data. The discussion would benefit from a more detailed analysis of the current literature in the field and needs critical reflection on what the shown data really support in form of a section on "limitations".

The study is of broader interest for numerous research fields (developmental neurobiology, axon biology, AIS biology, neuronal plasticity). Given the focus that AcD neurons have received recently in the field of learning and memory consolidation, this study also provides interesting future questions for researchers with a background in network function and behavior.

Reviewer's expertise: Developmental neurobiology, axonal plasticity, AcD neuron morphology and development, in vivo rodent behavior, human slices, confocal and superresolution microscopy, patch-clamp electrophysiology

3. How much time do you estimate the authors will need to complete the suggested revisions:

Estimated time to Complete Revisions (Required)

(Decision Recommendation)

Between 1 and 3 months

4. Review Commons values the work of reviewers and encourages them to get credit for their work. Select 'Yes' below to register your reviewing activity at Web of Science Reviewer Recognition Service (formerly Publons); note that the content of your review will not be visible on Web of Science.

No

Revision Plan

Manuscript number: RC-2024-02349

Corresponding author(s): Kay Grünewald Marina and Marina Mikhaylova

[The “revision plan” should delineate the revisions that authors intend to carry out in response to the points raised by the referees. It also provides the authors with the opportunity to explain their view of the paper and of the referee reports.]

The document is important for the editors of affiliate journals when they make a first decision on the transferred manuscript. It will also be useful to readers of the reprint and help them to obtain a balanced view of the paper.

*If you wish to submit a full revision, please use our "Full Revision" template. **It is important to use the appropriate template to clearly inform the editors of your intentions.**]*

1. General Statements [optional]

This section is optional. Insert here any general statements you wish to make about the goal of the study or about the reviews.

2. Description of the planned revisions

Insert here a point-by-point reply that explains what revisions, additional experimentations and analyses are planned to address the points raised by the referees.

(Note: the planned work for revision is highlighted in green colour)

Reviewer_01

Major comments:

1. The authors cite that acetylated and tyrosinated microtubules have different spatial and compartmental distribution in dendrites and axons and investigate the distribution in the AIS of nonAcD cells and AcD cells, as well as the stem dendrites. However, they just show one example of two different cells (Figure 2D and E) without any statistical analysis. Either, they should remove this part or provide a thorough quantification.

Reply: The spatial and compartmentalized distribution of stable and dynamic MTs in the dendrites and axons of nonAcD neurons has been extensively studied and reviewed (see Kapitein & Hoogenraad, 2011; Katrukha et al., 2021; Tas et al., 2017 for reference). However, the organization of the MT cytoskeleton in AcD neurons is still unknown. Here, we provide the very first evidence on the distribution of tyrosinated and acetylated MTs in AcD neurons, as well as data on MT orientations. We agree with the reviewer that to make our results on the spatial organization of these post-translational modifications in AcD neurons more complete, we need to provide a more thorough quantification analysis.

Revision Plan

To achieve this, we plan to perform immunostainings on DIV10 neurons using antibodies against tyrosinated (tyr) and acetylated (ac-) tubulin to label dynamic and stable MTs, respectively. Subsequently, we will conduct high-resolution 3D confocal imaging and measure fluorescent intensity to illustrate the abundance and staining patterns of tyr- and ac- MTs in the axons and dendrites of AcD neurons. Since the spatial distribution of tyr- and ac-MTs is distinguishable with confocal microscopy, we will retain STED examples in the figures but conduct new analyses on confocal imaging data. We will measure the total fluorescent intensity of tyr- and ac- MTs in different compartments of AcD neurons and normalize it to the size of the measured area. We will then compare the normalized intensity values between the axons and dendrites of AcD neurons to examine whether there is a specific distribution pattern of stable and dynamic MTs. We will analyse at least 3 independent primary culture preparations with a minimum of 30 cells. Using the same dataset, we will also quantify the percentage of AcD neurons with ac-MTs specifically elongating into the axon compared to AcD.

2. The authors use EGFP-Rab3A vesicle to investigate anterograde transport at the axon and dendrites. They find a slightly faster transport of these vesicles at the AIS of AcD cells and conclude the axonal cargos in general are transported faster across the AIS in AcD cells. In my opinion, this generalization based on one type of vesicle is too farfetched.

Reply: The Rab3A protein is associated with pre-synaptic vesicles that are transported by KIF1A and KIF1B β , members of the kinesin-3 family, towards pre-synaptic buttons (see Guedes-Dias & Holzbaur, 2019; Niwa et al., 2008 for reference). Since KIF1A and KIF1B β are common motor proteins that mediate MT-based transport of different types of vesicles (e.g., synaptic vesicles and dense-core vesicles, see Carabalona et al., 2016; Helmer & Vallee, 2023 for reference), we reasoned that Rab3A should be a representative marker for an axonal cargo. However, this indeed does not rule out whether the faster trafficking effect we saw is specific to presynaptic vesicles, as different types of vesicles tend to recruit different modulators that could lead to different trafficking features.

To address this question, we will perform a live-imaging experiment including two additional organelle marker proteins, Neuropeptide Y (NPY) and Lysosome-associated membrane protein 1 (Lamp1). NPY is transported into the axon via KIF1A and KIF1B β -mediated dense-core vesicles (see Helmer & Vallee, 2023; Lipka et al., 2016 for reference). Lamp1 is associated with lysosomes and a range of endocytic organelles that recruit both kinesin-1 and kinesin-3, and are transported into both axons and dendrites (as reviewed in Cabukusta & Neefjes, 2018). By introducing two additional types of vesicles, we should be able to answer whether AcD neurons, in general, tend to transport cargoes into the axon faster than nonAcD neurons.

Minor comments:

In the introduction, the authors describe how synaptic inputs are received at the dendrites and propagated to the soma in the form of membrane depolarizations. They should add 'excitatory' to synaptic inputs or also describe the impact of inhibitory synaptic inputs at the dendrites.

In my opinion, Figure 2 could be presented in a slightly better way. The lower part of panel A better fits to panel B, which is next to the upper part of panel A. I understand that the authors systematically present their data first for nonAcD cells and then for AcD cells. However, in this special case it is a little bit more difficult to read the current figure in that order. The results

Revision Plan

displayed in Figure 4 are presented in a slightly confusing order. The authors jump from 4D to 4G, then to 4I and 4E, 4H, 4F. Similarly, 4M and N are addressed before 4O and P to finally get to 4K and L. It would be beneficial to present and address the data in a stringent way.

Reply: Thank you for the suggestions on how to improve the data representation in the figures. We will change Figures 2 and 4 and make adjustments in the text upon revision since we also plan to include additional data.

Reviewer_02

Major comments:

1. The authors suggest that there is reduced Na⁺ channel density at AcD AIS compared to other AIS arising from the cell body. This is not convincing. Immunostaining for Na⁺ channels is notoriously difficult and sensitive to fixation since the epitopes of the anti-Pan Nav antibodies are highly sensitive to fixation. In addition, this is based on immunofluorescence intensity quantification. Since the mechanism of localization is through binding to AnkG, the authors should also measure other AIS proteins like AnkG, β 4 spectrin, and Nfasc. Do these change? If all uniformly change I would be much more inclined to accept the conclusion. If they do not change, it still doesn't rule out the concern about fixation conditions and slight differences in the cultures. The authors indicate there is about a 40% reduction in fluorescence intensity. That is quite large. This big difference should also be confirmed in brain sections.

Reply: The potential fixation issue and antibody sensitivity on Na⁺ channel staining are indeed valid considerations, and we are aware of them. However, it should be noted that we used pan-Na⁺ channel antibodies that were previously characterised and widely used in literature (see Solé et al., 2019; Yang et al., 2020 for references). Furthermore, our samples underwent the same fixation and staining protocol, and comparable numbers of AcD and nonAcD neurons were imaged from the same preparation and coverslip for each experiment. Imaging settings were also kept constant. Any loss of Na⁺ channel staining at the AIS due to fixation should affect both neuron types and therefore our conclusion is justified. Nevertheless, the reviewer's point regarding other AIS components is valid and will be investigated further in the revised manuscript.

Following the reviewer's suggestion to further strengthen our conclusion, we will measure the intensity of AnkG, β IV-spectrin, and neurofascin in DIV21 AcD and nonAcD neurons. We will compare a minimum of 3 independent cultures, each containing at least 10 cells of each type per culture.

We agree with the reviewer that confirming observed differences in Na⁺ channel staining using brain slices would be beneficial. However, conducting such experiments presents several challenges. Firstly, one approach could involve immunostaining with antibodies against AIS marker AnkG, in combination with somatodendritic marker MAP2 and pan-Nav. However, this method lacks the advantage of clearly identifying neuronal morphology as seen in dissociated cultures, making the outcome unclear and difficult for analysis and interpretation. Alternatively, the use of Thy1-GFP rats, where a subset of neurons is labelled with GFP, could allow for

Revision Plan

morphological studies. Unfortunately, we do not have access to this rat line, and the process of importing it, obtaining permits, and establishing a colony is beyond the timeframe for manuscript revision. Additionally, while pan-Nav antibodies have shown reliability in dissociated cultures, their efficacy in tissue staining is less certain. We could provide example images upon request. Secondly, endogenously labelling of Na⁺ channels is another option, but remains a significant challenge. Recent developments in endogenous labelling, such as the CRISPR/Cas9-based method using pORANGE by Fréal et al. (Fréal et al., 2023), and the generation of Scn1a-GFP transgenic mice by Yamagata et al. (Yamagata et al., 2023), offer potential solutions. However, the labelling efficiency of pORANGE is uncertain, and both methods are time-consuming and cannot be completed within the three-month revision period.

As an alternative, we propose emphasising that our results are based on *in vitro* experiments and discussing the advantages and limitations of this approach in the discussion section.

2. The analysis of inhibitory synapse differences at the AIS are also not compelling - this is a limitation of the culture system. The authors have no control over the density of inhibitory neurons in the culture well. This interaction is not intrinsic to the AcD neuron, but rather a feature of neuron-neuron interactions which should only be modelled in the animal.

Reply: The reviewer is correct in pointing out that establishing inhibitory synapses at the AIS is not an intrinsic feature of AcD neurons; it depends on the network and should be modelled in animals. We will include this limitation of the cell culture model in the discussion section in the revised manuscript. We also understand the reviewer's concern that the lower amount of inhibitory synapses at AcD neuron AIS might be due to uneven density of inhibitory neurons between cultures. Nonetheless, assuming that the number of inhibitory neurons is constant between preparations, it is an interesting observation that AcD neurons form fewer inhibitory synapses at the AIS. This may be related to the features of the AIS and its morphology and should be further investigated.

To make our study more comprehensive and also address the reviewer's concern regarding the presence of inhibitory neurons, we will perform immunostainings in dissociated cultures (40,000 cells per 18 mm coverslip, same as in experiments with synapse quantification) with antibodies against pCaMKIIa, an excitatory neuron marker, and GAD1, a marker for inhibitory neurons. Then, we will quantify the density of inhibitory neurons in the culture. We will perform measurements from 3-6 independent cultures by analysing large fields of view in different areas of a coverslip (20-30 neurons per area) to determine if the density of inhibitory neurons varies between cultures as well as preparations. Furthermore, as also requested by reviewer 4, we will perform new immunostainings where pre- and post-synaptic markers (VGAT and Gephyrin) will be included in the same sample together with the AIS (AnkG or Neurofascin) and dendritic marker (MAP2). Synapses that contain pre- and post-synaptic components will be analysed and included in the revised version of the manuscript.

4. Finally, the major limitation of this study is that it is performed *in vitro*. Surprisingly, the authors actually argue this is a feature of their system. While it is true some of the questions can be addressed perfectly well *in vitro*, many cannot. In the first paragraph of the results the authors state an advantage of their system is that there are no microenvironments to influence the development of the AcDs. I'm afraid I view this as a drawback. The authors suggest this is an opportunity to examine intrinsic mechanisms of development - true, but it also foregoes the

Revision Plan

opportunity to determine if the outcomes are different from what occurs *in vivo*. To this point, the authors report that only 15-20% of the population of hippocampal neurons in culture are AcD neurons. But in their introduction they cite other literature indicating 50% of hippocampal neurons *in vivo* are AcD neurons - this suggests that the environment of the hippocampus *in vivo* influences whether a neuron becomes an AcD neuron or not.

Reply: The reviewer is right in pointing that the *in vivo* environment could indeed affect AcD neuron development, and we also find this to be a very interesting topic to investigate in the future. Even more intriguingly, as shown in a preprint by Lehmann et al. (doi: <https://doi.org/10.1101/2023.07.31.551236>), network activity stimulates neurons to acquire AcD morphology. While it is true that the impact of the microenvironment on AcD neuron development cannot be studied in dissociated cultures, our *in vitro* data undoubtedly support the fact that hippocampal neurons can intrinsically develop into AcD morphology independent of the *in vivo* environment. As also mentioned in the next point, our statement "...their development must be driven by genetically encoded factors rather than specific..." might sound too definitive and therefore eliminate possible effects from the microenvironment. We will revise this part. Although it is highly desirable to move cell biological studies from neuronal cell cultures to tissue, to date, it is still very challenging to perform many of experiments which we did in this study in slices or living animals due to a lack of appropriate technologies and tools. We are convinced that many basic biological questions can be and should be studied in simplified culturing models because they are truly fundamental, they should also be reproducible in these models.

To address the reviewer's question regarding the percentage difference between our data and the previous study by Thome et al. (2014), several factors should be considered. First, as noted by the reviewer, our results were obtained from an *in vitro* system, which is not directly comparable to the *in vivo* model system used in Thome et al.'s study (Thome et al., 2014). Second, the age of the neurons quantified in our developmental experiments is DIV5 and DIV7. This young age disparity could contribute to the percentage difference, as Thome et al. analyzed neurons from P28-35 adult animals, where 50% of the AcD neuron population was observed, specifically in the CA1 region. Third, it's important to note that in other hippocampal regions, the percentage of AcD neurons is lower (approximately 20-30%). Since our hippocampal primary cultures contain neurons from all hippocampal regions, this may have averaged out our quantification of AcD neuron percentage. Additionally, in the study by Benavides-Piccione et al. (Benavides-Piccione et al., 2020), they reported 20% AcD neurons in the CA1 region of hippocampi isolated from 8-week-old mouse pups, a number similar to what we observed *in vitro*. Interestingly, Thome et al. reported that in P8 pups, AcD neuron population in hippocampal CA1 region is 30%. This number increased to 50% in adult animals at age of P28-35, suggesting there is perhaps an age dependent increase of AcD neuron population. This could be an additional reason of why we only saw 15-20% of AcD neurons in our *in vitro* system, regardless of the *in vivo* environment.

In the revised version, we will clarify these points in the introduction and discussion sections. Additionally, we will quantify the proportion of AcD neurons in mature DIV21 dissociated hippocampal cultures and compare it to DIV7 cultures to assess whether there is an increase in the AcD population over time. We believe that this experiment, combined with the explanations provided above, will sufficiently address the reviewer's question. However, it is important to

Revision Plan

acknowledge that the establishment of neuronal networks *in vitro* differ from those *in vivo*. Therefore, there may be potential differences in the outcomes.

5. I appreciated the balanced discussion of whether this is a stochastic or genetically programmed process. This could have been emphasized earlier in the results since the authors invoke the concept that "...their development must be driven by genetically encoded factors rather than specific...". The authors have not shown this and cannot show it in this system. Indeed, as stated in point 4 above, I think their data argue against a simple genetic program.

Reply: As suggested by the reviewer and noted in point 4, we will revise the section on AcD neuron development in our manuscript to emphasize that hippocampal neurons may adopt AcD morphology through genetic or stochastic mechanisms. While we acknowledge that environmental and activity factors may also influence this process, particularly in mature neurons, our study focuses on developing neurons where genetic and stochastic factors are likely to be predominant. This conclusion is supported by the observation that neurons develop into AcD morphology *in vitro*, where environmental and activity patterns do not mimic those of *in vivo* systems.

Indeed, our current manuscript does not explore genetic factors involved in AcD neuron development. To address this question, one approach could be to label AIS markers endogenously in dissociated cultures using the PORANGE method (see Willems et al., 2020 for reference) or utilize AnkG-GFP transgenic mice (Fréal et al., 2023; Thome et al., 2023) along with a volume marker like mRuby or GFP. This would allow for the identification of AcD and nonAcD neurons *in vivo* and *in vitro*, followed by single-cell transcriptomics analysis to uncover potential genetic factors. Subsequently, candidate genes could be manipulated to demonstrate their essential role in AcD neuron development. However, such experiments require significant time and resources beyond the scope of our current revision timeframe. Nonetheless, this question presents an exciting direction for future research.

Reviewer 3

Major comments:

1. The authors classify neurons into axon-carrying dendrite (AcD) and non-AcD neurons by measuring the stem dendrite length ($> 3 \mu\text{m}$). I could not find the validity for this cut-off. The non-AcD neurons in Fig. 6B appear more AcD to this reviewer, and, in addition, other researchers have proposed a third category of 'shared root' neurons (doi: 10.7554/eLife.76101). For purposes of reproducibility and transparency, please provide first a comprehensive overview of the entire population of morphologies (i.e. all cells in control conditions). The distances from the soma could be plotted in histogram (etc.) and authors may want to think about independent supporting evidence for the cut-off to classify AcD and non-AcD neurons.

Reply: Concerning the validity of AcD neuron classification, we did measure the length of the stem dendrite, as shown in Figure S4G, with an average distance of around $10 \mu\text{m}$. However, we admit that this information is presented relatively late in the manuscript. To address the reviewer's criticism, in the revised version, we will include a supplementary figure displaying a gallery of representative images of both AcD and nonAcD neurons analyzed in our study (please refer to

Revision Plan

Hodapp et al., 2022; Fig S1 C&D; Fig S3 as an example). Given the sample size of AcD and nonAcD neurons in our study, including all images would result in a very large figure (for example, Figure 1: DIV5: 83 AcD neurons out of 427 cells, DIV7: 47 AcD neurons out of 387 cells). We will only show representative examples of AcD neurons in the gallery. Additionally, as suggested, we will plot the length of the stem dendrite (or axon distance) of AcD neurons as a histogram to demonstrate that the AcD neurons included in our study indeed have a stem dendrite longer than 3 μm . To further validate the used classification method, we will measure the diameter of the stem dendrite in all analyzed AcD neurons and then compare the distance between the soma and the start of the axon in each analyzed AcD neuron to the diameter of its stem dendrite. As described by Hodapp et al. (Hodapp et al., 2022; Fig S1A), AcD neurons are expected to have a stem dendrite longer than their diameter.

We have considered having independent evidence to support the classification of nonAcD and AcD neurons. However, the method used by Thome et al. and Wahle et al. for AcD and nonAcD neuron classification is well established and widely accepted (see Thome et al., 2014; Wahle et al., 2022 for references). Similar standards were also employed by Benavides-Piccione et al. (Benavides-Piccione et al., 2020). Introducing independent evidence could potentially raise further doubts, so we have chosen to maintain consistency with previous studies.

As for the "shared root" neurons described by Wahle et al., we did not analyze this category separately and included them in the nonAcD subtype. Nonetheless, it is an interesting direction to explore in the future. For completeness, we will discuss this point in the revised manuscript.

2. Related to point #1 the primary hippocampal neuron system is excellent for cell biological questions but comes with the drawback of imaginative morphologies including neurons with multiple axons and AISs. It is not mentioned here but literature indicates up to 20% of neurons have two axons (e.g. doi: 10.1007/s12264-017-0169-3, 10.1083/jcb.200707042). How did the authors classify the double axon cells? Since the main hypothesis is the existence of an intrinsic program for AcD neurons (p. 5 top), the two axons from one neuron should develop similarly. The authors can easily test this with the data.

Reply: We appreciate the reviewer's comment regarding the choice of the model system for this type of study. Indeed, as they pointed out, in primary cultures, some neurons develop more than one axon. Since we did not find any supporting evidence from the literature reporting that hippocampal neurons have multiple axons *in vivo*, we only analyzed neurons with one axon for both AcD and nonAcD neurons. We will clarify this in our method section of the revised manuscript.

3. Some interpretations about function are not correct and the authors should reconsider these. A role of cisternal organelles on neuronal excitability remains to be demonstrated (and see doi.org/10.1002/cne.21445 showing there is none). In addition, the statement that lower fluorescence intensity of Pan-Nav1 is indicating reduced excitability is flawed. Antibody staining does not scale linearly with voltage-gated sodium channel density and since the AIS of AcD neurons is further from the soma it is most likely smaller in diameter which may account for apparent fluorescent differences. For biophysical reasons (for details I refer to 10.3389/fncel.2019.00570, 10.1016/j.conb.2018.02.016 and 10.7554/eLife.53432) smaller diameter axons will be easier to depolarize by depolarizing voltage-gated channels or excitatory

Revision Plan

synapses. Finally, in AcD neurons the AIS distance from the soma poses all sorts of interesting cable properties with the soma and the local dendritic membrane and the electrotonic properties alone suffice to make these neurons more excitable.

Reply: The reviewer brings up very valid and important points that we will address in the revised manuscript. First, we will rephrase and adjust our interpretations regarding the functions of the cisternal organelle in the AIS. As also mentioned by reviewer #2, we are aware that antibody staining does not properly reflect Na⁺ channel density. As discussed above, we will also measure other AIS proteins that anchor Na⁺ channels to see if there are any correlations in fluorescence intensity between them and Nav1. We agree with the reviewer that AcD neuron's AIS could have a smaller diameter, resulting in fewer Na⁺ channels. Indirect evidence is already available in the study of Benavides-Piccione et al., showing a smaller axon diameter in AcD neurons compared to nonAcD neurons in both human and mouse brain sections (Figure S4). To test this in our model system, we propose to measure the AIS diameter in AcD neurons. If this is indeed the case, we will indicate it in our revised manuscript and edit the section on Na⁺ channels.

Exploring the biophysical properties of the AIS and axons of AcD neurons is indeed a highly interesting direction to pursue and is the project in its own. It would necessitate the use of computational modeling approaches, which require considerable time and resources that are not feasible within the timeframe of this revision.

4. Comparing AcD and non-AcD neurons for AIS plasticity is an excellent idea but the present statistical design is not suitable for answering this question. The authors should directly compare non-AcD and AcD neurons within a two-way ANOVA design, asking the question whether the independent variable axon type is significantly different and interacts with plasticity.

Related points: 'AIS distance' in Figure 7 seems to refer to something else than distance from soma (Figure 1). Please clarify. What were the absolute distances from the soma for the AcD neurons and was this dependent on treatment?

Reply: We appreciate reviewer's comment and in the revised version we will perform the analysis using two-way ANOVA.

Regarding the terminology and definitions used in our manuscript, the "AIS distance" refers to the measurement between the start of the AIS and the axon initiating point, as depicted in Figure S4 of the manuscript. We adopted this parameter from the previous study by Grubb et al. (Grubb & Burrone, 2010), ensuring consistency in our investigation of AIS plasticity. For AcD neurons, where the axon branches out from the dendrite, we defined the AIS distance as the length between the start of the AIS and the border of the stem dendrite, as illustrated in Figure S4B.

In Figure 1, the term "distance from soma" represents the length of stem dendrite and used for AcD and nonAcD neuron classification. As shown in Figure S4G, the absolute distance from the soma for AcD neurons is approximately 10 μm and remains consistent across treatments. We will explain these points more clearly in the revised manuscript.

Minor comments:

1. At p. 7 is stated that "The percentage of none-AcD forming collaterals at DIV1 is much lower than for AcD neurons" but statistical support is lacking. The conclusion in the next line is that "AcD

Revision Plan

neurons follow consensus development". That is puzzling given the difference just mentioned before. Please clarify.

Reply: We will provide statistical support for comparing collateral formation between nonAcD and AcD neurons at DIV1.

Regarding the second point concerning consensus development, we were referring to the general developmental sequence of AcD neurons, as described by Dotti et al. (see Dotti et al., 1988 for reference), where neurons typically first establish an axon and then dendrites. This sequence is not necessary related to collateral formation, which indeed differs between nonAcD and AcD neurons. The ability to form collaterals may come from local differences in microtubule (MT) and actin dynamics at AcD neuron precursor axons, but it does not alter the fact that AcD neurons initially establish an axon and subsequently dendrites. We will clarify it in the revised manuscript.

2. A study not cited in this manuscript showed distinct dendritic morphologies (doi: 10.1073/pnas.1607548113) and AcD interneurons are different for their axonal arborization (doi: 10.1242/dev.202305). Differences in growth of branch arborization could hint to subtypes. Are the AcD and non-AcD neurons different in their adult morphology? A detailed account of the axonal and dendritic trees would strengthen the data.

Reply: Thank you for pointing this out. We will include this citation. In the study by Hodapp et al., it was shown that AcD and nonAcD neurons exhibit similar dendritic morphology and do not differ in spine density, number of dendritic branches, and total dendritic length. However, in hippocampal AcD neurons, the AcD occupies 35% of the total basal dendrite length, which is larger than basal dendrites in nonAcD neurons, suggesting that AcD neurons do possess specific features in their dendritic trees.

Regarding the axons of AcD neurons, there is currently no detailed study available, and it would be more appropriate to investigate neuronal connectivity through tracing studies in animals rather than in primary cultures. Therefore, this question falls outside the scope of the current manuscript.

3. Some key references are not included here, and a number of these are mentioned above. In the context of the detailed MT and Rab3A vesicle and cargo transport studies, please acknowledge some of the pioneering work of Alan Peters revealing the ultrastructure of axons emerging from dendrites. See Figs. 5-7 in Peters, Proskauer and Kaiserman-Abramof IR., J Cell Biol 39:604 (1968). What is the identity of the neurons? It makes a difference if the cells are interneurons or pyramidal neurons, CA1 or CA3-like. For plasticity experiments the authors uses cells as independent measurements, but this is inflating the power. How many cultures were used?

Reply: Thank you for pointing this out; we will include the suggested references in the revised manuscript. In our study, we focused on excitatory neurons from the hippocampus. We distinguished neuron types morphologically or with the inhibitory neuron marker GAD1. Identifying CA1, CA2, CA3, and DG subtypes in dissociated culture is more challenging, and this would be an interesting avenue to explore in an in vivo system. Here, we focused on fundamental cell biology aspects related to the AIS structure and its trafficking barrier function, which should be similar in all these neuron types. While there may be subtype-specific differences in AIS plasticity, investigating this is beyond the scope of our manuscript.

Revision Plan

For the plasticity experiments, we used a total of 3 independent cultures, from which we collected a comparable number of neurons. In response to the reviewer's concern, we will also plot the mean of each culture to illustrate the variability of our data points.

Reviewer 4

Major comments:

1. A general limitation of this study is the low N for some critical experiments. In several experiments, individual cells become an N, therefore boosting the power of the analysis when in reality, due to the known heterogeneity of AIS length, position, and general cell morphology in vitro, the aim should be to compare means across animals / preparations, each consisting of a comparable number of individual cells. This is especially important for the analyses of COs, axo-axonic synapses and channel expression at the AIS.

Reply: We would like to mention that this is a cell biological study where neurons are grown in dissociated cultures. To prepare one such culture, we typically use hippocampi from 6-8 E18 rat embryos, which are then mixed in one suspension before plating. The cells are then plated on coverslips in a 12-well plate format. When referring to replicates, for all experiments except for the longitudinal study of 5-day-long time-lapse imaging of developmental sequences (Figure 1), we used between 3 to 6 independent preparations. From each preparation, we took a comparable number of cells derived from 4-6 different coverslips. For each experiment, we measured more than a hundred cells, which is standard practice in the field. To address the issue with individual measurements, in the revised manuscript, we will additionally plot the means of each independent preparation.

2. Such critical parameters as e.g. synaptic innervation at the AIS are investigated in a way that does not support the clear statements given, e.g. "The AIS of AcD neurons receives fewer inhibitory inputs" (Highlights statement) or "AcD neurons have less inhibitory synapses at the AIS" (header of Fig. 6). The overall number of analyzed cells is low (3 and 4 preparations, respectively and approximately 50-cells for each marker). The combination of a pre- and postsynaptic marker for inhibitory / excitatory neurons is a solid decision, but the analysis is not done based on the close approximation of these markers, in 3D, along an AIS, but rather in maxIPs and without any regard of whether pre-and postsynaptic markers are actually close to each other not. The expression of these markers alone just points towards the epitopes being expressed, but are they localized to each other in such a manner that they could form bona fide synapses? The methods are not totally clear on the image depth (tile scans with 5 μm in z will not provide the detail of information to resolve synapses, so how did the authors address the subcellular analysis here and for the CO and VGSCs?). And generally, were Nyquist conditions taken into consideration throughout the study? This can be clarified in text and does not require additional experiments.

Reply: The overall number of cells for quantifying inhibitory synapses along the AIS was approximately 80 cells for each synaptic marker. To clarify this, we will indicate the number of cells in the figure legend of our revised manuscript and will additionally plot mean values across independent preparations.

In the current manuscript, our main goal was to provide an initial quantitative measurement of AIS features in AcD neurons to see if they differ from nonAcD neurons. Hence, maxIPs are sufficient

Revision Plan

for this purpose as they summarize the 3D information. To make our study more comprehensive, following the reviewer's suggestion, we will conduct additional experiments to co-label pre- and post-inhibitory synapses at the AIS with VGAT and gephyrin, respectively. Then, we will image samples in 3D to measure the density as well as the distance between pre- and post-synapses at the AIS of AcD neurons and compare them to nonAcD neurons.

The Nyquist condition was taken into consideration throughout the study. The pixel size of our data collection was 0.081 μm for the laser scanning microscope, as indicated in our methods section. Given the optical setup of our microscope and the fluorophores used to label target proteins (information available in the methods section of our manuscript), the acceptable Nyquist lateral sampling size (or pixel size, in other words) for confocal images is between 0.083 to 0.093 μm and 0.2 μm in the z-plane. In our data collection for laser scanning confocal images, the z-step size was 0.5 μm (see methods section of our manuscript), which is indeed undersampling the data. However, this should not significantly affect our analysis based on maxIPs. The new stainings with matched pre- and post-synaptic markers will be imaged with a smaller z-step (0.2 μm) and then reconstructed in 3D.

2. The chapter on AIS plasticity is certainly an interesting addition to the study, but is a bit superficial, yet reaches strong conclusions ("More importantly, it further indicates that the AIS of AcD neurons is insensitive to activity changes"). This is based on un-physiological concentrations of KCl, and certainly not on network manipulation that truly tests synaptic activity. It also comes back to the 1st point above. A suggestion would be to edit the conclusion.

Reply: KCl treatment globally depolarizes the membrane potential of neurons, leading to an increase in intracellular calcium via voltage-sensitive calcium channels as well as NMDA and AMPA receptors (Rienecker et al., 2020). This protocol has been used in several initial studies describing the plasticity of the AIS (see Evans et al., 2013, 2017; Grubb & Burrone, 2010; Jamann et al., 2021; Muir & Kittler, 2014; Wefelmeyer et al., 2015 for references). Moreover, as shown by Evans et al. and Grubb et al. (see Evans et al., 2013; Grubb & Burrone, 2010 for references), AIS plasticity is not abolished by TTX, which blocks Na⁺ channels, but is prevented by L-type calcium channel blockers. This suggests that the occurrence of AIS plasticity is independent of action potentials but more sensitive to calcium-related pathways downstream of membrane potential depolarization and post-synaptic activation. Hence, we believe our results are indicative of how the AIS would react when calcium signaling pathways are altered by activity levels. To address the reviewer's concern, we will focus our conclusion more on membrane potential depolarization and calcium signalling and edit out statements.

As discussed above in response to reviewer #3, the quantification of AIS plasticity includes 3 independent preparations, comprising approximately 200 neurons in total. To prevent inflation of statistical power in the analysis, we will also plot the means and standard error of the mean (SEM) for each independent experiment and assess whether any differences persist.

3. The rationale behind looking at the cisternal organelle (CO) in this study is outlined in the Introduction, where the authors state that "...and is responsible for calcium handling". What is "calcium-handling" and where is the evidence cited? Furthermore, in the Results, they state that "...both compounds (VGSCs and COs) are critical for the AIS to regulate neuronal excitability".

Revision Plan

While this is the case for VGSCs, there is no conclusive evidence in the literature whether or not the CO is "critical" for neuronal excitability. In fact, a number of neurons have no CO in the AIS (as much as 50% of all AIS in mouse primary visual cortex for example do not express synpo at the AIS at all, Schlüter et al., 2017). The CO can therefore not be as critical for AP initiation as the authors state. Furthermore, the authors state that "AIS plasticity in excitatory neurons is triggered by calcium signaling". While certainly shown and adequately cited here, other factors (independent of calcium) can also play a role, therefore this statement is a bit absolute and should be edited accordingly.

Reply: Thank you for constructive editorial suggestions. Regarding the first question on calcium handling, we were referring to Ca²⁺ storage and release mechanisms. Benedeczyk et al. already showed the existence of SERCA-type Ca²⁺ pumps at the membrane of the cisternal organelle (CO) to demonstrate the involvement of Ca²⁺ sequestering/storage by the CO at the AIS (Benedeczyk et al., 1994). Although indirect, Sánchez-Ponce et al. showed the presence of IP3R, which promotes Ca²⁺ release from internal storage, at the AIS and partially colocalizes with synaptodin (Sánchez-Ponce et al., 2011). This is also the same case for the Ca²⁺-binding protein annexin 6. Together, this evidence indicates a putative role of the CO in regulating Ca²⁺ dynamics (storage/release) at the AIS. Since Ca²⁺ levels have a significant impact on action potential generation and timing at the AIS (see Bender & Trussell, 2009; Yu et al., 2010 for references), and therefore should be strictly regulated, it is likely that the CO at the AIS is important for regulating neuronal excitability by controlling Ca²⁺ dynamics. However, as mentioned by the reviewer, there are no conclusive pieces of evidence showing the relationship between the CO and neuron excitability regulation. We will edit our statement accordingly.

In contrast to the findings of Schlüter et al. (Schlüter et al., 2019), which were conducted in the mouse primary visual cortex, Sánchez-Ponce et al. showed that nearly 90% of hippocampal neurons contain synaptodin, the CO marker protein, at the AIS. Furthermore, Schlüter et al. also demonstrated that in the other 50% of neurons containing COs at the AIS, the COs change size during visual deprivation, and their presence correlates with AIS length changes as well as eye-opening. These observations do suggest that COs are related to neuronal activity. However, this correlation and the formation of COs may be specific to neuro subtypes or require certain triggers. This is another interesting direction to explore, and we will include it in the discussion of the revised manuscript.

Regarding the last point on Ca²⁺ and AIS plasticity, we were not excluding other factors that could potentially participate in AIS plasticity and will also discuss it in the revised version.

4. The Introduction ends with the rationale of the study, namely that the authors seek to "provide a detailed characterization of the AIS, including its structural and functional properties....". Structure is investigated, but function is limited to the barrier function of the AIS. Since the authors provide no electrophysiology that would really dissect AIS function, I suggest to rephrase this part and focus on transport.

Reply: As suggested, we will certainly emphasize the cargo barrier function of the AIS in AcD neurons in our introduction. But we would like to keep the term "AIS function", because it has already been nicely demonstrated electrophysiologically by previous studies that the plasticity effect of the AIS is very important for maintaining cellular homeostasis.

Revision Plan

5. The Discussion is more a list of future plans than a context to current data. The authors could move some of the new questions they identify into an "outlook" section at the end? Also, again have a critical look at the literature that is cited and which statements are accurate.

For example, the 2nd phrase in the Discussion states that it was shown that AcD neurons have a "role in memory consolidation", referenced to Hodapp et al., 2022. However, that paper does not provide direct evidence of such a role for AcD neurons. The statement "Collectively, our data provide new insights into the development of AcD neurons and demonstrate that there are differences in AIS functionality between AcD and nonAcD neurons", is not correct. AIS function was not investigated outside of the axonal barrier, and here, the AcD and nonAcD cells do not differ. Also, although the Discussion is geared towards excitatory / glutamatergic neurons, it has been shown by others that interneurons show an even stronger trend to exhibit AcD morphology (work by the Wahle lab and others). This is not clear from the current text (also compare "...AcD neurons being a different subtype if pyramidal neuron").

Further original publications should be included in the paragraph highlighting patch-clamp recordings (see above). In the same context, the statement "...showed that rapid AID plasticity occurs mainly in hippocampal dentate gyrus cells but not in principal excitatory neurons" is not accurate (see Kim, Kuba, Jamann and others). Generally, the Introduction and Discussion would benefit from a very clear distinction between studies done in vitro versus those done ex vivo or in vivo. This needs to be stated in the Abstract as well.

Methods: For the imaging of synapses, the CO and VGSCs, it is not clear to me from the methods whether Nyquist conditions were applied to produce data that can support the quantification of nanoscale structures. Basing the analysis and interpretation of channel expression on fluorescence intensity profiles is problematic (variance in staining quality from samples to sample, lack of an internal standard). This should be noted in the text. In the text, the first two references given for "Induction of plasticity" do not reference the correct papers.

Reply: Thank you for the valuable suggestions; we will incorporate them into the revised version of the manuscript. The structure will undoubtedly benefit from these improvements. We will also have a further look into our interpretation of the literatures as well as citations during our revision time frame.

Regarding methods, as stated in response to the second point raised by this reviewer, we ensured that the Nyquist condition was adhered to throughout the study. The pixel size, z-step size, and optical setup of the microscopes used were already indicated in our methods section. With respect to Na⁺ channel staining, we were indeed aware of the potential issues posed by the experimental setup, and we will explicitly mention this in our revised manuscript. Additionally, we plan to measure other AIS scaffolding and membrane proteins that anchor Na⁺ channels to assess for potential changes, which could indirectly support our Na⁺ channel staining results.

Finally, the text is lacking a discussion of limitations of the study, especially from a methodological point of view. In the Abstract/Summary already, the authors could point out that this is a pure in vitro study. Interestingly, to this day, AIS relocation during plasticity events has only been shown in cell culture systems, and not in vivo. Therefore, this needs to be put into context here - the chosen system is great for the type of imaging approach presented here, but may look at a type of AIS plasticity that is not seen in vivo.

Revision Plan

Reply: These are very good points. We will include the limitations of the study in the discussion. Indeed, due to technical and methodological challenges, the relocation of the AIS has not yet been demonstrated using animal models. However, in the study by Wefelmeyer et al. (Wefelmeyer et al., 2015), a similar relocation of the AIS resulting from chronic stimulation was observed in hippocampal organotypic slices, and it was accompanied by reduced excitability of neurons. Furthermore, in the same study, neurons with axons/AIS originating from basal dendrites were also mentioned. However, the measurement of chronic AIS plasticity in their study was not performed based on different classes of neuron types. Hence, our work complements their results. Given that the network connectivity of organotypic slices is much closer to real physiological conditions, it is likely that similar plastic adaptations could occur in vivo.

Minor comments

1. How does intrinsic neuronal activity play into developmental programs in vitro? Electrical activity in maturing neurons is a major part of how networks are shaped, and cells differentiate. This is not genetically encoded per se, but has been shown to be a major driving force of neuronal development in vivo. Is this reflected in the culture setting in any way? And have the authors considered testing early changes in activity patterns in their cultures to see whether AcDs and nonAcDs develop in similar percentages? To clarify, I am not asking for additional experiments.

Reply: It is indeed a valid point that activity can influence neuronal morphology. Lehmann et al. (pre-print, doi: <https://doi.org/10.1101/2023.07.31.551236>) have recently demonstrated that increased network activity leads to more excitatory principal neurons adopting AcD morphology. However, our developmental data were collected from DIV0 to DIV5, an age at which dissociated neurons do not yet form functional excitatory synapses. Therefore, it is highly unlikely that network activity plays a role in shaping AcD neuron development during this early stage.

2. The authors may want to add a bit of a technical discussion on the choice of KCl and TTX as triggers for plasticity, especially at the non-physiological concentrations offered here and elsewhere (15 mM KCl).

Reply: We appreciate the reviewer for pointing this out. We will add this in our revised manuscript.

3. Some key statements would benefit from citing the appropriate original literature (some examples would be the original work by Kole, Bender and Brette on the role of the AIS in AP initiation; original work by D'Este and Letterier on the dendritic and axonal scaffold using nanoscopy; work by Kim, Kuba and Jamann on AIS plasticity in vitro and in vivo that is critical for a more informed discussion of AIS plasticity here, and others)

Reply: These are very good points, we will make suggested edits in the revised version.

4. In the Introduction, the authors word their text explicitly for excitatory neurons. However, AIS plasticity has also been observed in interneurons (work by the Grubb lab for example), and axo-axonic synapses are in fact not all inhibitory - this is an important factor to consider given the embryonic state of the culture material. Does the DIV maturation reflect how axo-axonic synapses "switch" from excitatory to inhibitory in vivo (also see work of the Burrone lab)? Can the conclusions from the paper really be drawn based on this type of system?

Revision Plan

Reply: The AIS plasticity was indeed also observed in inhibitory interneurons (see Chand et al., 2015 for reference) and show opposite phenotypes compared to excitatory neurons. Also related to major comment #5, we did take the potential influence of AcD interneurons on the outcome of AIS plasticity experiment into consideration. Therefore, we also did a control experiment where inhibitory interneurons were labelled with GAD1 after chronic KCl treatment and these neurons were excluded from the analysis. Consistently, we got the same results that excitatory AcD neurons do not undergo chronic AIS plasticity. We will include this data in our revised manuscript. Further, in our current manuscript, we decided to focus on excitatory AcD neurons not only because they are the major functional unit in neuronal circuits, but also because the majority of the electrophysiological features were studied in excitatory AcD neurons. But we agree with the reviewer that AcD interneuron is definitely an interesting subject for follow up research in the future.

As mentioned by the reviewer, Pan-Vazquez et al. (Pan-Vazquez et al., 2020) nicely showed that axo-axonic synapses made by GABAergic Chandelier cells (ChCs) depolarise neurons in brain slices obtained from P12-18 animals. But this effect is reversed in slices obtained from older animals (>>P40). Of note, their results were based on cortical neurons but not hippocampal neurons, hence cell type specificity should be considered. More importantly, previous study reported that this conversion or switch of GABAergic interneurons from excitatory to inhibitory occurs on hippocampal neurons in P12-13 animals (Leinekugel et al., 1995). In dissociated hippocampal neurons from E18 rat embryos, this switch of GABAergic interneurons takes place on DIV9-11 and completes on DIV19, which should have a comparable neuronal developmental stage as the P12-13 in *in vivo* system (see Ganguly et al., 2001 for reference). Therefore, the conclusion could be drawn in an *in vitro* system, but it certainly needs to be validated in *in vivo* system.

5. The authors state that "less COs account for higher intrinsic excitability". Why is that the case?

Reply: According to Yu et al. and Bender et al., Ca²⁺ transient at the AIS regulates the generation of action potentials (APs). For instance, reducing Ca²⁺ transient at the AIS by blocking Ca²⁺ channels with either mibefradil (a T-type Ca²⁺ channel antagonist) or Ni²⁺ (which blocks R- and T-type channels) decreased the number of spikelets evoked by EPSP-like current injection and delayed the timing of spike generation (please see Bender & Trussell, 2009 for details). Therefore, we speculate that Ca²⁺ transients are less affected when there are fewer cisternal organelles (COs) at the AIS, which could have a more direct impact on AP initiation. However, this is just our hypothesis, and there is indeed no direct evidence showing that COs regulate Ca²⁺ dynamics. We will discuss this in the revised manuscript.

6. Last but not least, some very recent studies on AcD biology (Stevens, Thome, Lehmann, Wahle) is available online also on preprint servers and may provide additional support for the current study.

Reply: We will check these pre-prints and include relevant information into the revised version.

3. Description of the revisions that have already been incorporated in the transferred manuscript

Please insert a point-by-point reply describing the revisions that were already carried out and included in the transferred manuscript. If no revisions have been carried out yet, please leave this section empty.

Reviewer #1

Major comments:

3. As stated above, the manuscript is well structured and generally reads well. However, throughout the text there are always small mistakes that should be corrected by careful proofreading. Examples are

Page 6, last paragraph: ...AcD neurons generated [a] collateral...

Page 24, last paragraph: ... line was then drew [drawn] along ...

Page 24, last paragraph: Neurons with ... was consider [were considered] as ...

Page 25, first paragraph: Antibodies ... was [were]

Page 41, (E) Percentage of AcD neurons [that] generate [a] collateral or bifurcate

Reply: In the current version of revised manuscript, we already corrected most of the text errors. The edited part is indicated with blue colour.

Minor comments:

In the introduction, the authors describe how synaptic inputs are received at the dendrites and propagated to the soma in the form of membrane depolarizations. They should add 'excitatory' to synaptic inputs or also describe the impact of inhibitory synaptic inputs at the dendrites.

Reply: We mentioned excitatory synaptic inputs in the current version of revised manuscript (please see line 52 in the revised manuscript).

Reviewer #3

Minor comments:

1. At p. 7 is stated that "The percentage of none-AcD forming collaterals at DIV1 is much lower than for AcD neurons" but statistical support is lacking. **The conclusion in the next line is that "AcD neurons follow consensus development". That is puzzling given the difference just mentioned before. Please clarify.**

Reply to reviewer's point highlighted in yellow:

Regarding the second point concerning consensus development, we were referring to the general developmental sequence of AcD neurons, as described by Dotti et al. (see Dotti et al., 1988 for reference), where neurons typically first establish an axon and then dendrites. This sequence is not necessary related to collateral formation, which indeed differs between nonAcD and AcD neurons. The ability to form collaterals may come from local differences in microtubule (MT) and

Revision Plan

actin dynamics at AcD neuron precursor axons, but it does not alter the fact that AcD neurons initially establish an axon and subsequently dendrites. We clarified it in the edited manuscript.

Reviewer #4

Major comments:

2. Such critical parameters as e.g. synaptic innervation at the AIS are investigated in a way that does not support the clear statements given, e.g. "The AIS of AcD neurons receives fewer inhibitory inputs" (Highlights statement) or "AcD neurons have less inhibitory synapses at the AIS" (header of Fig. 6). The overall number of analyzed cells is low (3 and 4 preparations, respectively and approximately 50-cells for each marker). The combination of a pre- and postsynaptic marker for inhibitory / excitatory neurons is a solid decision, but the analysis is not done based on the close approximation of these markers, in 3D, along an AIS, but rather in maxIPs and without any regard of whether pre-and postsynaptic markers are actually close to each other or not. The expression of these markers alone just points towards the epitopes being expressed, but are they localized to each other in such a manner that they could form bona fide synapses? **The methods are not totally clear on the image depth (tile scans with 5 μm in z will not provide the detail of information to resolve synapses, so how did the authors address the subcellular analysis here and for the CO and VGSCs?). And generally, were Nyquist conditions taken into consideration throughout the study? This can be clarified in text and does not require additional experiments.**

Reply to reviewer's point highlighted in yellow:

The detailed description of microscopy was already written in the method section. Nyquist condition was considered during data acquisition, and pixel size of confocal image was already indicated in the method section for both laser scanning and spinning disk confocal microscopy. In the current version of revised manuscript, we indicated the pixel size also for phase contrast imaging (see line 586, text is coloured in blue).

3. **The rationale behind looking at the cisternal organelle (CO) in this study is outlined in the Introduction, where the authors state that "..... and is responsible for calcium handling". What is "calcium-handling" and where is the evidence cited?** Furthermore, in the Results, they state that "...both compounds (VGSCs and COs) are critical for the AIS to regulate neuronal excitability". While this is the case for VGSCs, there is no conclusive evidence in the literature whether or not the CO is "critical" for neuronal excitability. In fact, a number of neurons have no CO in the AIS (as much as 50% of all AIS in mouse primary visual cortex for example do not express synpo at the AIS at all, Schlüter et al., 2017). The CO can therefore not be as critical for AP initiation as the authors state. Furthermore, the authors state that "AIS plasticity in excitatory neurons is triggered by calcium signaling". While certainly shown and adequately cited here, other factors (independent of calcium) can also play a role, therefore this statement is a bit absolute and should be edited accordingly.

Reply to reviewer's point highlighted in yellow:

Revision Plan

For Ca^{2+} handling, we were referring to Ca^{2+} storage and release. In the revised manuscript, we rephrased the sentence to "... and is putatively responsible for Ca^{2+} storage and release..." (see line 112 blue text) as suggested by the reviewer.

5. The Discussion is more a list of future plans than a context to current data. The authors could move some of the new questions they identify into an "outlook" section at the end? Also, again have a critical look at the literature that is cited and which statements are accurate.

For example, the 2nd phrase in the Discussion states that it was shown that AcD neurons have a "role in memory consolidation", referenced to Hodapp et al., 2022. However, that paper does not provide direct evidence of such a role for AcD neurons. The statement "Collectively, our data provide new insights into the development of AcD neurons and demonstrate that there are differences in AIS functionality between AcD and nonAcD neurons", is not correct. AIS function was not investigated outside of the axonal barrier, and here, the AcD and nonAcD cells do not differ. Also, although the Discussion is geared towards excitatory / glutamatergic neurons, it has been shown by others that interneurons show an even stronger trend to exhibit AcD morphology (work by the Wahle lab and others). This is not clear from the current text (also compare "...AcD neurons being a different subtype of pyramidal neuron").

Further original publications should be included in the paragraph highlighting patch-clamp recordings (see above). In the same context, the statement "...showed that rapid AID plasticity occurs mainly in hippocampal dentate gyrus cells but not in principal excitatory neurons" is not accurate (see Kim, Kuba, Jamann and others). Generally, the Introduction and Discussion would benefit from a very clear distinction between studies done in vitro versus those done ex vivo or in vivo. This needs to be stated in the Abstract as well.

Reply to reviewer's point highlighted in yellow:

We appreciate reviewer for pointing it out. In the current version of revised manuscript, we rephrased sentence (see line 400-401) according to reviewer's suggestion. Together with comment from reviewer #3, we will have a further look into our interpretation of the literatures as well as citations during our revision time frame.

Minor comment:

4. The second header in Results is not clearly formulated. What is meant by "consensus developmental sequence"?

Reply: By consensus, we meant that AcD neuron still follow the canonical developmental sequence shown by Dotti et al. in vitro (also related to minor comment of reviewer #3). In the current version of revised manuscript, we replaced the word "consensus" to "canonical" for better understanding in the section of AcD neuron development.

4. Description of analyses that authors prefer not to carry out

Please include a point-by-point response explaining why some of the requested data or additional analyses might not be necessary or cannot be provided within the scope of a revision. This can be due to time or resource limitations or in case of disagreement about the necessity of such additional data given the scope of the study. Please leave empty if not applicable.

Reviewer #2

Major comments:

1. The authors suggest that there is reduced Na⁺ channel density at AcD AIS compared to other AIS arising from the cell body. This is not convincing. Immunostaining for Na⁺ channels is notoriously difficult and sensitive to fixation since the epitopes of the anti-Pan Nav antibodies are highly sensitive to fixation. In addition, this is based on immunofluorescence intensity quantification. Since the mechanism of localization is through binding to AnkG, the authors should also measure other AIS proteins like AnkG, b4 spectrin, and Nfasc. Do these change? If all uniformly change I would be much more inclined to accept the conclusion. If they do not change, it still doesn't rule out the concern about fixation conditions and slight differences in the cultures. **The authors indicate there is about a 40% reduction in fluorescence intensity. That is quite large. This big difference should also be confirmed in brain sections.**

Reply to reviewer's comment highlighted in yellow:

We agree with the reviewer that confirming observed differences in Na⁺ channel staining using brain slices would be beneficial. However, conducting such experiments presents several challenges. Firstly, one approach could involve immunostaining with antibodies against AIS marker AnkG, in combination with somatodendritic marker MAP2 and pan-Nav. However, this method lacks the advantage of clearly identifying neuronal morphology as seen in dissociated cultures, making the outcome unclear and difficult for analysis and interpretation. Alternatively, the use of Thy1-GFP rats, where a subset of neurons is labeled with GFP, could allow for morphological studies. Unfortunately, we do not have access to this rat line, and the process of importing it, obtaining permits, and establishing a colony is beyond the timeframe for manuscript revision. Additionally, while pan-Nav antibodies have shown reliability in dissociated cultures, their efficacy in tissue staining is less certain. We could provide example images upon request. Secondly, endogenously labelling of Na⁺ channels is another option, but remains a significant challenge. Recent developments in endogenous labeling, such as the CRISPR/Cas9-based method using pORANGE by Fréal et al. (Fréal et al., 2023), and the generation of Scn1a-GFP transgenic mice by Yamagata et al. (Yamagata et al., 2023), offer potential solutions. However, the labeling efficiency of pORANGE is uncertain, and both methods are time-consuming and cannot be completed within the three-month revision period.

3. Finally, this reviewer is also sceptical of the chronic plasticity changes in AIS 'distance.' The authors claim their results are consistent with prior reports, but they see about a 1.5 um shift. Prior studies (Grubb et al.) report 15-17 um change - a full order of magnitude larger than what is

Revision Plan

reported here. The authors also show no differences in other previously described changes at the AIS. Together with the other results showing AcD neurons and non- AcD neuron AIS are mostly the same, the conclusion that one behaves differently is not compelling with the tiny shift reported.

Reply: Yes, we are aware that in the prior study by Grubb et al. (see Grubb & Burrone, 2010 for reference), the difference in AIS shifting is much larger than what we discovered. However, it's worth mentioning that such dramatic activity-dependent shift of AIS was no longer reproducible by the same lab (see Evans et al., 2013 for reference). Several other papers have also reported changes in a more modest range (see Evans et al., 2017; Muir & Kittler, 2014; Wefelmeyer et al., 2015 for references). Possible differences may lie in the culturing conditions. In our study, we cultured neurons in BrainPhys medium with SM1 supplements, which is optimized for mature cultures with functional synapses. This is different from the Neurobasal medium with B27 supplements which was used by Grubb et al. and also in other papers. Since the experimental setups are not the same, and the concentration of inorganic salts required for electrophysiological activity in Neurobasal (only half of what neurons are exposed to in the brain) is much lower than in BrainPhys, these two studies cannot be directly compared.

More importantly, according to Bardy et al. (see Bardy et al., 2015 for reference), Neurobasal medium with B27 supplements significantly reduces or even abolishes the spontaneous excitatory synaptic activity of cultured neurons (Bardy et al.; Figure 2). Furthermore, the voltage-dependent sodium currents, rapidly inactivating potassium currents, and also the amplitude and frequency of evoked and spontaneous action potentials are reduced and impaired when neurons are cultured in Neurobasal medium (Bardy et al.; Figure 1D). However, this is not the case for BrainPhys with SM1 supplements (Figure 2), and the amplitudes of Nav and rapidly inactivating Kv currents, as well as synaptic activity of neurons cultured in BrainPhys, are similar to ACSF (Bardy et al.; Figure 2), meaning that neurons cultured in BrainPhys are much closer to the real physiological situation than in Neurobasal. This significant difference in activity level could impact various signaling pathways related to AIS plasticity. Therefore, we reasoned that the smaller AIS shift upon chronic activity, which we observed, is due to the higher level of spontaneous activity resulting from BrainPhys medium. Since we consistently observed differences in AIS shifting between the chronic KCl-stimulated and control groups over three independent experiments/preparations, we believe our conclusion is compelling and supports the idea that AcD neuron's AIS acts differently when they encounter chronic activity changes.

As mentioned in a previous in vitro study by Evans et al., rapid AIS plasticity, reflected as AIS shortening, occurs specifically in hippocampal dentate granule cells and to a lesser extent in CA3 neurons, but not in CA1 excitatory neurons (see Evans et al., 2015 for reference; Figure S3). This rapid form of AIS plasticity was also observed by Jahan et al. and Kuba et al. (see Jahan et al., 2023; Kuba et al., 2006 for references) in chicken auditory neurons, and in vivo by Jamann et al. in cortical neurons (see Jamann et al., 2021 for reference). Collectively, this evidence strongly suggests that rapid AIS plasticity might be specific to certain neuron types and regions in the central or peripheral nervous system.

As hippocampal primary neuronal cultures contain heterogeneous cell types derived from the CA1, CA2, CA3, and DG regions of the hippocampus, and it is possible that neurons from specific subareas survive less well, it is hard to estimate the proportion of neurons deriving from different subregions and the effects might be averaged out. We also proposed in the discussion that

Revision Plan

specific activation of post-excitatory synapses might be required to trigger rapid AIS plasticity. As nicely shown by Fréal et al. (see Fréal et al., 2023 for reference), after applying NMDA on slices for 3 hours to activate post-excitatory synapses, the AIS of CA1 excitatory neurons is shortened. This is indeed a very interesting aspect. However, given the amount of time required to set up and perform this experiment, we would like to omit this experiment from the current study.

5. I appreciated the balanced discussion of whether this is a stochastic or genetically programmed process. This could have been emphasized earlier in the results since the authors invoke the concept that "...their development must be driven by genetically encoded factors rather than specific...". **The authors have not shown this and cannot show it in this system.** Indeed, as stated in point 4 above, I think their data argue against a simple genetic program.

Reply to reviewer's point highlighted in yellow: Indeed, our current manuscript does not explore genetic factors involved in AcD neuron development. To address this question, one approach could be to label AIS markers endogenously in dissociated cultures using the PORANGE method (see Willems et al., 2020 for reference) or utilize AnkG-GFP transgenic mice (Fréal et al., 2023; Thome et al., 2023) along with a volume marker like mRuby or GFP. This would allow for the identification of AcD and nonAcD neurons in vivo and in vitro, followed by single-cell transcriptomics analysis to uncover potential genetic factors. Subsequently, candidate genes could be manipulated to demonstrate their essential role in AcD neuron development. However, such experiments require significant time and resources beyond the scope of our current revision timeframe. Nonetheless, this question presents an exciting direction for future research.

Reviewer #3

Major comments:

1. The authors classify neurons into axon-carrying dendrite (AcD) and non-AcD neurons by measuring the stem dendrite length ($> 3 \mu\text{m}$). I could not find the validity for this cut-off. The non-AcD neurons in Fig. 6B appear more AcD to this reviewer, and, in addition, other researchers have proposed a third category of 'shared root' neurons (doi: 10.7554/eLife.76101). For purposes of reproducibility and transparency, please provide first a comprehensive overview of the entire population of morphologies (i.e. all cells in control conditions). The distances from the soma could be plotted in histogram (etc.) **and authors may want to think about independent supporting evidence for the cut-off to classify AcD and non-AcD neurons.**

Reply to reviewer's comment highlighted in yellow:

We have considered having independent evidence to support the classification of nonAcD and AcD neurons. However, the method used by Thome et al. and Wahle et al. for AcD and nonAcD neuron classification is well established and widely accepted (see Thome et al., 2014; Wahle et al., 2022 for references). Similar standards were also employed by Benavides-Piccione et al. (Benavides-Piccione et al., 2020). Introducing independent evidence could potentially raise further doubts, so we have chosen to maintain consistency with previous studies.

3. Some interpretations about function are not correct and the authors should reconsider these. A role of cisternal organelles on neuronal excitability remains to be demonstrated (and see doi.org/10.1002/cne.21445 showing there is none). In addition, the statement that lower

Revision Plan

fluorescence intensity of Pan-Nav1 is indicating reduced excitability is flawed. Antibody staining does not scale linearly with voltage-gated sodium channel density and since the AIS of AcD neurons is further from the soma it is most likely smaller in diameter which may account for apparent fluorescent differences. For biophysical reasons (for details I refer to 10.3389/fncel.2019.00570, 10.1016/j.conb.2018.02.016 and 10.7554/eLife.53432) smaller diameter axons will be easier to depolarize by depolarizing voltage-gated channels or excitatory synapses. **Finally, in AcD neurons the AIS distance from the soma poses all sorts of interesting cable properties with the soma and the local dendritic membrane and the electrotonic properties alone suffice to make these neurons more excitable.**

Reply to reviewer's comment highlighted in yellow:

Exploring the biophysical properties of the AIS and axons of AcD neurons is indeed a highly interesting direction to pursue and is the project in its own. It would necessitate the use of computational modeling approaches, which require considerable time and resources that are not feasible within the timeframe of this revision.

Minor comments:

2. A study not cited in this manuscript showed distinct dendritic morphologies (doi: 10.1073/pnas.1607548113) and AcD interneurons are different for their axonal arborization (doi: 10.1242/dev.202305). **Differences in growth of branch arborization could hint to subtypes. Are the AcD and non-AcD neurons different in their adult morphology? A detailed account of the axonal and dendritic trees would strengthen the data.**

Reply to point highlighted in yellow:

Regarding the axons of AcD neurons, there is currently no detailed study available, and it would be more appropriate to investigate neuronal connectivity through tracing studies in animals rather than in primary cultures. Therefore, this question falls outside the scope of the current manuscript.

3. Some key references are not included here, and a number of these are mentioned above. In the context of the detailed MT and Rab3A vesicle and cargo transport studies, please acknowledge some of the pioneering work of Alan Peters revealing the ultrastructure of axons emerging from dendrites. See Figs. 5-7 in Peters, Proskauer and Kaiserman-Abramof IR., J Cell Biol 39:604 (1968). **What is the identity of the neurons? It makes a difference if the cells are interneurons or pyramidal neurons, CA1 or CA3-like.** For plasticity experiments the authors uses cells as independent measurements, but this is inflating the power. How many cultures were used?

Reply to point highlighted in yellow:

In our study, we focused on excitatory neurons from the hippocampus. We distinguished neuron types morphologically or with the inhibitory neuron marker GAD1. Identifying CA1, CA2, CA3, and DG subtypes in dissociated culture is more challenging, and this would be an interesting avenue to explore in an in vivo system. Here, we focused on fundamental cell biology aspects related to the AIS structure and its trafficking barrier function, which should be similar in all these neuron types. While there may be subtype-specific differences in AIS plasticity, investigating this is beyond the scope of our manuscript.

Revision Plan

Reviewer 4

Major comments:

4. The Introduction ends with the rationale of the study, namely that the authors seek to "provide a detailed characterization of the AIS, including its structural and functional properties....". Structure is investigated, but function is limited to the barrier function of the AIS. Since the authors provide no electrophysiology that would really dissect AIS function, I suggest to rephrase this part and focus on transport.

Reply to point highlighted in yellow:

We would like to keep the term "AIS function", because it has already been nicely demonstrated electrophysiologically by previous studies that the plasticity effect of the AIS is very important for maintaining cellular homeostasis.

References:

- Bardy, C., Van Den Hurk, M., Eames, T., Marchand, C., Hernandez, R. V., Kellogg, M., Gorris, M., Galet, B., Palomares, V., Brown, J., Bang, A. G., Mertens, J., Böhnke, L., Boyer, L., Simon, S., & Gage, F. H. (2015). Neuronal medium that supports basic synaptic functions and activity of human neurons in vitro. *Proceedings of the National Academy of Sciences of the United States of America*, *112*(20), E2725–E2734. <https://doi.org/10.1073/pnas.1504393112>
- Benavides-Piccione, R., Regalado-Reyes, M., Fernaud-Espinosa, I., Kastanauskaite, A., Tapia-González, S., León-Espinosa, G., Rojo, C., Insausti, R., Segev, I., & Defelipe, J. (2020). Differential Structure of Hippocampal CA1 Pyramidal Neurons in the Human and Mouse. *Cerebral Cortex*, *30*(2), 730–752. <https://doi.org/10.1093/cercor/bhz122>
- Bender, K. J., & Trussell, L. O. (2009). Axon Initial Segment Ca²⁺ Channels Influence Action Potential Generation and Timing. *Neuron*, *61*(2), 259–271. <https://doi.org/10.1016/j.neuron.2008.12.004>
- Benedeczy, I., Molnár, E., & Somogyi, P. (1994). The cisternal organelle as a Ca²⁺-storing compartment associated with GABAergic synapses in the axon initial segment of hippocampal pyramidal neurones. *Experimental Brain Research*, *101*(2), 216–230. <https://doi.org/10.1007/BF00228742>
- Cabukusta, B., & Neefjes, J. (2018). Mechanisms of lysosomal positioning and movement. *Traffic*, *19*(10), 761–769. <https://doi.org/10.1111/tra.12587>
- Carabalona, A., Hu, D. J. K., & Vallee, R. B. (2016). KIF1A inhibition immortalizes brain stem cells but blocks BDNF-mediated neuronal migration. *Nature Neuroscience*, *19*(2), 253–262. <https://doi.org/10.1038/nn.4213>
- Chand, A. N., Galliano, E., Chesters, R. A., & Grubb, M. S. (2015). A distinct subtype of dopaminergic interneuron displays inverted structural plasticity at the axon initial segment. *Journal of Neuroscience*, *35*(4), 1573–1590. <https://doi.org/10.1523/JNEUROSCI.3515-14.2015>
- Dotti, C. G., Sullivan, C. A., & Banker, G. A. (1988). The establishment of polarity by hippocampal neurons in culture. *Journal of Neuroscience*, *8*(4). <https://doi.org/10.1523/jneurosci.08-04-01454.1988>
- Evans, M. D., Dumitrescu, A. S., Kruijssen, D. L. H., Taylor, S. E., & Grubb, M. S. (2015). Rapid Modulation of Axon Initial Segment Length Influences Repetitive Spike Firing. *Cell Reports*, *13*(6), 1233–1245. <https://doi.org/10.1016/j.celrep.2015.09.066>
- Evans, M. D., Sammons, R. P., Lebron, S., Dumitrescu, A. S., Watkins, T. B. K., Uebele, V. N., Renger, J. J., & Grubb, M. S. (2013). Calcineurin signaling mediates activity-dependent relocation of the Axon Initial segment. *Journal of Neuroscience*, *33*(16), 6950–6963. <https://doi.org/10.1523/JNEUROSCI.0277-13.2013>
- Evans, M. D., Tufo, C., Dumitrescu, A. S., & Grubb, M. S. (2017). Myosin II activity is required for structural plasticity at the axon initial segment. *European Journal of Neuroscience*, *46*(2), 1751–1757. <https://doi.org/10.1111/ejn.13597>
- Fréal, A., Jamann, N., Bos, J. Ten, Jansen, J., Petersen, N., Ligthart, T., Hoogenraad, C. C., & Kole, M. H. P. (2023). Sodium channel endocytosis drives axon initial segment plasticity. *Science Advances*, *9*(37), 1–16. <https://doi.org/10.1126/sciadv.adf3885>

Revision Plan

- Ganguly, K., Schinder, A. F., Wong, S. T., & Poo, M. ming. (2001). GABA itself promotes the developmental switch of neuronal GABAergic responses from excitation to inhibition. *Cell*, 105(4), 521–532. [https://doi.org/10.1016/S0092-8674\(01\)00341-5](https://doi.org/10.1016/S0092-8674(01)00341-5)
- Grubb, M. S., & Burrone, J. (2010). Activity-dependent relocation of the axon initial segment fine-tunes neuronal excitability. *Nature*, 465(7301), 1070–1074. <https://doi.org/10.1038/nature09160>
- Guedes-Dias, P., & Holzbaur, E. L. F. (2019). Axonal transport: Driving synaptic function. *Science*, 366(6462), 1–35. <https://doi.org/10.1126/science.aaw9997>
- Helmer, P., & Vallee, R. B. (2023). A two-kinesin mechanism controls neurogenesis in the developing brain. *Communications Biology*, 6(1). <https://doi.org/10.1038/s42003-023-05604-5>
- Hodapp, A., Kaiser, M. E., Thome, C., Ding, L., Rozov, A., Klumpp, M., Stevens, N., Stingl, M., Sackmann, T., Lehmann, N., Draguhn, A., Burgalossi, A., Engelhardt, M., & Both, M. (2022). Dendritic axon origin enables information gating by perisomatic inhibition in pyramidal neurons. *Science*, 377(6613), 1448–1452. <https://doi.org/10.1126/science.abj1861>
- Jahan, I., Adachi, R., Egawa, R., Nomura, H., & Kuba, H. (2023). CDK5/p35-Dependent Microtubule Reorganization Contributes to Homeostatic Shortening of the Axon Initial Segment. *Journal of Neuroscience*, 43(3), 359–372. <https://doi.org/10.1523/JNEUROSCI.0917-22.2022>
- Jamann, N., Dannehl, D., Lehmann, N., Wagener, R., Thielemann, C., Schultz, C., Staiger, J., Kole, M. H. P., & Engelhardt, M. (2021). Sensory input drives rapid homeostatic scaling of the axon initial segment in mouse barrel cortex. *Nature Communications*, 12(1), 1–14. <https://doi.org/10.1038/s41467-020-20232-x>
- Kapitein, L. C., & Hoogenraad, C. C. (2011). Which way to go? Cytoskeletal organization and polarized transport in neurons. *Molecular and Cellular Neuroscience*, 46(1), 9–20. <https://doi.org/10.1016/j.mcn.2010.08.015>
- Katrukha, E. A., Jurriens, D., Pastene, D. S., & Kapitein, L. C. (2021). Quantitative mapping of dense microtubule arrays in mammalian neurons. *ELife*, 10, 1–25. <https://doi.org/10.7554/eLife.67925>
- Kuba, H., Ishii, T. M., & Ohmori, H. (2006). Axonal site of spike initiation enhances auditory coincidence detection. *Nature*, 444(7122), 1069–1072. <https://doi.org/10.1038/nature05347>
- Leinekugel X, Tseeb V, Ben-Ari Y, Bregestovski P. (1995) Synaptic GABAA activation induces Ca²⁺ rise in pyramidal cells and interneurons from rat neonatal hippocampal slices. *J Physiol.* 1;487, 319-29. doi: 10.1113/jphysiol.1995.sp020882.
- Lipka, J., Kapitein, L. C., Jaworski, J., & Hoogenraad, C. C. (2016). Microtubule-binding protein doublecortin-like kinase 1 (DCLK1) guides kinesin-3-mediated cargo transport to dendrites. *The EMBO Journal*, 35(3), 302–318. <https://doi.org/10.15252/embj.201592929>
- Muir, J., & Kittler, J. T. (2014). Plasticity of GABAA receptor diffusion dynamics at the axon initial segment. *Frontiers in Cellular Neuroscience*, 8(JUN), 1–11. <https://doi.org/10.3389/fncel.2014.00151>
- Niwa, S., Tanaka, Y., & Hirokawa, N. (2008). KIF1B β - and KIF1A-mediated axonal transport of presynaptic regulator Rab3 occurs in a GTP-dependent manner through DENN/MADD. *Nature Cell Biology*,

Revision Plan

10(11), 1269–1279. <https://doi.org/10.1038/ncb1785>

- Pan-Vazquez, A., Wefelmeyer, W., Gonzalez Sabater, V., Neves, G., & Burrone, J. (2020). Activity-Dependent Plasticity of Axo-axonic Synapses at the Axon Initial Segment. *Neuron*, *106*(2), 265–276.e6. <https://doi.org/10.1016/j.neuron.2020.01.037>
- Rienecker, K. D. A., Poston, R. G., & Saha, R. N. (2020). Merits and Limitations of Studying Neuronal Depolarization-Dependent Processes Using Elevated External Potassium. *ASN Neuro*, *12*. <https://doi.org/10.1177/1759091420974807>
- Sánchez-Ponce, D., DeFelipe, J., Garrido, J. J., & Muñoz, A. (2011). In vitro maturation of the cisternal organelle in the hippocampal neuron's axon initial segment. *Molecular and Cellular Neuroscience*, *48*(1), 104–116. <https://doi.org/10.1016/j.mcn.2011.06.010>
- Schlüter, A., Rossberger, S., Dannehl, D., Janssen, J. M., Vorwald, S., Hanne, J., Schultz, C., Mauceri, D., & Engelhardt, M. (2019). Dynamic Regulation of Synaptopodin and the Axon Initial Segment in Retinal Ganglion Cells During Postnatal Development. *Frontiers in Cellular Neuroscience*, *13*(July), 1–18. <https://doi.org/10.3389/fncel.2019.00318>
- Solé, L., Wagnon, J. L., Akin, E. J., Meisler, M. H., & Tamkun, M. M. (2019). The MAP1B binding domain of Nav1.6 is required for stable expression at the axon initial segment. *Journal of Neuroscience*, *39*(22), 4238–4251. <https://doi.org/10.1523/JNEUROSCI.2771-18.2019>
- Tas, R. P., Chazeau, A., Cloin, B. M. C., Lambers, M. L. A., Hoogenraad, C. C., & Kapitein, L. C. (2017). Differentiation between Oppositely Oriented Microtubules Controls Polarized Neuronal Transport. *Neuron*, *96*(6), 1264–1271.e5. <https://doi.org/10.1016/j.neuron.2017.11.018>
- Thome, C., Janssen, J. M., Karabulut, S., Acuna, C., D'Este, E., Soyka, S. J., Baum, K., Bock, M., Lehmann, N., Hasegawa, M., Ganea, D. A., Benoit, C. M., Gründemann, J., Schultz, C., Bennett, V., Jenkins, P. M., & Engelhardt, M. (2023). Live imaging of excitable axonal microdomains in ankyrin-G-GFP mice. *BioRxiv*, February, 1–11.
- Thome, C., Kelly, T., Yanez, A., Schultz, C., Engelhardt, M., Cambridge, S. B., Both, M., Draguhn, A., Beck, H., & Egorov, A. V. (2014). Axon-carrying dendrites convey privileged synaptic input in hippocampal neurons. *Neuron*, *83*(6), 1418–1430. <https://doi.org/10.1016/j.neuron.2014.08.013>
- Wahle, P., Sobierajski, E., Gasterstädt, I., Lehmann, N., Weber, S., Lübke, J. H. R., Engelhardt, M., Distler, C., & Meyer, G. (2022). Neocortical pyramidal neurons with axons emerging from dendrites are frequent in non-primates, but rare in monkey and human. *ELife*, *11*, 1–25. <https://doi.org/10.7554/eLife.76101>
- Wefelmeyer, W., Cattaert, D., & Burrone, J. (2015). Activity-dependent mismatch between axo-axonic synapses and the axon initial segment controls neuronal output. *Proceedings of the National Academy of Sciences of the United States of America*, *112*(31), 9757–9762. <https://doi.org/10.1073/pnas.1502902112>
- Willems, J., de Jong, A. P. H., Scheefhals, N., Mertens, E., Catsburg, L. A. E., Poorthuis, R. B., de Winter, F., Verhaagen, J., Meye, F. J., & MacGillavry, H. D. (2020). Orange: A CRISPR/Cas9-based genome editing toolbox for epitope tagging of endogenous proteins in neurons. In *PLoS Biology* (Vol. 18, Issue 4). <https://doi.org/10.1371/journal.pbio.3000665>

Revision Plan

- Yamagata, T., Ogiwara, I., Tatsukawa, T., Suzuki, T., Otsuka, Y., Imaeda, N., Mazaki, E., Inoue, I., Tokonami, N., Hibi, Y., Itohara, S., & Yamakawa, K. (2023). Scn1a-GFP transgenic mouse revealed Nav1.1 expression in neocortical pyramidal tract projection neurons. *eLife*, *12*, 1–29. <https://doi.org/10.7554/eLife.87495>
- Yang, S. M., Michel, K., Jokhi, V., Nedivi, E., Arlotta, P., Biology, R., & Sciences, C. (2020). Neuron class-specific responses govern adaptive myelin remodeling in the neocortex. *370*(6523), 1–19. <https://doi.org/10.1126/science.abd2109>.Neuron-class
- Yu, Y., Maureira, C., Liu, X., & McCormick, D. (2010). P/Q and N channels control baseline and spike-triggered calcium levels in neocortical axons and synaptic boutons. *Journal of Neuroscience*, *30*(35), 11858–11869. <https://doi.org/10.1523/JNEUROSCI.2651-10.2010>

April 1, 2024

Re: JCB manuscript #202403141T

Prof. Marina Mikhaylova
Humboldt-Universität zu Berlin
AG Optobiology
Invalidenstrasse 42
Berlin, Berlin 10115
Germany

Dear Prof. Mikhaylova,

Thank you for submitting your manuscript entitled "Unveiling the cell biology of hippocampal neurons with dendritic axon origin" from Review Commons. We have assessed your manuscript and revision plan in response to the reviewers from Review Commons, and agree that a study revised as described is suitable for further consideration at JCB. Therefore, we invite you to submit the revised manuscript to JCB.

GENERAL GUIDELINES:

Text limits: Character count for an Transfer is < 40,000, not including spaces. Count includes title page, abstract, introduction, results, discussion, and acknowledgments. Count does not include materials and methods, figure legends, references, tables, or supplemental legends.

Figures: Transfers may have up to 10 main text figures. Figures must be prepared according to the policies outlined in our Instructions to Authors, under Data Presentation, <https://jcb.rupress.org/site/misc/ifora.xhtml>. All figures in accepted manuscripts will be screened prior to publication.

*****IMPORTANT:** It is JCB policy that if requested, original data images must be made available. Failure to provide original images upon request will result in unavoidable delays in publication. Please ensure that you have access to all original microscopy and blot data images before submitting your revision. ***

Supplemental information: There are strict limits on the allowable amount of supplemental data. Transfers may have up to 5 supplemental figures. Up to 10 supplemental videos or flash animations are allowed. A summary of all supplemental material should appear at the end of the Materials and methods section.

Please note that JCB now requires authors to submit Source Data used to generate figures containing gels and Western blots with all revised manuscripts. This Source Data consists of fully uncropped and unprocessed images for each gel/blot displayed in the main and supplemental figures. Since your paper includes cropped gel and/or blot images, please be sure to provide one Source Data file for each figure that contains gels and/or blots along with your revised manuscript files. File names for Source Data figures should be alphanumeric without any spaces or special characters (i.e., SourceDataF#, where F# refers to the associated main figure number or SourceDataFS# for those associated with Supplementary figures). The lanes of the gels/blots should be labeled as they are in the associated figure, the place where cropping was applied should be marked (with a box), and molecular weight/size standards should be labeled wherever possible.

The typical timeframe for revisions is three to four months. While most universities and institutes have reopened labs and allowed researchers to begin working at nearly pre-pandemic levels, we at JCB realize that the lingering effects of the COVID-19 pandemic may still be impacting some aspects of your work, including the acquisition of equipment and reagents. Therefore, if you anticipate any difficulties in meeting this aforementioned revision time limit, please contact us and we can work with you to find an appropriate time frame for resubmission. Please note that papers are generally considered through only one revision cycle, so any revised manuscript will likely be either accepted or rejected.

Thank you for this interesting contribution to Journal of Cell Biology. You can contact us at the journal office with any questions at cellbio@rockefeller.edu.

Sincerely,

Kassandra Ori-McKenney, PhD
Monitoring Editor

Andrea L. Marat, PhD
Senior Scientific Editor

Journal of Cell Biology

Point-by-point reply to reviewers' comments

Manuscript number: RC-2024-02349 (transfer from Review Commons)

Corresponding author(s): Kay Grünewald and Marina Mikhaylova

We would like to thank all four reviewers for their valuable comments. We have conducted all the experiments and analyses proposed in the revision plan. A point-by-point response to specific comments can be found below.

Reviewer_01

Major comments:

1. The authors cite that acetylated and tyrosinated microtubules have different spatial and compartmental distribution in dendrites and axons and investigate the distribution in the AIS of nonAcD cells and AcD cells, as well as the stem dendrites. However, they just show one example of two different cells (Figure 2D and E) without any statistical analysis. Either, they should remove this part or provide a thorough quantification.

Reply: The spatial and compartmentalized distribution of stable and dynamic MTs in the dendrites and axons of nonAcD neurons has been extensively studied and reviewed (see Kapitein & Hoogenraad, 2011; Katrukha et al., 2021; Tas et al., 2017 for reference). However, the organization of the MT cytoskeleton in AcD neurons is still unknown. Here, we provide the very first evidence on the distribution of tyrosinated and acetylated MTs in AcD neurons, as well as data on MT orientations. We agree with the reviewer that to make our results on the spatial organization of these post-translational modifications in AcD neurons more complete, we need to provide a more thorough quantification analysis.

To achieve this, we plan to perform immunostainings on DIV10 neurons using antibodies against tyrosinated (tyr) and acetylated (ac-) tubulin to label dynamic and stable MTs, respectively. Subsequently, we will conduct high-resolution 3D confocal imaging and measure fluorescent intensity to illustrate the abundance and staining patterns of tyr- and ac- MTs in the axons and dendrites of AcD neurons. Since the spatial distribution of tyr- and ac-MTs is distinguishable with confocal microscopy, we will retain STED examples in the figures but conduct new analyses on confocal imaging data. We will measure the total fluorescent intensity of tyr- and ac- MTs in different compartments of AcD neurons and normalize it to the size of the measured area. We will then compare the normalized intensity values between the axons and dendrites of AcD neurons to examine whether there is a specific distribution pattern of stable and dynamic MTs. We will analyse at least 3 independent primary culture preparations with a minimum of 30 cells. Using the same dataset, we will also quantify the percentage of AcD neurons with ac-MTs specifically elongating into the axon compared to dendrites.

Outcome:

To address the reviewer's point, we performed super-resolution confocal imaging of tyr- and ac-MTs in AcD and nonAcD neurons using the SoRa spinning disk confocal system. We included at least 30 AcD and nonAcD neurons over three independent experiments and analysed the fluorescent intensity of tyr- and ac- MTs in different compartments of AcD and nonAcD neurons. We found no statistical difference in fluorescent intensity of ac-MTs between different neuronal compartments in both AcD and nonAcD neurons. Interestingly, this new analysis showed that the stem dendrite of AcD neurons is enriched in dynamic MTs, as the intensity of tyr-MTs is higher at this region compared to the other compartments. Consistently, at the stem dendrite of AcD neurons, the density of EB3 comets marking mostly dynamic MTs, is also higher than in the axon and somatic dendrites. These results are now included in the revised Figure 2.

2. The authors use EGFP-Rab3A vesicle to investigate anterograde transport at the axon and dendrites. They find a slightly faster transport of these vesicles at the AIS of AcD cells and conclude the axonal cargos in general are transported faster across the AIS in AcD cells. In my opinion, this generalization based on one type of vesicle is too farfetched.

Reply: The Rab3A protein is associated with pre-synaptic vesicles that are transported by KIF1A and KIF1B β , members of the kinesin-3 family, towards pre-synaptic buttons (see Guedes-Dias & Holzbaur, 2019; Niwa et al., 2008 for reference). Since KIF1A and KIF1B β are common motor proteins that mediate MT-based transport of different types of vesicles (e.g., synaptic vesicles and dense-core vesicles, see Carabalona et al., 2016; Helmer & Vallee, 2023 for reference), we reasoned that Rab3A should be a representative marker for an axonal cargo. However, this indeed does not rule out whether the faster trafficking effect we saw is specific to presynaptic vesicles, as different types of vesicles tend to recruit different modulators that could lead to different trafficking features.

To address this question, we will perform a live-imaging experiment including two additional organelle marker proteins, Neuropeptide Y (NPY) and Lysosome-associated membrane protein 1 (Lamp1). NPY is transported into the axon via KIF1A and KIF1B β -mediated dense-core vesicles (see Helmer & Vallee, 2023; Lipka et al., 2016 for reference), and Lamp1 is associated with lysosomes and a range of endocytic organelles that recruit both kinesin-1 and kinesin-3 and are transported into both axons and dendrites (as reviewed in Cabukusta & Neefjes, 2018). By introducing two additional types of vesicles, we should be able to answer whether AcD neurons, in general, tend to transport cargoes into the axon faster than nonAcD neurons.

Outcome:

As proposed, we included live imaging of two additional organelle markers, NPY and LAMP1. Interestingly, we found that there are no differences in the velocity of NPY- and LAMP1-positive vesicles when they pass through the AIS of AcD or nonAcD neurons. This indicates that the higher velocity within the AIS of AcD neurons is specific to Rab3A vesicles. These data are now included into **Figure S3**.

Minor comments:

In the introduction, the authors describe how synaptic inputs are received at the dendrites and propagated to the soma in the form of membrane depolarizations. They should add 'excitatory' to synaptic inputs or also describe the impact of inhibitory synaptic inputs at the dendrites.

In my opinion, Figure 2 could be presented in a slightly better way. The lower part of panel A better fits to panel B, which is next to the upper part of panel A. I understand that the authors systematically present their data first for nonAcD cells and then for AcD cells. However, in this special case it is a little bit more difficult to read the current figure in that order. The results displayed in Figure 4 are presented in a slightly confusing order. The authors jump from 4D to 4G, then to 4I and 4E, 4H, 4F. Similarly, 4M and N are addressed before 4O and P to finally get to 4K and L. It would be beneficial to present and address the data in a stringent way.

Reply: Thank you for the suggestions on how to improve the data representation in the figures.

We will change Figures 2 and 4 and make adjustments in the text upon revision since we also plan to include additional data.

Outcome:

Following the reviewer's suggestions, we edited the content of **Figure 4** and the corresponding text in the results section. For **Figure 2**, we revised the sections on MT tyrosination and acetylation to include our new analysis, as suggested by the reviewer in "major comment 1". Regarding the MT orientation assay, we did not swap the kymographs but included two new measurements for EB3 comet density and growth rate. The flow of the content should now be improved.

Reviewer_02

Major comments:

1. The authors suggest that there is reduced Na⁺ channel density at AcD AIS compared to other AIS arising from the cell body. This is not convincing. Immunostaining for Na⁺ channels is notoriously difficult and sensitive to fixation since the epitopes of the anti-Pan Nav antibodies are highly sensitive to fixation. In addition, this is based on immunofluorescence intensity quantification. Since the mechanism of localization is through binding to AnkG, the authors should also measure other AIS proteins like AnkG, β 4 spectrin, and Nfasc. Do these change? If all uniformly change I would be much more inclined to accept the conclusion. If they do not change, it still doesn't rule out the concern about fixation conditions and slight differences in the cultures. The authors indicate there is about a 40% reduction in fluorescence intensity. That is quite large. This big difference should also be confirmed in brain sections.

Reply: The potential fixation issue and antibody sensitivity on Na⁺ channel staining are indeed valid considerations, and we are aware of them. However, it should be noted that we used pan-Na⁺ channel antibodies that were previously characterized and widely used in the literature (see Solé et al., 2019; Yang et al., 2020 for references). Furthermore, our samples underwent the same fixation and staining protocol, and comparable numbers of AcD and nonAcD neurons were imaged from the same preparation and coverslip for each experiment. Imaging settings were also kept constant. Any loss of Na⁺ channel staining at the AIS due to fixation should affect both neuron types and therefore our conclusion is justified. Nevertheless, the reviewer's point regarding other AIS components is valid and will be investigated further in the revised manuscript.

Following the reviewer's suggestion to further strengthen our conclusion, we will measure the intensity of AnkG, β IV-spectrin, and neurofascin in DIV21 AcD and nonAcD neurons. We will compare a minimum of 3 independent cultures, each containing at least 10 cells of each type per culture.

We agree with the reviewer that confirming observed differences in Na⁺ channel staining using brain slices would be beneficial. However, conducting such experiments presents several challenges. Firstly, one approach could involve immunostaining with antibodies against AIS marker AnkG, in combination with somatodendritic marker MAP2 and pan-Nav. However, this method lacks the advantage of clearly identifying neuronal morphology as seen in dissociated cultures, making the outcome unclear and difficult for analysis and interpretation. Alternatively, the use of Thy1-GFP rats, where a subset of neurons is labeled with GFP, could allow for morphological studies. Unfortunately, we do not have access to this rat line, and the process of importing it, obtaining permits, and establishing a colony is beyond the timeframe for manuscript revision. Additionally, while pan-Nav antibodies have shown reliability in dissociated cultures, their efficacy in tissue staining is less certain. We could provide example images upon request. Secondly, endogenously labelling of Na⁺ channels is another option, but remains a significant challenge. Recent developments in endogenous labeling, such as the

CRISPR/Cas9-based method using pORANGE by Fréal et al. (Fréal et al., 2023), and the generation of Scn1a-GFP transgenic mice by Yamagata et al. (Yamagata et al., 2023), offer potential solutions. However, the labeling efficiency of pORANGE is uncertain, and both methods are time-consuming and cannot be completed within the three-month revision period.

As an alternative, we propose emphasizing that our results are based on *in vitro* experiments and discussing the advantages and limitations of this approach in the discussion section.

Outcome:

To address the reviewer's concern regarding the reduction of sodium channel intensity, we performed additional experiments to measure the intensity of other AIS proteins and compared AcD neurons with nonAcD neurons. We primarily focused on AnkG and neurofascin that directly bind to sodium channels at the AIS. However, we found no differences in their fluorescent intensity between the AIS of the two types of neuron, suggesting the amount of AnkG and neurofascin is likely the same. β 4-spectrin staining was analysed using dSTORM and we found no structural difference in the MPS at the AIS of AcD and nonAcD neurons. Furthermore, as reviewed by Eshed-Eisenbach et al. (doi: 10.1002/dneu.22728), β 4-spectrin interacts with sodium channels in a more indirect way and mostly through AnkG, therefore we have not included further characterization of β 4-spectrin intensities.

Since we did not find any differences in other AIS proteins, we next examined our data and statistical analysis for sodium channels very carefully as well as performed additional experiments, as this was also requested by Reviewer#4. We first identified potential outliers of our dataset by calculating the interquartile range (IQR) of the dataset and set the outlier limit to 1.5 times below Q1 (25th percentile) and above Q3 (75th percentile). We also checked the sample size of our experiments to make sure they are comparable between different groups. After careful examination, we found that the difference we saw in sodium channel intensity was likely due to an outlier in the dataset, as the p value changed from 0.042 to 0.056 after removing it. Therefore, to avoid overstatements we edited our conclusion to sodium channels as non-significant.

Regarding the second point of the reviewer, in the revised manuscript we now mention that our results are based on *in vitro* work and have included the discussion about advantages and disadvantages of such a model system.

2. The analysis of inhibitory synapse differences at the AIS are also not compelling - this is a limitation of the culture system. The authors have no control over the density of inhibitory neurons in the culture well. This interaction is not intrinsic to the AcD neuron, but rather a feature of neuron-neuron interactions which should only be modelled in the animal.

Reply: The reviewer is correct in pointing out that establishing inhibitory synapses at the AIS is not an intrinsic feature of AcD neurons; it depends on the network and should be modelled in animals. We will include this limitation of the cell culture model in the discussion section in the revised manuscript. We also understand the reviewer's concern that the lower amount of inhibitory synapses at AcD neuron AIS might be due to uneven density of inhibitory neurons between cultures. Nonetheless, assuming that the number of inhibitory neurons is constant

between preparations, it is an interesting observation that AcD neurons form fewer inhibitory synapses at the AIS. This may be related to the features of the AIS and its morphology and should be further investigated.

To make our study more comprehensive and also address the reviewer's concern regarding the presence of inhibitory neurons, we will perform immunostainings in dissociated cultures (40.000 cells per 18 mm coverslip, same as in experiments with synapse quantification) with antibodies against pCaMKIIa, an excitatory neuron marker, and GAD1, a marker for inhibitory neurons. Then, we will quantify the density of inhibitory neurons in the culture. We will perform measurements from 3-6 independent cultures by analyzing large fields of view in different areas of a coverslip (20-30 neurons per area) to determine if the density of inhibitory neurons varies between cultures as well as preparations. Furthermore, as also requested by reviewer 4, we will perform new immunostainings where pre- and post-synaptic markers (VGAT and Gephyrin) will be included in the same sample together with the AIS (AnkG or Neurofascin) and dendritic marker (MAP2). Synapses that contain pre- and post-synaptic components will be analyzed and included in the revised version of the manuscript.

Outcome:

In the revised manuscript, we performed the proposed experiments (updated **Figure 6**). First, by calculating the coefficient of variation, we found the variation of inhibitory interneurons between cultures is very low. Second, quantification of putative inhibitory synapses detected as co-localisation of gephyrin and VGAT puncta revealed that there are fewer synapses on the AIS of AcD neurons compared to nonAcD. Therefore, our previous conclusion remains correct. Nonetheless, as mentioned in the previous comment, we are aware of the limitations of our model system and have included this notion in the revised manuscript.

4. Finally, the major limitation of this study is that it is performed in vitro. Surprisingly, the authors actually argue this is a feature of their system. While it is true some of the questions can be addressed perfectly well in vitro, many cannot. In the first paragraph of the results the authors state an advantage of their system is that there are no microenvironments to influence the development of the AcDs. I'm afraid I view this as a drawback. The authors suggest this is an opportunity to examine intrinsic mechanisms of development - true, but it also foregoes the opportunity to determine if the outcomes are different from what occurs in vivo. To this point, the authors report that only 15-20% of the population of hippocampal neurons in culture are AcD neurons. But in their introduction they cite other literature indicating 50% of hippocampal neurons in vivo are AcD neurons - this suggests that the environment of the hippocampus in vivo influences whether a neuron becomes an AcD neuron or not.

Reply: The reviewer is right in pointing that the in vivo environment could indeed affect AcD neuron development, and we also find this to be a very interesting topic to investigate in the future. Even more intriguingly, as shown by Lehmann et al. (doi: <https://doi.org/10.1101/2023.07.31.551236>), network activity stimulates neurons to acquire AcD morphology. While it is true that the impact of the microenvironment on AcD neuron

development cannot be studied in dissociated cultures, our *in vitro* data undoubtedly support the fact that hippocampal neurons can intrinsically develop into AcD morphology independent of the *in vivo* environment. As also mentioned in the next point, our statement "...their development must be driven by genetically encoded factors rather than specific..." might sound too definitive and therefore eliminate possible effects from the microenvironment. We will revise this part. Although it is highly desirable to move cell biological studies from neuronal cell cultures to tissue, to date, it is still very challenging to perform many of experiments which we did in this study in slices or living animals due to a lack of appropriate technologies and tools. We are convinced that many basic biological questions can be and should be studied in simplified culturing models because they are truly fundamental, they should also be reproducible in these models.

To address the reviewer's question regarding the percentage difference between our data and the previous study by Thome et al., several factors should be considered. First, as noted by the reviewer, our results were obtained from an *in vitro* system, which is not directly comparable to the *in vivo* model system used in Thome et al.'s study (Thome et al., 2014). Second, the age of the neurons quantified in our developmental experiments is DIV5 and DIV7. This young age disparity could contribute to the percentage difference, as Thome et al. analyzed neurons from P28-35 adult animals, where 50% of the AcD neuron population was observed, specifically in the CA1 region. Third, it's important to note that in other hippocampal regions, the percentage of AcD neurons is lower (approximately 20-30%). Since our hippocampal primary cultures contain neurons from all hippocampal regions, this may have averaged out our quantification of AcD neuron percentage. Additionally, in the study by Benavides-Piccione et al. (Benavides-Piccione et al., 2020), they reported 20% AcD neurons in the CA1 region of hippocampi isolated from 8-week-old mouse pups, a number similar to what we observed *in vitro*. Interestingly, Thome et al. reported that in P8 pups, AcD neuron population in hippocampal CA1 region is 30%. This number increased to 50% in adult animals at age of P28-35, suggesting there is perhaps an age dependent increase of AcD neuron population. This could be an additional reason of why we only saw 15-20% of AcD neurons in our *in vitro* system, regardless of the *in vivo* environment.

In the revised version, we will clarify these points in the introduction and discussion sections. Additionally, we will quantify the proportion of AcD neurons in mature DIV21 dissociated hippocampal cultures and compare it to DIV7 cultures to assess whether there is an increase in the AcD population over time. We believe that this experiment, combined with the explanations provided above, will sufficiently address the reviewer's question. However, it is important to acknowledge that the establishment of neuronal networks *in vitro* differ from those *in vivo*. Therefore, there may be potential differences in the outcomes.

Outcome:

In the revised manuscript, we rephrased our conclusion saying that the development into the AcD neuronal cell type can occur independently of the *in vivo* environment which does not

exclude a possibility of transitions from nonAcD to AcD types in adult brain. We also included more elaborated discussion regarding this point.

As requested by this reviewer, we indicated in the introduction that the 50% of neurons with AcD morphology are found in the hippocampal CA1 region and discussed possible reasons why there is a lower percentage of AcD neurons in dissociated hippocampal cultures. Also, we quantified the population of AcD neurons in DIV21 hippocampal cultures and found only 10% of neurons were AcD neurons. As mentioned above, this number is not directly comparable to the quantifications done *in vivo*. Further, in the discussion we pointed out that this might be due to the differences in neuronal connectivity between *in vivo* and *in vitro* models, as shown by Lehmann et al. (doi: <https://doi.org/10.1101/2023.07.31.551236>) and that network activity can influence the transition of nonAcD neurons to AcD morphology.

5. I appreciated the balanced discussion of whether this is a stochastic or genetically programmed process. This could have been emphasized earlier in the results since the authors invoke the concept that "...their development must be driven by genetically encoded factors rather than specific...". The authors have not shown this and cannot show it in this system. Indeed, as stated in point 4 above, I think their data argue against a simple genetic program.

Reply: As suggested by the reviewer and noted in point 4, we will revise the section on AcD neuron development in our manuscript to emphasize that hippocampal neurons may adopt AcD morphology through genetic or stochastic mechanisms. While we acknowledge that environmental and activity factors may also influence this process, particularly in mature neurons, our study focuses on developing neurons where genetic and stochastic factors are likely to be predominant. This conclusion is supported by the observation that neurons develop into AcD morphology *in vitro*, where environmental and activity patterns do not mimic those of *in vivo* systems.

Indeed, our current manuscript does not explore genetic factors involved in AcD neuron development. To address this question, one approach could be to label AIS markers endogenously in dissociated cultures using the PORANGE method (see Willems et al., 2020 for reference) or utilize AnkG-GFP transgenic mice (Fréal et al., 2023; Thome et al., 2023) along with a volume marker like mRuby or GFP. This would allow for the identification of AcD and nonAcD neurons *in vivo* and *in vitro*, followed by single-cell transcriptomics analysis to uncover potential genetic factors. Subsequently, candidate genes could be manipulated to demonstrate their essential role in AcD neuron development. However, such experiments require significant time and resources beyond the scope of our current revision timeframe. Nonetheless, this question presents an exciting direction for future research.

Outcome:

Based on the reviewer's suggestion, we now mentioned that AcD neuron development can be genetically encoded or a fully stochastic process in the results section.

Reviewer 3

Major comments:

1. The authors classify neurons into axon-carrying dendrite (AcD) and non-AcD neurons by measuring the stem dendrite length ($> 3 \mu\text{m}$). I could not find the validity for this cut-off. The non-AcD neurons in Fig. 6B appear more AcD to this reviewer, and, in addition, other researchers have proposed a third category of 'shared root' neurons (doi: 10.7554/eLife.76101). For purposes of reproducibility and transparency, please provide first a comprehensive overview of the entire population of morphologies (i.e. all cells in control conditions). The distances from the soma could be plotted in histogram (etc.) and authors may want to think about independent supporting evidence for the cut-off to classify AcD and non-AcD neurons.

Reply: Concerning the validity of AcD neuron classification, we did measure the length of the stem dendrite, as shown in Figure S4G, with an average distance of around $10 \mu\text{m}$. However, we admit that this information is presented relatively late in the manuscript. To address the reviewer's criticism, in the revised version, we will include a supplementary figure displaying a gallery of representative images of both AcD and nonAcD neurons analyzed in our study (please refer to Hodapp et al., 2022; Fig S1 C&D; Fig S3 as an example). Given the sample size of AcD and nonAcD neurons in our study, including all images would result in a very large figure (for example, Figure 1: DIV5: 83 AcD neurons out of 427 cells, DIV7: 47 AcD neurons out of 387 cells), we will only show representative examples of AcD neurons in the gallery. Additionally, as suggested, we will plot the length of the stem dendrite (or axon distance) of AcD neurons as a histogram to demonstrate that the AcD neurons included in our study indeed have a stem dendrite longer than $3 \mu\text{m}$. To further validate the used classification method, we will measure the diameter of the stem dendrite in all analyzed AcD neurons and then compare the distance between the soma and the start of the axon in each analyzed AcD neuron to the diameter of its stem dendrite. As described by Hodapp et al. (Hodapp et al., 2022; Fig S1A), AcD neurons are expected to have a stem dendrite longer than their diameter.

We have considered having independent evidence to support the classification of nonAcD and AcD neurons. However, the method used by Thome et al. and Wahle et al. for AcD and nonAcD neuron classification is well established and widely accepted (see Thome et al., 2014; Wahle et al., 2022 for references). Similar standards were also employed by Benavides-Piccione et al. (Benavides-Piccione et al., 2020). Introducing independent evidence could potentially raise further doubts, so we have chosen to maintain consistency with previous studies.

As for the "shared root" neurons described by Wahle et al., we did not analyze this category separately and included them in the nonAcD subtype. Nonetheless, it is an interesting direction to explore in the future. For completeness, we will discuss this point in the revised manuscript.

Outcome:

As proposed, we updated **Figure S1** and included a gallery with images of nonAcD and AcD neurons (**Figure S1D**), so the reviewer and readers can have more direct comparison. We appreciate this good suggestion.

Further, we measured the axon distance (also considered as length of stem dendrite) of the AcD neurons included in our study and plotted this as a histogram with a box plot to show the median (**Figure S1B**). We also plotted the axon distance against the diameter of the stem dendrite to show that the axon distance is mostly larger than the diameter of stem dendrite diameter (**Figure S1C**). We hope these two plots in combination with the gallery are sufficient to address the reviewer's concern. Of note, previously Thome et al. (doi: 10.1016/j.neuron.2014.08.013) and Hodapp et al. (doi: 10.1126/science.abj1861) classified AcD neurons as neurons having an axon distance longer than 2 μm and larger than the diameter of the stem dendrite. In our study, we increased the threshold of stem dendrite length to longer than 3 μm for AcD neuron classification. The reason for this is to prevent potential subjective bias and reduce an unintended contribution of nonAcD neurons to our AcD dataset. When re-measuring the axon distance of AcD neurons, we noticed that a minor population of neurons have axon distance of 2.2 to 2.7 μm . The axon distance of these neurons is also larger than the diameter, and their morphology also looked similar to AcD neurons where the axon branches out from a MAP2-positive dendrite. To keep our datasets consistent, we therefore decided to use the same classification standards established previously (please refer to Thome et al. doi: 10.1016/j.neuron.2014.08.013 and Hodapp et al. doi: 10.1126/science.abj1861). In the revised manuscript, we clarified this point in the method section.

2. Related to point #1 the primary hippocampal neuron system is excellent for cell biological questions but comes with the drawback of imaginative morphologies including neurons with multiple axons and AISs. It is not mentioned here but literature indicates up to 20% of neurons have two axons (e.g. doi: 10.1007/s12264-017-0169-3, 10.1083/jcb.200707042). How did the authors classify the double axon cells? Since the main hypothesis is the existence of an intrinsic program for AcD neurons (p. 5 top), the two axons from one neuron should develop similarly. The authors can easily test this with the data.

Reply: We appreciate the reviewer's comment regarding the choice of the model system for this type of study. Indeed, as they pointed out, in primary cultures, some neurons develop more than one axon. Since we did not find any supporting evidence from the literature reporting that hippocampal neurons have multiple axons *in vivo*, we only analyzed neurons with one axon for both AcD and nonAcD neurons. We will clarify this in our method section of the revised manuscript.

Outcome:

We clarified this point in the revised manuscript.

3. Some interpretations about function are not correct and the authors should reconsider these. A role of cisternal organelles on neuronal excitability remains to be demonstrated (and see

doi.org/10.1002/cne.21445 showing there is none). In addition, the statement that lower fluorescence intensity of Pan-Nav1 is indicating reduced excitability is flawed. Antibody staining does not scale linearly with voltage-gated sodium channel density and since the AIS of AcD neurons is further from the soma it is most likely smaller in diameter which may account for apparent fluorescent differences. For biophysical reasons (for details I refer to 10.3389/fncel.2019.00570, 10.1016/j.conb.2018.02.016 and 10.7554/eLife.53432) smaller diameter axons will be easier to depolarize by depolarizing voltage-gated channels or excitatory synapses. Finally, in AcD neurons the AIS distance from the soma poses all sorts of interesting cable properties with the soma and the local dendritic membrane and the electrotonic properties alone suffice to make these neurons more excitable.

Reply: The reviewer brings up very valid and important points that we will address in the revised manuscript. First, we will rephrase and adjust our interpretations regarding the functions of the cisternal organelle in the AIS. As also mentioned by reviewer #2, we are aware that antibody staining does not properly reflect Na⁺ channel density. As discussed above, we will also measure other AIS proteins that anchor Na⁺ channels to see if there are any correlations in fluorescence intensity between them and Nav1. We agree with the reviewer that AcD neuron's AIS could have a smaller diameter, resulting in fewer Na⁺ channels. Indirect evidence is already available in the study of Benavides-Piccione et al., showing a smaller axon diameter in AcD neurons compared to nonAcD neurons in both human and mouse brain sections (Figure S4). To test this in our model system, we propose to measure the AIS diameter in AcD neurons. If this is indeed the case, we will indicate it in our revised manuscript and edit the section on Na⁺ channels.

Exploring the biophysical properties of the AIS and axons of AcD neurons is indeed a highly interesting direction to pursue and is the project in its own. It would necessitate the use of computational modeling approaches, which require considerable time and resources that are not feasible within the timeframe of this revision.

Outcome:

As also mentioned by Reviewer#2, we measured the intensity of AnkG and neurofascin at the AIS of AcD neurons and compared to nonAcD neurons. However, we did not find any difference in fluorescent intensity, suggesting the amount of AIS specific membrane associated proteins is similar. Also, after carefully examining our data and including a comparable number of measured neurons for each phenotype, we found that the difference in sodium channel intensity is no longer statistically significant, although the trend remains. We now edited the results in the revised manuscript, and we appreciate the reviewer for pointing this out. Furthermore, we measured the AIS diameter of the AcD neurons as purposed and found it is slightly shorter than nonAcD neurons, but this difference is not statistically significant. This is now mentioned in the revised manuscript (**Figure S1E**).

4. Comparing AcD and non-AcD neurons for AIS plasticity is an excellent idea but the present statistical design is not suitable for answering this question. The authors should directly

compare non-AcD and AcD neurons within a two-way ANOVA design, asking the question whether the independent variable axon type is significantly different and interacts with plasticity.

Related points: 'AIS distance' in Figure 7 seems to refer to something else than distance from soma (Figure 1). Please clarify. What were the absolute distances from the soma for the AcD neurons and was this dependent on treatment?

Reply: We appreciate reviewer's comment and in the revised version we will perform the analysis using two-way ANOVA.

Regarding the terminology and definitions used in our manuscript, the "AIS distance" refers to the measurement between the start of the AIS and the axon initiating point, as depicted in Figure S4 of the manuscript. We adopted this parameter from the previous study by Grubb et al. (Grubb & Burrone, 2010), ensuring consistency in our investigation of AIS plasticity. For AcD neurons, where the axon branches out from the dendrite, we defined the AIS distance as the length between the start of the AIS and the border of the stem dendrite, as illustrated in Figure S4B.

In Figure 1, the term "distance from soma" represents the length of stem dendrite and used for AcD and nonAcD neuron classification. As shown in Figure S4G, the absolute distance from the soma for AcD neurons is approximately 10 μm and remains consistent across treatments. We will explain these points more clearly in the revised manuscript.

Outcome:

We performed two-way ANOVA analysis for the chronic AIS plasticity data. Interestingly, this analysis indicated that the AcD morphology indeed affects the ability of the AIS to undergo an activity-dependent shift. This new result is now included in the revised manuscript (**Figure S4I**).

To better explain the terms "distance from soma" and "AIS distance", we changed the terminology in the revised manuscript. Now, "distance from soma" corresponds to "axon distance" which is also consistent with published terminology (please refer to Thome et al. doi: 10.1016/j.neuron.2014.08.013 and Hodapp et al. doi: 10.1126/science.abj1861). We also included an explanation of the term "axon distance" and the "AIS distance" in results and method sections. We hope this addresses the reviewer's concern and makes the manuscript easier to follow.

Minor comments:

1. At p. 7 is stated that "The percentage of none-AcD forming collaterals at DIV1 is much lower than for AcD neurons" but statistical support is lacking. The conclusion in the next line is that "AcD neurons follow consensus development". That is puzzling given the difference just mentioned before. Please clarify.

We will provide statistical support for comparing collateral formation between nonAcD and AcD neurons at DIV1.

Outcome:

We performed Chi-Square test as statistical support to compare collateral formation between nonAcD and AcD neurons at DIV1. Analysis showed significant differences between AcD and nonAcD neurons. We now indicated this in the revised manuscript.

2. A study not cited in this manuscript showed distinct dendritic morphologies (doi: 10.1073/pnas.1607548113) and AcD interneurons are different for their axonal arborization (doi: 10.1242/dev.202305). Differences in growth of branch arborization could hint to subtypes. Are the AcD and non-AcD neurons different in their adult morphology? A detailed account of the axonal and dendritic trees would strengthen the data.

Reply: Thank you for pointing this out. We will include this citation. In the study by Hodapp et al., it was shown that AcD and nonAcD neurons exhibit similar dendritic morphology and do not differ in spine density, number of dendritic branches, and total dendritic length. However, in hippocampal AcD neurons, the AcD occupies 35% of basal dendrite length, which is longer than basal dendrites in nonAcD neurons (total length), suggesting that AcD neurons do possess specific features in their dendritic trees.

Regarding the axons of AcD neurons, there is currently no detailed study available, and it would be more appropriate to investigate neuronal connectivity through tracing studies in animals rather than in primary cultures. Therefore, this question falls outside the scope of the current manuscript.

Outcome:

Citation is included.

3. Some key references are not included here, and a number of these are mentioned above. In the context of the detailed MT and Rab3A vesicle and cargo transport studies, please acknowledge some of the pioneering work of Alan Peters revealing the ultrastructure of axons emerging from dendrites. See Figs. 5-7 in Peters, Proskauer and Kaiserman-Abramof IR., J Cell Biol 39:604 (1968). What is the identity of the neurons? It makes a difference if the cells are interneurons or pyramidal neurons, CA1 or CA3-like. For plasticity experiments the authors uses cells as independent measurements, but this is inflating the power. How many cultures were used?

Reply: Thank you for pointing this out; we will include the suggested references in the revised manuscript. In our study, we focused on excitatory neurons from the hippocampus. We distinguished neuron types morphologically or with the inhibitory neuron marker GAD1. Identifying CA1, CA2, CA3, and DG subtypes in dissociated culture is more challenging, and this would be an interesting avenue to explore in an in vivo system. Here, we focused on fundamental cell biology aspects related to the AIS structure and its trafficking barrier function,

which should be similar in all these neuron types. While there may be subtype-specific differences in AIS plasticity, investigating this is beyond the scope of our manuscript.

For the plasticity experiments, we used a total of 3 independent cultures, from which we collected a comparable number of neurons. In response to the reviewer's concern, we will also plot the mean of each culture to illustrate the variability of our data points.

Outcome:

In the revised manuscript, we included references suggested by the reviewer. We also changed the graph related to the AIS plasticity including the value from each individual cell and the mean of each experiment. In the figure legend, we indicated the number of cells and independent experiments. To more thoroughly examine our data, we now identified outliers in our plasticity dataset by calculating the interquartile range (IQR) and then performed the same statistical test. Consistently, we got the same results.

Reviewer 4

Major comments:

1. A general limitation of this study is the low N for some critical experiments. In several experiments, individual cells become an N, therefore boosting the power of the analysis when in reality, due to the known heterogeneity of AIS length, position, and general cell morphology in vitro, the aim should be to compare means across animals / preparations, each consisting of a comparable number of individual cells. This is especially important for the analyses of COs, axo-axonic synapses and channel expression at the AIS.

Reply: We would like to mention that this is a cell biological study where neurons are grown in dissociated cultures. To prepare one such culture, we typically use hippocampi from 6-8 E18 rat embryos, which are then mixed in one suspension before plating. The cells are then plated on coverslips in a 12-well plate format. When referring to replicates, for all experiments except for the longitudinal study of 5-day-long time-lapse imaging of developmental sequences (Figure 1), we used between 3 to 6 independent preparations. From each preparation, we took a comparable number of cells derived from 4-6 different coverslips. For each experiment, we measured more than a hundred cells, which is standard practice in the field. To address the issue with individual measurements, in the revised manuscript, we will additionally plot the means of each independent preparation.

Outcome:

To address the reviewer's concern, we have updated most of our graphs in the revised manuscript now including the values of individual cells and means of each independent experiment, so the reviewer and readers can observe the variability of our data. Furthermore, as mentioned above in reply to other reviewers, to rule out false positive results caused by potential outliers in the dataset, we also calculated the interquartile range (IQR) of our dataset to remove outliers and then performed statistical analysis. Except for the sodium channel data, we got the same statistical results for all other experiments. Accordingly, we edited results related to sodium channels as non-significant in the revised version.

2. Such critical parameters as e.g. synaptic innervation at the AIS are investigated in a way that does not support the clear statements given, e.g. "The AIS of AcD neurons receives fewer inhibitory inputs" (Highlights statement) or "AcD neurons have less inhibitory synapses at the AIS" (header of Fig. 6). The overall number of analyzed cells is low (3 and 4 preparations, respectively and approximately 50-cells for each marker). The combination of a pre- and postsynaptic marker for inhibitory / excitatory neurons is a solid decision, but the analysis is not done based on the close approximation of these markers, in 3D, along an AIS, but rather in maxIPs and without any regard of whether pre-and postsynaptic markers are actually close to each other not. The expression of these markers alone just points towards the epitopes being expressed, but are they localized to each other in such a manner that they could form bona fide synapses? The methods are not totally clear on the image depth (tile scans with 5

μm in z will not provide the detail of information to resolve synapses, so how did the authors address the subcellular analysis here and for the CO and VGSCs?). And generally, were Nyquist conditions taken into consideration throughout the study? This can be clarified in text and does not require additional experiments.

Reply to the points highlighted in yellow: The overall number of cells for quantifying inhibitory synapses along the AIS was approximately 80 cells for each synaptic marker. To clarify this, we will indicate the number of cells in the figure legend of our revised manuscript and will additionally plot mean values across independent preparations.

In the current manuscript, our main goal was to provide an initial quantitative measurement of AIS features in AcD neurons to see if they differ from nonAcD neurons. Hence, maxIPs are sufficient for this purpose as they summarize the 3D information. To make our study more comprehensive, following the reviewer's suggestion, we will conduct additional experiments to co-label pre- and post-inhibitory synapses at the AIS with VGAT and gephyrin, respectively. Then, we will image samples in 3D to measure the density as well as the distance between pre- and post-synapses at the AIS of AcD neurons and compare them to nonAcD neurons.

Outcome:

We did additional experiments to quantify co-localised gephyrin and VGAT puncta. The quantification showed a significant reduction of co-localised puncta in the AIS of AcD compared to nonAcD neurons, suggesting there is indeed a difference in the number of inhibitory synapses. To further address the reviewer's concern, we plotted the distance between co-localised gephyrin and VGAT particles. Additionally, all graphs related to inhibitory synapses now include value of each cell and the mean of each experiment. The number of neurons and experiments are indicated in figure legends of the revised manuscript. We also 3D re-constructed re-representative examples of AcD and nonAcD neurons labelled with pre- and post-synaptic markers (VGAT and gephyrin), the AIS marker (TRIM46) and the somatodendritic marker (MAP2).

Furthermore, to address the reviewer's question regarding Nyquist condition relevant for our microscopy images, we indicated the z step size as well as the pixel size in the method section.

2. The chapter on AIS plasticity is certainly an interesting addition to the study, but is a bit superficial, yet reaches strong conclusions ("More importantly, it further indicates that the AIS of AcD neurons is insensitive to activity changes"). This is based on un-physiological concentrations of KCl, and certainly not on network manipulation that truly tests synaptic activity. It also comes back to the 1st point above. A suggestion would be to edit the conclusion.

Reply: KCl treatment globally depolarizes the membrane potential of neurons, leading to an increase in intracellular calcium via voltage-sensitive calcium channels as well as NMDA and AMPA receptors (Rienecker et al., 2020). This protocol has been used in several initial studies describing the plasticity of the AIS (see Evans et al., 2013, 2017; Grubb & Burrone, 2010; Jamann et al., 2021; Muir & Kittler, 2014; Wefelmeyer et al., 2015 for references). Moreover,

as shown by Evans et al. and Grubb et al. (see Evans et al., 2013; Grubb & Burrone, 2010 for references), AIS plasticity is not abolished by TTX, which blocks Na⁺ channels, but is prevented by L-type calcium channel blockers. This suggests that the occurrence of AIS plasticity is independent of action potentials but more sensitive to calcium-related pathways downstream of membrane potential depolarization and post-synaptic activation. Hence, we believe our results are indicative of how the AIS would react when calcium signaling pathways are altered by activity levels. To address the reviewer's concern, we will focus our conclusion more on membrane potential depolarization and calcium signalling and edit our statements.

As discussed above in response to reviewer #3, the quantification of AIS plasticity includes 3 independent preparations, comprising approximately 200 neurons in total. To prevent inflation of statistical power in the analysis, we will also plot the means and standard error of the mean (SEM) for each independent experiment and assess whether any differences persist.

Outcome:

The conclusion related to AIS plasticity in the revised manuscript is now discussed in light of membrane depolarization.

The graphs of AIS plasticity now contain values for each cell and the mean of each experiment (**Figure 7** and **Figure S5**). Moreover, for the AIS plasticity experiment, we also compared means of experiments between different groups and obtained the same results as when comparing individual cells. For consistency with other sections of the manuscript, we did not put this statistical result into the revised manuscript, but it is available upon request.

3. The rationale behind looking at the cisternal organelle (CO) in this study is outlined in the Introduction, where the authors state that "..... and is responsible for calcium handling". What is "calcium-handling" and where is the evidence cited? Furthermore, in the Results, they state that "...both compounds (VGSCs and COs) are critical for the AIS to regulate neuronal excitability". While this is the case for VGSCs, there is no conclusive evidence in the literature whether or not the CO is "critical" for neuronal excitability. In fact, a number of neurons have no CO in the AIS (as much as 50% of all AIS in mouse primary visual cortex for example do not express synpo at the AIS at all, Schlüter et al., 2017). The CO can therefore not be as critical for AP initiation as the authors state. Furthermore, the authors state that "AIS plasticity in excitatory neurons is triggered by calcium signaling". While certainly shown and adequately cited here, other factors (independent of calcium) can also play a role, therefore this statement is a bit absolute and should be edited accordingly.

Reply to points highlighted in yellow:

In contrast to the findings of Schlüter et al. (Schlüter et al., 2019), which were conducted in the mouse primary visual cortex, Sánchez-Ponce et al. showed that nearly 90% of hippocampal neurons contain synaptopodin, the CO marker protein, at the AIS. Furthermore, Schlüter et al. also demonstrated that in the other 50% of neurons containing COs at the AIS, the COs change size during visual deprivation, and their presence correlates with AIS length changes as well

as eye-opening. These observations do suggest that COs are related to neuronal activity. However, this correlation and the formation of COs may be specific to neuro subtypes or require certain triggers. This is another interesting direction to explore, and we will include it in the discussion of the revised manuscript.

Regarding the last point on Ca²⁺ and AIS plasticity, we were not excluding other factors that could potentially participate in AIS plasticity and will also discuss it in the revised version.

Outcome:

We included additional information regarding the number of cisternal organelles in different types of neurons in the discussion. We have also discussed other possible mechanism that can potentially influence the AIS plasticity.

4. The Introduction ends with the rationale of the study, namely that the authors seek to "provide a detailed characterization of the AIS, including its structural and functional properties....". Structure is investigated, but function is limited to the barrier function of the AIS. Since the authors provide no electrophysiology that would really dissect AIS function, I suggest to rephrase this part and focus on transport.

Reply: As suggested, we will certainly emphasize the cargo barrier function of the AIS in AcD neurons in our introduction. But we would like to keep the term "AIS function", because it has already been nicely demonstrated electrophysiologically by previous studies that the plasticity effect of the AIS is very important for maintaining cellular homeostasis.

Outcome:

We emphasized the cargo barrier function of the AIS in AcD neurons in the introduction.

5. The Discussion is more a list of future plans than a context to current data. The authors could move some of the new questions they identify into an "outlook" section at the end? Also, again have a critical look at the literature that is cited and which statements are accurate.

For example, the 2nd phrase in the Discussion states that it was shown that AcD neurons have a "role in memory consolidation", referenced to Hodapp et al., 2022. However, that paper does not provide direct evidence of such a role for AcD neurons. The statement "Collectively, our data provide new insights into the development of AcD neurons and demonstrate that there are differences in AIS functionality between AcD and nonAcD neurons", is not correct. AIS function was not investigated outside of the axonal barrier, and here, the AcD and nonAcD cells do not differ. Also, although the Discussion is geared towards excitatory / glutamatergic neurons, it has been shown by others that interneurons show an even stronger trend to exhibit AcD morphology (work by the Wahle lab and others). This is not clear from the current text (also compare "...AcD neurons being a different subtype if pyramidal neuron").

Further original publications should be included in the paragraph highlighting patch-clamp recordings (see above). In the same context, the statement "...showed that rapid AID plasticity occurs mainly in hippocampal dentate gyrus cells but not in principal excitatory neurons" is not

accurate (see Kim, Kuba, Jamann and others). Generally, the Introduction and Discussion would benefit from a very clear distinction between studies done in vitro versus those done ex vivo or in vivo. This needs to be stated in the Abstract as well.

Methods: For the imaging of synapses, the CO and VGSCs, it is not clear to me from the methods whether Nyquist conditions were applied to produce data that can support the quantification of nanoscale structures. Basing the analysis and interpretation of channel expression on fluorescence intensity profiles is problematic (variance in staining quality from samples to sample, lack of an internal standard). This should be noted in the text. In the text, the first two references given for "Induction of plasticity" do not reference the correct papers.

Reply: Thank you for the valuable suggestions; we will incorporate them into the revised version of the manuscript. The structure will undoubtedly benefit from these improvements. We will also have a further look into our interpretation of the literatures as well as citations during our revision time frame.

Outcome:

In the revised manuscript, we have included an outlook section, we also adjusted our interpretations and added the missing references.

Finally, the text is lacking a discussion of limitations of the study, especially from a methodological point of view. In the Abstract/Summary already, the authors could point out that this is a pure in vitro study. Interestingly, to this day, AIS relocation during plasticity events has only been shown in cell culture systems, and not in vivo. Therefore, this needs to be put into context here - the chosen system is great for the type of imaging approach presented here, but may look at a type of AIS plasticity that is not seen in vivo.

Reply: These are very good points. We will include the limitations of the study in the discussion. Indeed, due to technical and methodological challenges, the relocation of the AIS has not yet been demonstrated using animal models. However, in the study by Wefelmeyer et al. (Wefelmeyer et al., 2015), a similar relocation of the AIS resulting from chronic stimulation was observed in hippocampal organotypic slices, and it was accompanied by reduced excitability of neurons. Furthermore, in the same study, neurons with axons/AIS originating from basal dendrites were also mentioned. However, the measurement of chronic AIS plasticity in their study was not performed based on different classes of neuron types. Hence, our work complements their results. Given that the network connectivity of organotypic slices is much closer to real physiological conditions, it is likely that similar plastic adaptations could occur in vivo.

Outcome:

In the revised manuscript, we included discussions on limitations of our study.

Minor comments

2. The authors may want to add a bit of a technical discussion on the choice of KCl and TTX as triggers for plasticity, especially at the non-physiological concentrations offered here and elsewhere (15 mM KCl).

Reply: We appreciate the reviewer for pointing this out. We will add this in our revised manuscript.

3. Some key statements would benefit from citing the appropriate original literature (some examples would be the original work by Kole, Bender and Brette on the role of the AIS in AP initiation; original work by D'Este and Letterier on the dendritic and axonal scaffold using nanoscopy; work by Kim, Kuba and Jamann on AIS plasticity in vitro and in vivo that is critical for a more informed discussion of AIS plasticity here, and others)

Reply: These are very good points, we will make suggested edits in the revised version.

Outcome:

Regarding minor comments 2 and 3, we have now explained how KCl could induce AIS plasticity. We have also added another reference for AIS plasticity.

4. In the Introduction, the authors word their text explicitly for excitatory neurons. However, AIS plasticity has also been observed in interneurons (work by the Grubb lab for example), and axo-axonic synapses are in fact not all inhibitory - this is an important factor to consider given the embryonic state of the culture material. Does the DIV maturation reflect how axo-axonic synapses "switch" from excitatory to inhibitory in vivo (also see work of the Burrone lab)? Can the conclusions from the paper really be drawn based on this type of system?

Reply: The AIS plasticity was indeed also observed in inhibitory interneurons (see Chand et al., 2015 for reference) and show opposite phenotypes compared to excitatory neurons. Also related to major comment #5, we did take the potential influence of AcD interneurons on the outcome of AIS plasticity experiment into consideration. Therefore, we also did a control experiment where inhibitory interneurons were labelled with GAD1 after chronic KCl treatment and these neurons were excluded from the analysis. Consistently, we got the same results that excitatory AcD neurons do not undergo chronic AIS plasticity. We will include this data in our revised manuscript. Further, in our current manuscript, we decided to focus on excitatory AcD neurons not only because they are the major functional unit in neuronal circuits, but also because the majority of the electrophysiological features were studied in excitatory AcD neurons. But we agree with the reviewer that AcD interneuron is definitely an interesting subject for follow up research in the future.

As mentioned by the reviewer, Pan-Vazquez et al. (Pan-Vazquez et al., 2020) nicely showed that axo-axonic synapses made by GABAergic Chandelier cells (ChCs), depolarise neurons in brain slices obtained from P12-18 animals. But this effect is reversed in slices obtained from older animals (>>P40). Of note, their results were based on cortical neurons but not

hippocampal neurons, hence cell type specificity should be considered. More importantly, previous studies reported that this conversion or switch of GABAergic interneurons from excitatory to inhibitory occurs on hippocampal neurons in P12-13 animals (<https://doi.org/10.1113/jphysiol.1995.sp020882>). In dissociated hippocampal neurons from E18, this switch of GABAergic interneurons takes place on DIV9-11 and completes on DIV19, which should have a comparable neuronal developmental stage as the P12-13 *in vivo* system (see Ganguly et al., 2001 for reference). Therefore, the conclusion could be drawn in an *in vitro* system, but it certainly needs to be validated in *in vivo* system.

Outcome:

We included an additional chronic AIS plasticity experiment where we used the inhibitory marker GAD1 to exclude interneurons which may have the AcD morphology. Consistently, the quantification revealed no difference in the AIS distance in AcD neurons upon induction of chronic AIS plasticity, validating that the above-described effects are occurring in glutamatergic neurons.

5. The authors state that "less COs account for higher intrinsic excitability". Why is that the case?

Reply: According to Yu et al. and Bender et al., Ca²⁺ transient at the AIS regulates the generation of action potentials (APs). For instance, reducing Ca²⁺ transient at the AIS by blocking Ca²⁺ channels with either mibefradil (a T-type Ca²⁺ channel antagonist) or Ni²⁺ (which blocks R- and T-type channels) decreased the number of spikelets evoked by EPSP-like current injection and delayed the timing of spike generation (please see Bender & Trussell, 2009 for details). Therefore, we speculate that Ca²⁺ transients are less affected when there are fewer cisternal organelles (COs) at the AIS, which could have a more direct impact on AP initiation. However, this is just our hypothesis, and there is indeed no direct evidence showing that COs regulate Ca²⁺ dynamics. *We will discuss this in the revised manuscript.*

6. Last but not least, some very recent studies on AcD biology (Stevens, Thome, Lehmann, Wahle) is available online also on preprint servers and may provide additional support for the current study.

Reply: *We will check these pre-prints and include relevant information into the revised version.*

Outcome:

Regarding minor comments 5 and 6, we have discussed possible reasons for why cisternal organelles could be involved in the AIS plasticity and are important for neuronal excitability. We have also included new references related to AcD neurons.

August 30, 2024

RE: JCB Manuscript #202403141R

Prof. Marina Mikhaylova
Humboldt-Universität zu Berlin
AG Optobiology
Invalidenstrasse 42
Berlin, Berlin 10115
Germany

Dear Prof. Mikhaylova:

Thank you for submitting your revised manuscript entitled "Unveiling the cell biology of hippocampal neurons with dendritic axon origin". Your study has been assessed by the original reviewers from Review Commons. As you will see, the majority are supportive of publication of your revised study in JCB. We appreciate reviewer #2's concerns regarding the lack of mechanistic insight. However, given that your work provides insight into a relatively poorly understood structure which should open up future work, conceptually we find this appropriate for JCB. However, it is absolutely essential that all of reviewer #3's concerns about interpretation are thoroughly addressed with precise wording in your final revision. Pending addressing all remaining reviewer points and final revisions necessary to meet our formatting guidelines (see details below) we would be happy to publish your paper in JCB.

A. MANUSCRIPT ORGANIZATION AND FORMATTING:

- 1) Text limits: Character count for Articles is < 40,000, not including spaces. Count includes abstract, introduction, results, discussion, and acknowledgments. Count does not include title page, figure legends, materials and methods, references, tables, or supplemental legends.
- 2) Figures limits: Articles may have up to 10 main text figures.
- 3) Figure formatting: Scale bars must be present on all microscopy images, including inset magnifications. Molecular weight or nucleic acid size markers must be included on all gel electrophoresis.
- 4) Statistical analysis: Error bars on graphic representations of numerical data must be clearly described in the figure legend. The number of independent data points (n) represented in a graph must be indicated in the legend. Statistical methods should be explained in full in the materials and methods. For figures presenting pooled data the statistical measure should be defined in the figure legends. Please also be sure to indicate the statistical tests used in each of your experiments (either in the figure legend itself or in a separate methods section) as well as the parameters of the test (for example, if you ran a t-test, please indicate if it was one- or two-sided, etc.). Also, if you used parametric tests, please indicate if the data distribution was tested for normality (and if so, how). If not, you must state something to the effect that "Data distribution was assumed to be normal but this was not formally tested."
- 5) Abstract and title: The abstract should be no longer than 160 words and should communicate the significance of the paper for a general audience. The title should be less than 100 characters including spaces. Make the title concise but accessible to a general readership.
- 6) Materials and methods: Should be comprehensive and not simply reference a previous publication for details on how an experiment was performed. Please provide full descriptions in the text for readers who may not have access to referenced manuscripts.
- 7) All antibodies, cell lines, animals, and tools used in the manuscript should be described in full, including accession numbers for materials available in a public repository such as the Resource Identification Portal. Please be sure to provide the sequences for all of your primers/oligos and RNAi constructs in the materials and methods. You must also indicate in the methods the source, species, and catalog numbers (where appropriate) for all of your antibodies. Please also indicate the acquisition and quantification methods for immunoblotting/western blots.

8) Microscope image acquisition: The following information must be provided about the acquisition and processing of images:

- Make and model of microscope
- Type, magnification, and numerical aperture of the objective lenses
- Temperature
- Imaging medium
- Fluorochromes
- Camera make and model
- Acquisition software
- Any software used for image processing subsequent to data acquisition. Please include details and types of operations involved (e.g., type of deconvolution, 3D reconstitutions, surface or volume rendering, gamma adjustments, etc.).

10) Supplemental materials: There are strict limits on the allowable amount of supplemental data. Articles may have up to 5 supplemental figures. Please also note that tables, like figures, should be provided as individual, editable files. A summary of all supplemental material should appear at the end of the Materials and methods section.

13) ORCID IDs: ORCID IDs are unique identifiers allowing researchers to create a record of their various scholarly contributions in a single place. Please note that ORCID IDs are now *required* for all authors. At resubmission of your final files, please be sure to provide your ORCID ID and those of all co-authors.

Please note that JCB now requires authors to submit Source Data used to generate figures containing gels and Western blots with all revised manuscripts. This Source Data consists of fully uncropped and unprocessed images for each gel/blot displayed in the main and supplemental figures. File names for Source Data figures should be alphanumeric without any spaces or special characters (i.e., SourceDataF#, where F# refers to the associated main figure number or SourceDataFS# for those associated with Supplementary figures). The lanes of the gels/blots should be labeled as they are in the associated figure, the place where cropping was applied should be marked (with a box), and molecular weight/size standards should be labeled wherever possible. Source Data files will be made available to reviewers during evaluation of revised manuscripts and, if your paper is eventually published in JCB, the files will be directly linked to specific figures in the published article.

Journal of Cell Biology now requires a data availability statement for all research article submissions. These statements will be published in the article directly above the Acknowledgments. The statement should address all data underlying the research presented in the manuscript. Please visit the JCB instructions for authors for guidelines and examples of statements at (<https://rupress.org/jcb/pages/editorial-policies#data-availability-statement>).

B. FINAL FILES:

-- Cover images: If you have any striking images related to this story, we would be happy to consider them for inclusion on the

journal cover. Submitted images may also be chosen for highlighting on the journal table of contents or JCB homepage carousel. Images should be uploaded as TIFF or EPS files and must be at least 300 dpi resolution.

****It is JCB policy that if requested, original data images must be made available to the editors. Failure to provide original images upon request will result in unavoidable delays in publication. Please ensure that you have access to all original data images prior to final submission.****

****The license to publish form must be signed before your manuscript can be sent to production. A link to the electronic license to publish form will be sent to the corresponding author only. Please take a moment to check your funder requirements before choosing the appropriate license.****

Thank you for your attention to these final processing requirements. Please revise and format the manuscript and upload materials within 7 days. If you need an extension for whatever reason, please let us know and we can work with you to determine a suitable revision period.

Thank you for this interesting contribution, we look forward to publishing your paper in Journal of Cell Biology.

Sincerely,

Kassandra Ori-McKenney, PhD
Monitoring Editor

Andrea L. Marat, PhD
Deputy Editor

Journal of Cell Biology

Reviewer #1 (Comments to the Authors (Required)):

The authors have addressed all my points satisfactorily and I have no further comments

Reviewer #2 (Comments to the Authors (Required)):

In this revised paper Han and colleagues investigate primary cultured hippocampal neurons with and without axon carrying dendrites (AcD neurons). The authors have performed a tremendous amount of work and have substantially revised the paper. They are to be commended for this and for considering carefully all reviewers comments. Nevertheless, this reviewer's enthusiasm for the paper is reduced since it is entirely descriptive and provides little or no mechanistic insight into the cell biology of AcDs. In the introduction the authors write: We hypothesized that if AcD neurons form in dissociated cultures, other factors beyond specific neuronal interactions and the extracellular environment must be involved. This is a very interesting hypothesis if the authors actually made some discovery of what these mechanisms or factors are. However, this paper is entirely descriptive and not mechanistic. The details of AcD will be of interest to those working closely in the field of AIS biology, but the details are unlikely to be of great interest to a larger cell biology community. I think this would be a great paper for a more specialized and descriptive journal.

Comment:

Line 550 The authors can't say that they showed that AcD morphology is independent for neuronal connectivity and extracellular environment - these have not been removed. Perhaps they simply mean that AcD development occurs in a non-in vivo environment? After all, there is still a lot of neuronal connectivity and an extracellular environment in cell culture.

Reviewer #3 (Comments to the Authors (Required)):

The authors have satisfactorily addressed some of my comment and the manuscript generally improved. The new

superresolution imaging adds further evidence for a conserved AIS structure of the AcD AISs. What is not addressed, despite previous comments from me and other reviewers, is the erroneous physiological interpretation of the data. The authors are claiming insights into excitability throughout manuscript including the abstract which are wrong, probably rooted in a misunderstanding of neurophysiology. I recommend the authors focus on the cell biological aspects and/or edit the manuscript. A few examples:

1) Throughout the manuscript the authors make assertions like (Line 43) ".. unlike the soma-derived AIS, it does not undergo homeostatic plasticity, contains less cisternal organelles and receives fewer inhibitory inputs. These points of difference may offer insights which cast light on the increased intrinsic excitability of AcD neurons" These statements are not supported by neurophysiological evidence. The "intrinsic" excitability of a neuron refers to the component of neuronal excitability, which is independent of synaptic inputs, and encompasses the passive membrane properties and (voltage-gated) ion channels. That AcD AIS are observed to receive less inhibitory inputs and contain fewer CO is no evidence for increased intrinsic excitability, nor generally increased neuronal excitability.

2) (Line 26 and manuscript) "The development into AcD morphology is independent from neuronal connectivity". This is incorrect. Cultured primary neurons do receive synaptic inputs and are connected (see Figure 6 this manuscript).

3) My original point #3 and similar comments from Reviewer #4 (point #3) were raising concerns about the conceptualization of the cisternal organelle (CO) difference. The literature which was cited in the first reviews was unfortunately not consulted by the authors and the discussion (lines 602-623) is flawed. The authors are mixing up activity-dependent calcium transients (mostly through voltage-gated calcium channels, Bender et al) with cisternal organelles. In brief, COs are organelles involved in cytoplasmic calcium buffering but cannot directly affecting excitability which is confined to membrane associated ion channel mechanisms. If anything, a putative role of CO to excitability will be complex and indirect since it requires local microdomains of calcium-concentration dependent uptake and receptor-mediated release to impact the rapid influx and the calcium- and voltage-gated ion channels in the membrane. At present, there is no simple link found between the number of COs and membrane excitability.

Reviewer #4 (Comments to the Authors (Required)):

Han and colleagues present cell biological data on the development of axon-carrying dendrite (AcD) neurons - cells, in which the axon emerges from a dendrite and not as is more common, from the soma. This particular morphological feature is not new; Cajal already described AcD neurons and since then, their existence was demonstrated in a number of CNS and PNS neurons and across species. However, more recent work from the rodent hippocampus ex vivo and in vivo has pointed towards an interesting functional consequence of AcD morphology, in that these cells can circumvent perisomatic inhibition during network oscillations. Han and colleagues therefore investigate one of the core questions that arises from these previous studies: How is the AcD configuration achieved during development?

I want to congratulate the authors on this comprehensive revision. The data is very compelling now, and this study is a true milestone for the AcD field. The quality of the images, already high to begin with, is outstanding. If I may make one final suggestion - please use the official nomenclature for β IV-spectrin (instead of b4-spectrin).

Reply to reviewers' comments

We really appreciate the feedback from all four reviewers. We would also like to thank the editor for recognising the importance of our work despite lack of the mechanistic insights on why and how AcD neurons are formed. Answers on remaining questions from the reviewers are given below.

For comment from both reviewer 2 and reviewer 3 regarding the development of AcD neuron morphology, we were not trying to exclude the possibility that extracellular environment and connectivity can assist AcD neuron development. Those are in fact very interesting topic for future studies. As also mentioned by the reviewers, there are indeed extracellular environment in dissociated cultures and neurons are synaptically connected. However, it is important to note that the type of connectivity is very random and does not represent the natural organisation of hippocampal circuits. Therefore, primary dissociated neurons are a good model system to address the effect of specific connectivity pattern on the development of AcD morphology. We focused on AcD development at very early age (from DIV 0-5). Dissociated neurons from embryonic rat hippocampus (E18) were plated on glass surface that is homogenously coated with poly-lysine, so there is hardly any specific patterns of extracellular environment (e.g. gradient of guidance molecules) that is comparable to the *in vivo* environment. Also, these neurons do not have synapses yet as synaptogenesis begin at DIV5-7. In this model system, AcD neurons appear already at DIV3. Therefore, it is very unlikely that neuronal connectivity take effects on shaping neuronal morphology at the time our data was recorded. Nevertheless, to prevent further confusion, we now edited our results as "AcD neuron development is not dependent on specific connectivity pattern or a gradient of extracellular cue which are present in *in vivo* environment." We believe our data is strong enough to support this statement, and the fact that AcD neuron do form in dissociated cultures also prove that *in vivo* environment is not absolutely necessary for their development. Also, we mentioned in the revised manuscript that certain intrinsic factors might be involved in AcD neuron development or this is a fully stochastic process. We hope our solution is sufficient to address concerns of the reviewer.

As for the first comment from reviewer 3 regarding our statement of intrinsic excitability, we agree with the reviewer that our data does not provide direct proof related to excitability of AcD neurons. Our results by far may only suggest that there is a potential difference for AcD neuron's AIS to regulate neuronal activities than nonAcD neurons. As suggested by the reviewer, we now revised the manuscript towards cell biology perspective of AcD neurons and the differences of AIS structure between neurons with dendritic and somatic axon onset. To address concerns from reviewer 3 about cisternal organelle and neuronal excitability, we now removed this highly speculative part and only briefly discussed its potential role in triggering AIS plasticity, which has also been shown to be related with calcium signalling. We hope the reviewer is satisfied with our latest version of revised manuscript.

As indicated by reviewer 4, we now have changed the nomenclature for β IV-spectrin in our manuscript. Thanks to the reviewer for pointing this out. Finally, we would like to thank all the reviewers again for their valuable suggestions and comments.